# A neural network model of differentiation and integration of competing memories

**Victoria JH Ritvo**[1], **Alex Nguyen**[2], **Nicholas B Turk-Browne**[3,4], **Kenneth A Norman**[1,2]*

[1]Department of Psychology, Princeton University, Princeton, United States; [2]Princeton Neuroscience Institute, Princeton University, Princeton, United States; [3]Department of Psychology, Yale University, New Haven, United States; [4]Wu Tsai Institute, Yale University, New Haven, United States

**\*For correspondence:**
knorman@princeton.edu

**Competing interest:** The authors declare that no competing interests exist.

**Abstract** What determines when neural representations of memories move together (integrate) or apart (differentiate)? Classic supervised learning models posit that, when two stimuli predict similar outcomes, their representations should integrate. However, these models have recently been challenged by studies showing that pairing two stimuli with a shared associate can sometimes cause differentiation, depending on the parameters of the study and the brain region being examined. Here, we provide a purely unsupervised neural network model that can explain these and other related findings. The model can exhibit integration or differentiation depending on the amount of activity allowed to spread to competitors — inactive memories are not modified, connections to moderately active competitors are weakened (leading to differentiation), and connections to highly active competitors are strengthened (leading to integration). The model also makes several novel predictions — most importantly, that when differentiation occurs as a result of this unsupervised learning mechanism, it will be rapid and asymmetric, and it will give rise to anticorrelated representations in the region of the brain that is the source of the differentiation. Overall, these modeling results provide a computational explanation for a diverse set of seemingly contradictory empirical findings in the memory literature, as well as new insights into the dynamics at play during learning.

## eLife assessment

This paper presents **important** computational modeling work that provides a mechanistic account for how memory representations become integrated or differentiated (i.e., having distinct neural representations despite being similar in content). The authors provide **convincing** evidence that simple unsupervised learning in a neural network model, which critically weakens connections of units that are moderately activated by multiple memories, can account for three empirical findings of differentiation in the literature. The paper also provides insightful discussion on the factors contributing to differentiation as opposed to integration, and makes new predictions for future empirical work.

## Introduction

As we learn, our neural representations change: The representations of some memories move together (i.e. they integrate), allowing us to generalize, whereas the representations of other memories move apart (i.e. they differentiate), allowing us to discriminate. In this way, our memory system plays a delicate balancing act to create the complicated and vast array of knowledge we hold. But ultimately, learning itself is a very simple process: strengthening or weakening individual neural connections.

How can a simple learning rule on the level of individual neural connections account for both differentiation and integration of entire memories, and what factors lead to both outcomes?

Classic supervised learning models (*Rumelhart et al., 1986*; *Gluck and Myers, 1993*) provide one potential solution. These theories posit that the brain adjusts representations to predict outcomes in the world: When two stimuli predict similar outcomes, their representations become more similar; when they predict different outcomes, their representations become more distinct. Numerous fMRI studies have obtained evidence supporting these theories, by utilizing representational similarity analysis to track how the patterns evoked by similar stimuli change over time (e.g. *Schapiro et al., 2013*; *Schapiro et al., 2016*; *Tompary and Davachi, 2017*).

However, other studies have found that linking stimuli to shared associates can lead to differentiation rather than integration. For instance, in one fMRI study, differentiation was observed in the hippocampus when participants were tasked with predicting the same face in response to two similar scenes (i.e. two barns), so much so that the two scenes became less neurally similar to each other than they were to unrelated stimuli (*Favila et al., 2016*; for related findings, see *Schlichting et al., 2015*; *Molitor et al., 2021*; for reviews, see *Brunec et al., 2020*; *Duncan and Schlichting, 2018*; *Ritvo et al., 2019*).

Findings of this sort present a challenge to supervised learning models, which predict that the connection weights underlying these memories should be adjusted to make them more (not less) similar to each other. This kind of 'similarity reversal', whereby stimuli that have more features in common (or share a common associate) show *less* hippocampal pattern similarity, has now been observed in a wide range of studies (e.g. *Favila et al., 2016*; *Schlichting et al., 2015*; *Molitor et al., 2021*; *Chanales et al., 2017*; *Dimsdale-Zucker et al., 2018*; *Wanjia et al., 2021*; *Zeithamova et al., 2018*; *Jiang et al., 2020*; *Fernandez et al., 2023*; *Wammes et al., 2022*).

## Nonmonotonic plasticity hypothesis

How can we make sense of these findings? Supervised learning algorithms cannot explain the aforementioned results on their own, so they need to be supplemented by other learning principles. We previously argued (*Ritvo et al., 2019*) that learning algorithms positing a U-shaped relationship between neural activity and synaptic weight change — where low levels of activity at retrieval lead to no change, moderate levels of activity lead to synaptic weakening, and high levels of activity lead to synaptic strengthening — may be able to account for these results; the Bienenstock-Cooper-Munro (BCM) learning rule (*Bienenstock et al., 1982*; *Cooper, 2004*) is the most well-known learning algorithm with this property, but other algorithms with this property have also been proposed (*Norman et al., 2006*; *Diederich and Opper, 1987*). We refer to the U-shaped learning function posited by this type of algorithm as the nonmonotonic plasticity hypothesis (NMPH; *Detre et al., 2013*; *Newman and Norman, 2010*).

In addition to explaining how individual memories get stronger and weaker, the NMPH also explains how memory representations change with respect to each other as a function of competition during retrieval (*Hulbert and Norman, 2015*; *Norman et al., 2006*). If the activity of a competing memory is low while retrieving a target memory (*Figure 1C*), the NMPH predicts no representational change. If competitor activity is moderate (*Figure 1B*), the connections to the shared units will be weakened, leading to differentiation. If competitor activity is high (*Figure 1A*), the connections to the shared units will be strengthened, leading to integration. Consequently, the NMPH may provide a unified explanation for the divergent studies discussed above: Depending on the amount of excitation allowed to spread to the competitor (which may be affected by task demands, neural inhibition levels, and the similarity of stimuli, among other things), learning may result in no change, differentiation, or integration (for further discussion of the empirical justification for the NMPH, see the Learning subsection in the *Methods*).

## Research goal

In this paper, we present a neural network model that instantiates the aforementioned NMPH learning principles, with the goal of assessing how well these principles can account for extant data on when differentiation and integration occur. In *Ritvo et al., 2019*, we provided a sketch of how certain findings could potentially be explained in terms of the NMPH. However, intuitions about how a complex system should behave are not always accurate in practice, and verbally stated theories can contain

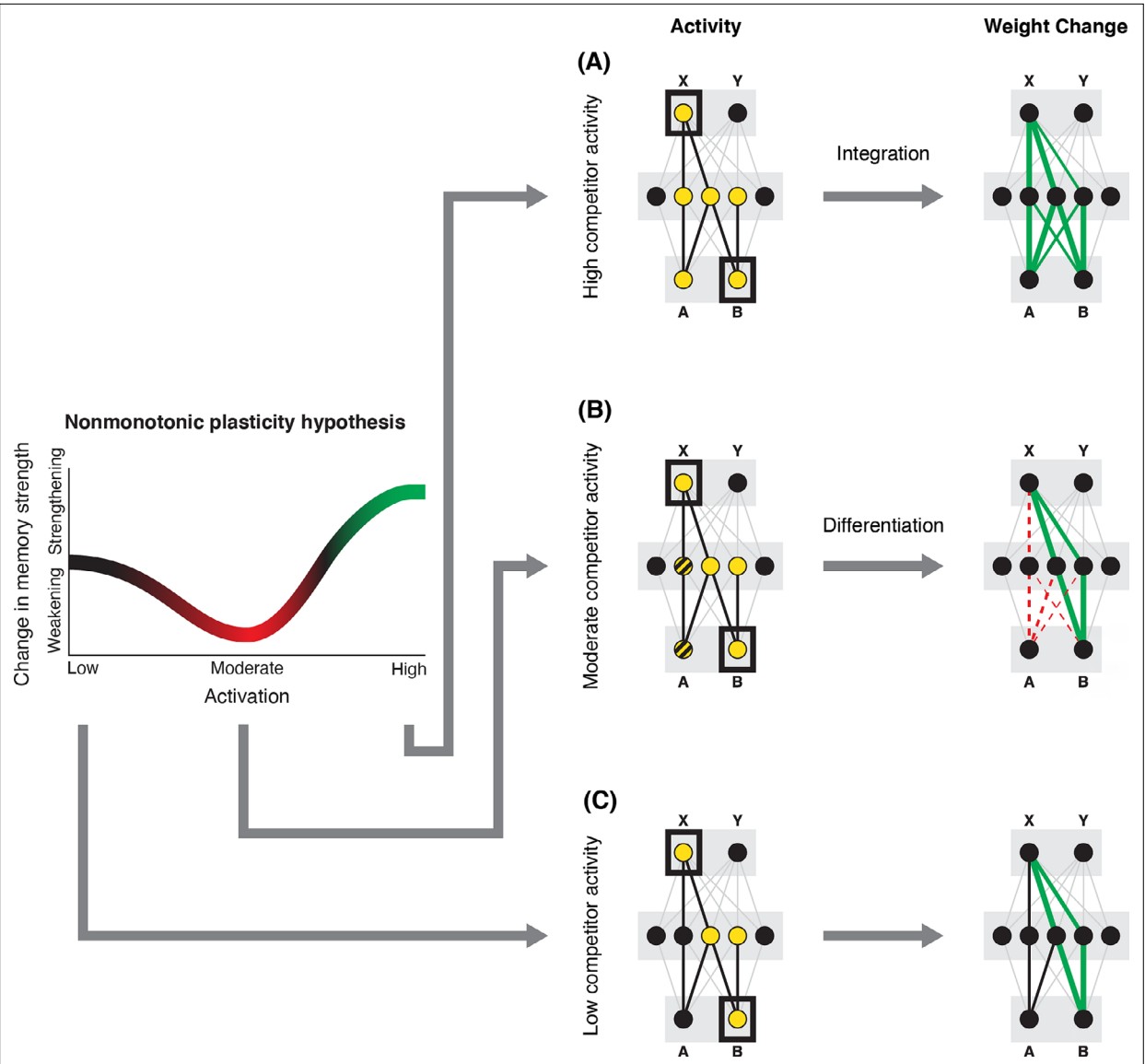

**Figure 1.** Nonmonotonic plasticity hypothesis: A has been linked to X, and B has some initial hidden-layer overlap with A. In this network, activity is allowed to spread bidirectionally. When B is presented along with X (corresponding to a BX study trial), activity can spread downward from X to the hidden-layer units associated with A, and also — from there — to the input-layer representation of A. (**A**) If activity spreads strongly to the input and hidden representations of A, integration of A and B occurs due to strengthening of connections between all of the strongly activated features (green connections indicate strengthened weights; AB integration can be seen by noting the increase in the number of hidden units receiving projections from both A and B). (**B**) If activity spreads only moderately to the input and hidden representations of A, differentiation of A and B occurs due to weakening of connections between the moderately activated features of A and the strongly activated features of B (green and red connections indicate weights that are strengthened and weakened, respectively; AB differentiation can be seen by noting the decrease in the number of hidden units receiving strong connections from both A and B — in particular, the middle hidden unit no longer receives a strong connection from A). (**C**) If activity does not spread to the features of A, then neither integration nor differentiation occurs. Note that the figure illustrates the consequences of differences in competitor activation for learning, without explaining why these differences would arise. For discussion of circumstances that could lead to varying levels of competitor activation, see the simulations described in the text.

ambiguities or internal contradictions that are only exposed when building a working model. Building a model also allows us to generate more detailed predictions (i.e. we can use the model to see what follows from a core set of principles). Relatedly, there are likely boundary conditions on how a model behaves (i.e. it will show a pattern of results in some conditions but not others). Here, we use the

model to characterize these boundary conditions, which (in turn) can be translated into new, testable predictions.

We use the model to simulate three experiments: (1) *Chanales et al., 2021*, which looked at how the amount of stimulus similarity affected the distortion of color memories; (2) *Favila et al., 2016*, which looked at representational change for items that were paired with the same or different associates; and (3) *Schlichting et al., 2015*, which looked at how the learning curriculum (whether pairs were presented in a blocked or interleaved fashion) modulated representational change in different brain regions. We chose these three experiments because each reported that similar stimuli (neurally) differentiate or (behaviorally) are remembered as being less similar than they actually are, and because they allow us to explore three different ways in which the amount of competitor activity can be modulated.

The model can account for the results of these studies, and it provides several novel insights. Most importantly, the model shows that differentiation must happen quickly (or it will not happen at all), and that representational change effects are often asymmetric (i.e. one item's representation changes but the other stays the same).

The following section provides an overview of general properties of the model. Later sections describe how we implemented the model separately for the three studies.

## Basic network properties

We set out to build the simplest possible model that would allow us to explore the role of the NMPH in driving representational change. The model was constructed such that it *only* used unsupervised, U-shaped learning and not supervised learning. Importantly, we do not think that unsupervised, U-shaped learning function is a replacement for error-driven learning; instead, we view it as a supplementary tool the brain uses to reduce interference of competitors (*Ritvo et al., 2019*). Nonetheless, we intentionally omitted supervised learning in order to explore whether unsupervised,

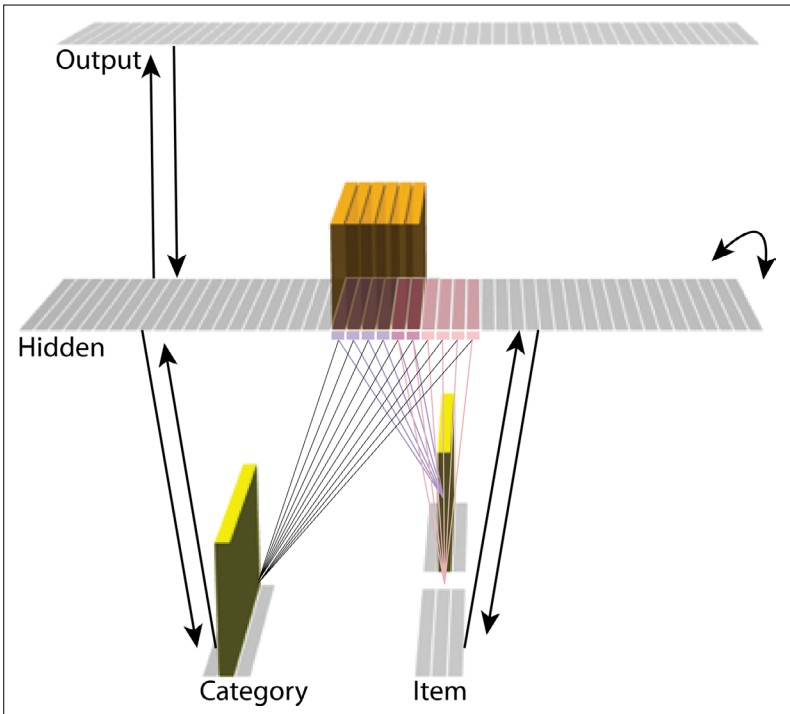

**Figure 2.** Basic network architecture: The characteristics of the model common to all versions we tested, with hidden- and input-layer activity for pairmate A. Black arrows indicate projections common to all versions of the model. All projections were fully connected. Pre-wired, stronger connections between the input A unit (top row of item layer) and hidden A units (purple) are shown, and pre-wired, stronger connections between the input B unit (bottom row of item layer) and hidden B units (pink) are shown. The category unit is pre-wired to connect strongly to all hidden A and B units. Hidden A units have strong connections to other hidden A units (not shown); the same is true for hidden B units. Pre-wired, stronger connections also exist between hidden and output layers (not shown). The arrangement of these hidden-to-output connections varies for each version of the model.

U-shaped learning on its own would be sufficient to account for extant findings on differentiation and integration. Achieving a better understanding of unsupervised learning is an important goal for computational neuroscience, given that learning agents have vastly more opportunities to learn in an unsupervised fashion than from direct supervision (for additional discussion of this point, see, e.g., *Zhuang et al., 2021*).

Because we were specifically interested in the way competition affects learning, we decided to focus on the key moment *after* the memories have been formed, when they first come into competition with each other. Consequently, we pre-wired the initial connections into the network for each stimulus, rather than the allowing the connections to self-organize through learning (see *Methods* for details). Doing so meant we could have control over the exact level of competition between pairmates. We then used the model to simulate different studies, with the goal of assessing how different manipulations that affect competitor activity modulate representational change. In the interest of keeping the simulations simple, we modeled a single task from each study rather than modeling all of them comprehensively. Our goal was to qualitatively fit key patterns of results from each of the aforementioned studies. We fit the parameters of the model by hand as they are highly interdependent (see the *Methods* section for more details).

## Model architecture

The model was built using the Emergent simulation framework (*Aisa et al., 2008*). All versions of the model have the same basic architecture (*Figure 2*; see the following sections for how the model was adapted for each version, and the *Methods* section for the details of all parameters).

We wanted to include the minimal set of layers needed to account for the data. Importantly, the model is meant to be as generic as possible, so none of the layers are meant to correspond to a particular brain region (we discuss possible anatomical correspondences in the *Discussion* section). With those points in mind, we built the model to include four layers: category, item, hidden, and output. The category and item layers are input layers that represent the sensory or semantic features of the presented stimuli; generally speaking, we use the category layer to represent features that are shared across multiple stimuli, and we use the item layer to represent features that are unique to particular stimuli. The hidden layer contains the model's internal representation of these stimuli, and the output layer represents either additional sensory features of the input stimulus, or else other stimuli that are associated with the input stimulus. In all models, the category and item layers are bidirectionally connected to the hidden layer; the hidden layer is bidirectionally connected to the output layer; and the hidden layer also has recurrent connections back to itself. All of these connections are modifiable through learning. Note that the hidden layer has modifiable connections to all layers in the network (including itself), which makes it ideally positioned to bind together the features of individual stimuli.

Each version of the model learns two pairmates, which are stimuli (binary vectors) represented by units in the input layers. The presentation of a stimulus involves activating a single unit in each of the input layers (i.e. the category and item layers). Pairmates share the same unit in the category layer, but differ in their item-layer unit.

We labeled the item-layer units such that the unit in the top row was associated with pairmate A and the unit in the bottom row was associated with pairmate B. Since either pairmate could be shown first, we refer to them as pairmate 1 or 2 when the order is relevant: Pairmate 1 is whichever pairmate is shown on the first trial and pairmate 2 is the item shown second.

All projections as described above have weak, randomly-sampled weight values, but we additionally pre-built some structured knowledge into the network. The item-layer input units have pre-wired, stronger connections to a selection of six pre-assigned 'pairmate A' hidden units and six pre-assigned 'pairmate B' units. The A and B hidden representations overlap to some degree (for instance, with two units shared, as shown in *Figure 2*). We decided to arrange the hidden layer along a single dimension so that the amount of overlap could be easily visualized and interpreted.

Hidden units representing each individual item start out strongly interconnected (that is, the six pairmate A units are linked together by maximally strong recurrent connections, as are the six pairmate B units). We also pre-wired some stronger connections between these hidden A and B units and the output units, but the setup of the hidden-to-output pre-wired connections depends on the modeled experiment.

## Inhibitory dynamics

Within a layer, inhibitory competition between units was enforced through an adapted version of the k-winners-take-all (kWTA) algorithm (*O'Reilly and Munakata, 2000*; see *Methods*) which limits the amount of activity in a layer to at most $k$ units. The kWTA algorithm provides a useful way of capturing the 'set-point' quality of inhibitory neurons without requiring the inclusion of these neurons directly.

The main method we use to allow activity to spread to the competitor is through inhibitory oscillations. Prior work has argued that inhibitory oscillations could play a key role in this kind of competition-dependent learning (*Norman et al., 2006*; *Norman et al., 2007*; *Singh et al., 2022*). Depending on how much excitation a competing memory is receiving, lowering the level of inhibition can allow competing memories that are inactive at baseline levels of inhibition to become moderately active (causing their connections to the target memory to be weakened) or even strongly active (causing their connections to the target memory to be strengthened). Our model implements oscillations through a sinusoidal function which lowers and raises inhibition over the course of the trial, allowing competitors to 'pop up' when inhibition is lower.

## Learning

Connection strengths in the model between pairs of connected units $x$ and $y$ were adjusted at the end of each trial (i.e. after each stimulus presentation) as a U-shaped function of the coactivity of $x$ and $y$, defined as the product of their activations on that trial. The parameters of the U-shaped learning function relating coactivity to change in connection strength (i.e. weakening / strengthening) were specified differently for each projection where learning occurs (bidirectionally between the input and hidden layers, the hidden layer to itself, and the hidden to output layer). Once the U-shaped learning function for each projection in each version of the model was specified, we did not change it for any of the various conditions. Details of how we computed coactivity and how we specified the U-shaped function can be found in the *Methods* section.

## Competition

Constructing the network in this way allows us to precisely control the amount of competition between pairmates. There are several ways beside amplitude of oscillations to alter the amount of excitation spreading to the competitor. For instance, competitor activity could be modulated by altering the pre-wired weights to force the hidden-layer representations for A and B to share more units. Each version of the model (for each experiment) relies on a different method to modulate the amount of competitor activity.

## Model of *Chanales et al., 2021*: repulsion and attraction of color memories

### Key experimental findings

The first experiment we modeled was *Chanales et al., 2021*. The study was inspired by recent neuro-imaging studies showing 'similarity reversals', wherein stimuli that have more features in common (or share a common associate) show less hippocampal pattern similarity (*Favila et al., 2016*; *Schlichting et al., 2015*; *Molitor et al., 2021*; *Chanales et al., 2017*; *Dimsdale-Zucker et al., 2018*; *Wanjia et al., 2021*; *Zeithamova et al., 2018*; *Jiang et al., 2020*; *Wammes et al., 2022*). *Chanales et al., 2021* tested whether a similar 'repulsion' effect is observed with respect to how the specific features of competing events are retrieved. In their experiments, participants learned associations between objects and faces (*Figure 3A*). Specifically, participants studied pairs of objects that were identical except for their color value; each of these object pairmates was associated with a unique face. Participants' memory was tested in several ways; in one of these tests — the color recall task — the face was shown as a cue alongside the colorless object, and participants were instructed to report the color of the object on a continuous color wheel.

*Chanales et al., 2021* found that, for low levels of pairmate color similarity (i.e. 72° and 48° color difference), participants were able to recall the colors accurately when cued with a face and an object. However, when color similarity was increased (i.e. 24° color difference), the color reports were biased *away* from the pairmate, leading to a repulsion effect. For instance, if the pairmates consisted of a red jacket and a red-but-slightly-orange jacket, repulsion would mean that the slightly-orange jacket would be recalled as less red and more yellow than it actually was. When the color similarity was

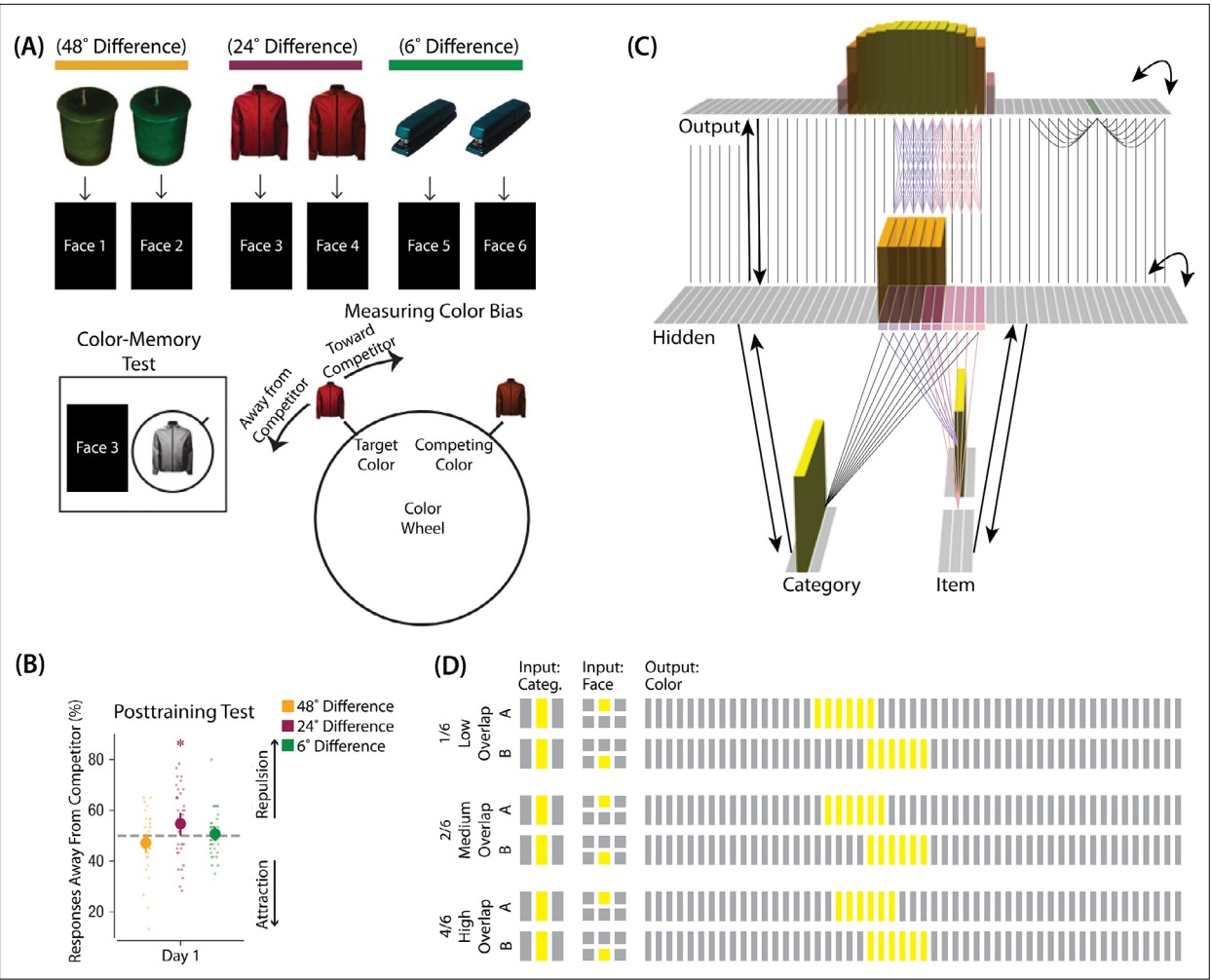

**Figure 3.** Modeling *Chanales et al., 2021*. (**A**) Participants in *Chanales et al., 2021* learned to associate objects and faces (faces not shown here due to bioRxiv rules). The objects consisted of pairs that were identical except for their color value, and the difference between pairmate color values was systematically manipulated to adjust competition. Color memory was tested in a task where the face and colorless object were given as a cue, and participants had to use a continuous color wheel to report the color of the object. Color reports could be biased toward (+) or away from the competitor (-). (**B**) When color similarity was low (48°), color reports were accurate. When color similarity was raised to a moderate level (24°), repulsion occurred, such that color reports were biased systematically away from the competitor. When color similarity was raised further (6°), the repulsion effect was eliminated. (**C**) To model this study, we used the network structure described in *Basic Network Properties*, with the following modifications: This model additionally has a non-modifiable recurrent projection in the output layer, to represent the continuous nature of the color space: Each output unit was pre-wired with fixed, maximally strong weights connecting it to the seven units on either side of it (one such set of connections is shown to the output unit colored in green); background output-to-output connections (outside of these seven neighboring units) were set up to be fixed, weak, and random. The hidden layer additionally was initialized to have maximally strong (but learnable) one-to-one connections with units in the output layer, thereby ensuring that the color topography of the output layer was reflected in the hidden layer. Each of the six hidden A units were connected in an all-to-all fashion to the six pre-assigned A units in the output layer via maximally strong, pre-wired weights (purple lines). The same arrangement was made for the hidden B units (pink lines). Other connections between the output and hidden layers were initialized to lower values. In the figure, activity is shown after pairmate A is presented — the recurrent output-to-output connections let activity spread to units on either side. (**D**) We included six conditions in this model, corresponding to different numbers of shared units in the hidden and output layers. Three conditions are shown here. The conditions are labeled by the number of hidden/output units shared by A and B. Thus, one unit is shared by A and B in 1/6, two units are shared by A and B in 2/6, and so on. Increased overlap in the hidden and output layers is meant to reflect higher levels of color similarity in the experiment. We included overlap types from 0/6 to 5/6.

increased even further (i.e. 6° color difference), the repulsion effects were eliminated (*Figure 3B*). For a related result showing repulsion with continuously varying face features (gender, age) instead of color, see *Drascher and Kuhl, 2022*.

## Potential NMPH explanation

Typical error-driven learning would not predict this outcome, because it would adjust weights to align the guess more closely with the true outcome, leading to no repulsion effects. In contrast, the NMPH potentially explains the results of this study well; as the authors note, "the relationship between similarity and the repulsion effect followed an inverted-U-shape function, suggesting a sweet spot at which repulsion occurs" (*Chanales et al., 2021*).

The amount of color similarity provides a way of modulating the amount of excitation that can flow to the competitor's neural representation. When color similarity is lower (i.e. 72° and 48°), the competing pairmate is less likely to come to mind. Consequently, the competitor's activity will be on the low/left side of the U-shaped function (*Figure 1C*). No weakening or strengthening will occur for the competitor, and both pairmate representations remain intact.

However, when color similarity is higher (i.e. 24°), the additional color overlap means the memories compete more. When the red jacket is shown, the competing red-but-slightly-orange jacket may come to mind moderately, so it falls in the dip of the U-shaped function. If this occurs, the NMPH predicts that the two pairmates will show differentiation, resulting in repulsion in color space.

When color overlap is highest (i.e. 6°), the repulsion effect was eliminated. The NMPH could explain this result in terms of the competitor getting so much excitation from the high similarity that it falls on the high/right side of the U-shaped function, in the direction of integration (or 'attraction' for the behavioral reports of color). *Chanales et al., 2021* did not observe an attraction effect, but this can be explained in terms of the similarity of the pairmates in this condition (at 6° of color separation, there is a limit on how much more similar the color reports could become, given the precision of manual responses).

## Model set-up

### Model architecture

To model this task (*Figure 3C*), we used the two input layers (i.e. category and item) to represent the colorless object and face associates, respectively. To represent a particular stimulus, we activated a single unit in each of the input layers; pairmates share the same unit in the object layer (i.e. 'jacket'), but differ in the unit for the face layer (i.e. 'face-for-jacket-A' and 'face-for-jacket B'). The output layer represents the color-selective units. Units that are closer to each other can be thought of as representing colors that are more similar to each other.

### Knowledge built into the network

As described in *Basic Network Properties*, each of the two input face units is pre-wired to connect strongly to the six corresponding hidden units (either the six hidden A units, or the six hidden B units). In this version of our model, we added several extra pre-wired connections, specifically, recurrent output-to-output connections (although we did not include learning for this projection). Neighboring units were pre-wired to have stronger connections, to instantiate the idea that color is represented in a continuous way in the brain (*Hanazawa et al., 2000*; *Komatsu et al., 1992*).

The six hidden A units were connected to the corresponding six units in the output layer in an all-to-all fashion via maximally strong weights (such that each A hidden unit was connected to each A output unit); an analogous arrangement was made for the units representing pairmate B. Additionally, each non-pairmate unit in the hidden layer was pre-wired to have a maximally strong connection with the unit directly above it in the output layer. Arranging the units in this way (so the hidden layer matches the topography of the output layer) makes it easier to interpret the types of distortion that occur in the hidden layer.

### Manipulation of competitor activity

In this experiment, competition is manipulated through the level of color similarity in each condition. We operationalized this by adjusting the level of overlap between the hidden (and output) color A and

B units. One advantage of modeling is that, because there is no constraint on experiment length, we were able to sample a wider range of overlap types than the actual study, which was limited to three conditions per experiment. Instead, we used six overlap conditions in the model. For all conditions, the two pairmates were each assigned six units in the hidden layer. However, for the different overlap conditions, we pre-wired the weights from the input layers to the hidden layer so that the two pairmates differed in the number of hidden-layer units that were shared (note that this overlap manipulation was also reflected in the output layer, because of the pre-wired connections between the hidden and output layers described above).

We labeled the six conditions based on the number of overlapping units between the pairmates (which varied) and the number of total units per pairmate (which was always six): 0/6 overlap means that the two pairmates are side-by-side but share zero units; 1/6 overlap means they share one unit out of six each; 2/6 overlap means they share two units out of six each, and so on.

### Task

The task simulated in the model is a simplified version of the paradigm used in *Chanales et al., 2021*. Specifically, we focused on the color recall task, where the colorless object was shown alongside the face cue, and the participant had to report the object's color on a color wheel. To model this, the external input is clamped to the network (object and face units), and activity is allowed to flow through the hidden layer to the color layer so the network can make a color 'guess'. We ran a test epoch after each training epoch so we could track the representations of pairmates A and B over time.

As described in *Basic Network Properties*, inhibitory oscillations allow units that are inactive at baseline levels of inhibition to 'pop up' toward the end of the trial. This is important for allowing potential competitor units to activate. There is no 'correct' color shown to the network, and all learning is based purely on an unsupervised U-shaped learning rule that factors in the coactivity of presynaptic and postsynaptic units (see *Methods* for parameter details).

## Results

### Effects of color similarity on color recall and neural representations

*Chanales et al., 2021* showed that, as color similarity was raised, a repulsion effect occurred where the colors of pairmates were remembered as being less similar to each other than they were in reality. When color similarity was raised even further, this repulsion effect went away. In our model, we expected to find a similar U-shaped pattern as hidden-layer overlap increased.

To measure repulsion, we operationalized the color memory 'report' as the center-of-mass of activity in the color output layer at test. Because the topography of the output layer is meaningful for this version of the model (with nearby units representing similar colors), we could measure color attraction and repulsion by the change in the distance between the centers-of-mass (*Figure 4A*): If A and B undergo attraction after learning, the distance between the color centers-of-mass should decrease. If A and B undergo repulsion, the distance should increase.

We found no difference in the color center-of-mass before and after training in the 0/6 and 1/6 conditions. As similarity increased to the 2/6 condition, the distance between the centers-of-mass increased after learning, indicating repulsion. When similarity increased further to the 3/6, 4/6, and 5/6 conditions, the distance between the centers-of-mass decreased with learning, indicating attraction. The overall pattern of results here mirrors what was found in the *Chanales et al., 2021* study: Low levels of color similarity were associated with no change in color perception, moderate levels of color similarity were associated with repulsion, and the repulsion effect went away when color similarity increased further.

The only salient difference between the simulation results and the experiment results is that our repulsion effects cross over into attraction for the highest levels of similarity, whereas *Chanales et al., 2021* did not observe attraction effects. It is possible that *Chanales et al., 2021* would have observed an attraction effect if they had sampled the color similarity space more densely (i.e. somewhere between 24°, where they observed repulsion, and 6°, where they observed neither attraction nor repulsion). As a practical matter, it may have been difficult for *Chanales et al., 2021* to observe attraction in the 6° condition given that color similarity was so high to begin with (i.e. there was no more room for the objects to 'attract').

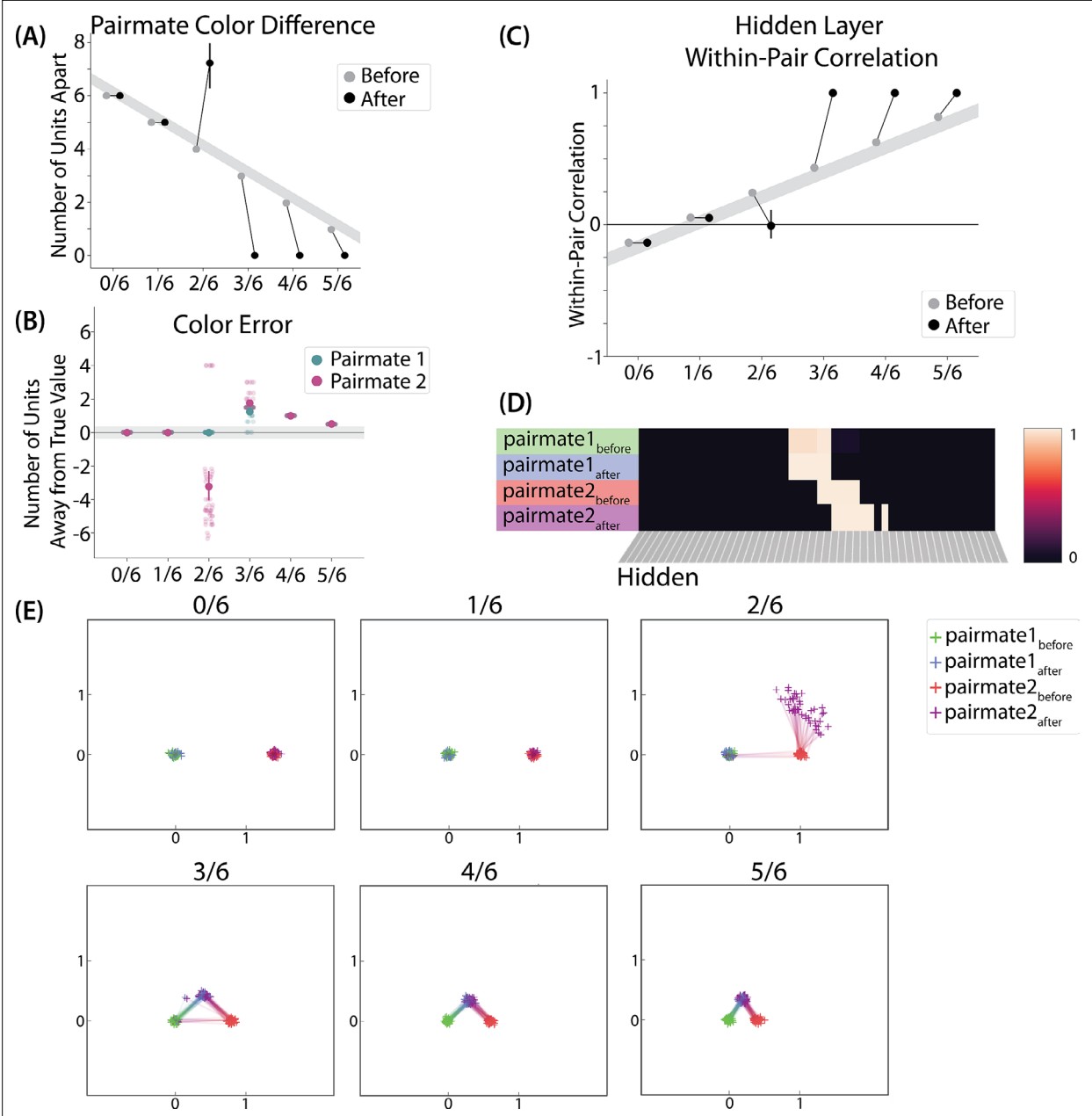

**Figure 4.** Model of *Chanales et al., 2021* results. (**A**) The distance (number of units apart) between the centers-of-mass of output activity for A and B is used to measure repulsion vs. attraction. The gray bar indicates what the color difference would be after learning if no change happens from before (gray dots) to after (black dots) learning; above the gray bar indicates repulsion and below indicates attraction. For lower levels of overlap (0/6 and 1/6), color distance remains unchanged. For a medium level of overlap (2/6), repulsion occurs, shown by the increase in the number of units between A and B. For higher levels of overlap (3/6, 4/6, and 5/6), attraction occurs, shown by the decrease in the number of units between A and B. (**B**) Color error (output-layer distance between the 'guess' and 'correct' centers-of-mass) is shown for each pairmate and condition (negative values indicate repulsion). When repulsion occurs (2/6), the change is driven by a distortion of pairmate 2, whereas pairmate 1 is unaffected. (**C**) Pairmate similarity is measured by the correlation of the hidden-layer patterns before and after learning. Here, above the gray line indicates integration and below indicates repulsion. The within-pair correlation decreases (differentiation) when competitor overlap is moderate (2/6). Within-pair correlation increases (integration) when competitor overlap is higher (3/6, 4/6, and 5/6). (**D**) Four hidden-layer activity patterns from a sample run in the 2/6 condition are shown: pairmate 1_{before}, pairmate 1_{after}, pairmate 2_{before}, and pairmate 2_{after}. The subscripts refer to the state of the memory before/after the learning that occurs in the color recall task; pairmate 1 designates the first of the pairmates to be presented during color recall. Brighter colors indicate the unit is more active. In this run, pairmate 1 stays in place and pairmate 2 distorts away from pairmate 1. (**E**) Multidimensional scaling (MDS) plots for each condition are shown, to illustrate the pattern of representational change in the hidden layer. The same four patterns as in D are plotted for each run. MDS plots were rotated, shifted, and scaled such that pairmate 1_{before} is located at (0,0), pairmate 2_{before} is located directly to the right of pairmate 1_{before}, and the distance

*Figure 4 continued on next page*

*Figure 4 continued*

between pairmate $1_{before}$ and pairmate $2_{before}$ is proportional to the baseline distance between the pairmates. A jitter was applied to all points. Asymmetry in distortion can be seen in 2/6 by the movement of pairmate $2_{after}$ away from pairmate 1. In conditions that integrate, most runs lead to symmetric distortion, although some runs in the 3/6 condition lead to asymmetric integration, where pairmate 2 moves toward pairmate $1_{before}$. For panels A, B, and C, error bars indicate the 95% confidence interval around the mean (computed based on 50 model runs).

We also measured representational change in the model's hidden layer, by computing the Pearson correlation of the patterns of activity evoked by A and B in the hidden layer from before to after learning in the color recall test. If no representational change occurred, then this within-pair correlation should not change over time. In contrast, differentiation (or integration) would result in a decrease (or increase) in within-pair correlation over time.

We found that representational change tracked the repulsion/attraction results *Figure 4C*: Lower overlap (i.e. 0/6 and 1/6), which did not lead to any change in color recall, also did not lead to change in the hidden layer; moderate overlap (i.e. 2/6), which showed repulsion in color recall, also showed a decrease in within-pair correlation (differentiation); and higher overlap (3/6, 4/6, and 5/6), which showed attraction in color recall, also showed an increase in within-pair correlation (integration). The association between behavioral repulsion/attraction and neural differentiation/integration that we observed in the model aligns well with results from *Zhao et al., 2021*. They used a similar paradigm to *Chanales et al., 2021* and found that the level of distortion of color memories was predicted by the amount of neural differentiation for those pairmates in parietal cortex (*Zhao et al., 2021*).

Crucially, we can inspect the model to see why it gives rise to the pattern of results outlined above. In the low-overlap conditions (0/6, 1/6), the competitor did not activate enough during target recall to trigger any competition-dependent learning (see *Video 1*). When color similarity increased in the 2/6 condition, the competitor pop-up during target recall was high enough for co-activity between shared units and unique competitor units to fall into the dip of the U-shaped function, severing the connections between these units. On the next trial, when the competitor was presented, the unique parts of the competitor activated in the hidden layer but — because of the severing that occurred on the previous trial — the formerly shared units did not. Because the activity 'set point' for the hidden layer (determined by the kWTA algorithm) involves having 6 units active, and the unique parts of the competitor only take up 4 of these 6 units, this leaves room for activity to spread to additional units. Given the topographic projections in the output layer, the model is biased to 'pick up' units that are adjacent in color space to the currently active units; because activity cannot flow easily from the competitor back to the target (as a result of the aforementioned severing of connections), it flows instead *away* from the target, activating two additional units, which are then incorporated into the competitor representation. This sequence of events (first a severing of the shared units, then a shift away from the target) completes the process of neural differentiation, and is what leads to the behavioral repulsion effect in color recall (because the center-of-mass of the color representation has now shifted away from the target) — the full sequence of events is illustrated in *Video 2*. Lastly, when similarity increased further (to 3/6 units or above), competitor pop-up increased — the co-activity between the shared units and the unique competitor units now falls on the right side of the U-shaped function, strengthening the connections between these units (instead of being severed). This strengthening leads to neural integration and behavioral attraction in color space, as shown in *Video 3*.

In summary, this version of our model qualitatively replicates the key pattern of results in *Chanales et al., 2021*, whereby increasing color similarity led to a repulsion effect in color recall, which went away as similarity increased further. The reasons why the model gives rise to these effects align well with the *Potential NMPH Explanation*

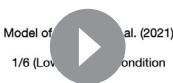

**Video 1.** Model of *Chanales et al., 2021* 1/6 (low overlap) condition. This video illustrates how the competitor does not pop up given low levels of hidden-layer overlap, so no representational change occurs.

https://elifesciences.org/articles/88608/figures#video1

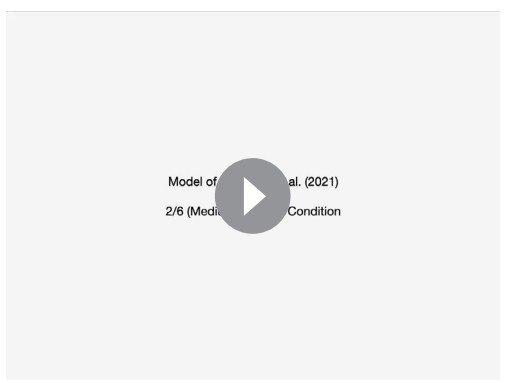

**Video 2.** Model of *Chanales et al., 2021* 2/6 (medium overlap) condition. This video illustrates how the competitor pops up moderately and differentiates given medium levels of hidden-layer overlap.
https://elifesciences.org/articles/88608/figures#video2

provided earlier: Higher color similarity increases competitor activity, which first leads to differentiation of the underlying representations, and then — as competitor activity increases further — to integration of these representations.

## Asymmetry of representational change

A striking feature of the model is that the repulsion effect in the 2/6 condition is *asymmetric*: One pairmate anchors in place and the other pairmate shifts its color representation. *Figure 4B* illustrates this asymmetry, by tracking how the center-of-mass for both pairmates changes over time in the output layer. Specifically, for each pairmate, we calculated the difference between the center-of-mass at the end of learning compared to the initial center-of-mass. A distortion away from the competitor is coded as negative, and toward the competitor is positive. When repulsion occurred in the 2/6 condition, the item shown first (pairmate 1) anchored in place (i.e., the final color report was unchanged and accurate), whereas the item shown second (pairmate 2) moved away from its competitor.

This asymmetry was also observed in the hidden layer (*Figure 4D and E*). In *Figure 4E*, we used multidimensional scaling (MDS) to visualize how the hidden-layer representations changed over time. MDS represents patterns of hidden-layer activity as points in a 2D plot, such that the distance between each pair of points corresponds to the Euclidean distance between their hidden-layer patterns. We made an MDS plot for four patterns per run: the initial hidden-layer pattern for the item shown first (pairmate $1_{before}$), the initial hidden-layer pattern for the item shown second (pairmate $2_{before}$), the final hidden-layer pattern for the item shown first (pairmate $1_{after}$), and the final hidden-layer pattern for the item shown second (pairmate $2_{after}$). Since the 0/6 and 1/6 conditions did not give rise to representational change, the MDS plot unsurprisingly shows that the before and after patterns of both pairmates remain unchanged. For the 2/6 condition (which shows differentiation), the pairmate $1_{before}$ item remains clustered around pairmate $1_{after}$. However, pairmate $2_{after}$ distorts, becoming relatively more dissimilar to pairmate 1, indicating that the differentiation is driven by pairmate 2 distorting away from its competitor. This asymmetry in differentiation arises for reasons described in the previous section — when pairmate 2 pops up as a competitor during recall of pairmate 1, the unique units of pairmate 2 are severed from the units that it (formerly) shared with pairmate 1, allowing it to acquire new units elsewhere given the inhibitory set point. After pairmate 2 has shifted away from pairmate 1, they are no longer in competition, so there is no need for pairmate 1 to adjust its representation and it stays in its original location in representational space (see *Video 2*).

In our model, the asymmetry in differentiation manifested as a clear order effect — pairmate 1 anchors in place and pairmate 2 shifts away from pairmate 1. It is important to remember, however, this effect is contingent on pairmate 2 popping up as a competitor when pairmate 1 is first shown as a target. If the dynamics had played out differently then different results might be obtained. For example, if pairmate 2 does not pop up as a competitor when pairmate 1 is first presented, and instead pairmate 1 pops up as a competitor when pairmate 2 is first presented, we would

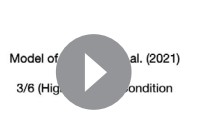

**Video 3.** Model of *Chanales et al., 2021* 3/6 (high overlap) condition. This video illustrates how the competitor pops up strongly and integrates given high levels of hidden-layer overlap.
https://elifesciences.org/articles/88608/figures#video3

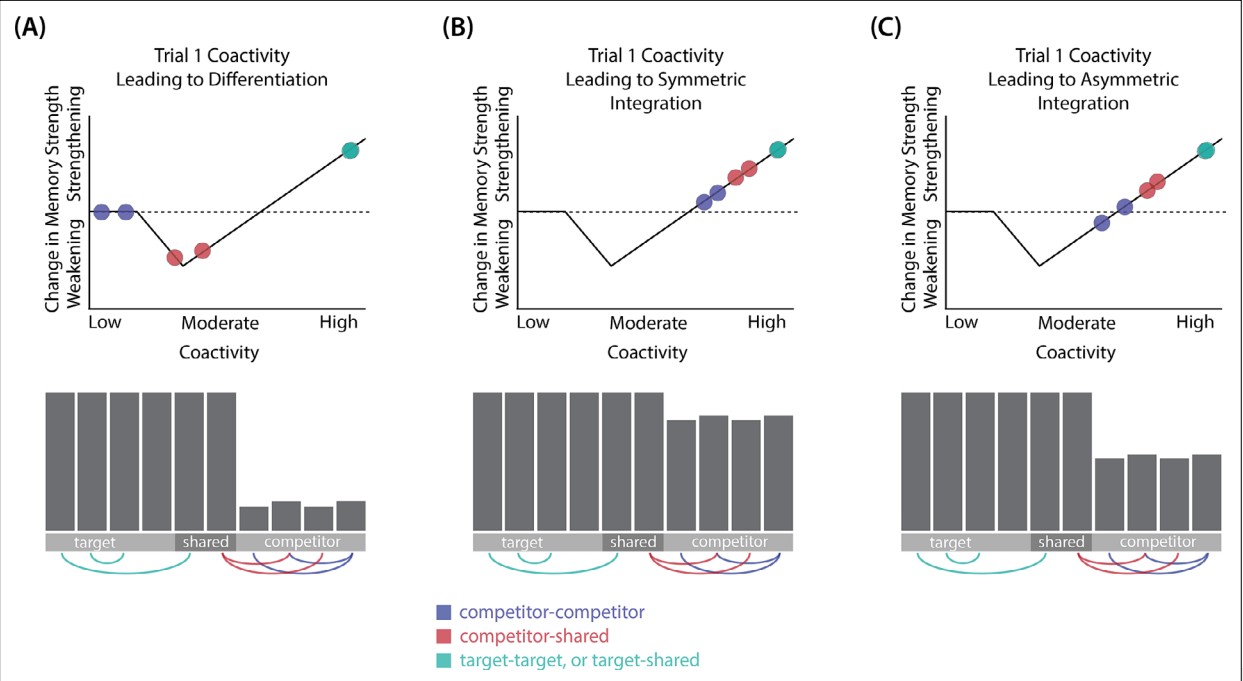

**Figure 5.**
Schematic of coactivities on Trial 1. Generally, coactivity between pairs of units that are unique to the competitor (*competitor-competitor coactivity*) is less than coactivity between unique competitor units and shared units (*competitor-shared coactivity*), which is less than target-target, target-shared, or shared-shared coactivity. The heights of the vertical bars in the bottom row indicate activity levels for particular hidden units. The top row plots coactivity values for a subset of the pairings between units (those highlighted by the arcs in the bottom row), illustrating where they fall on the U-shaped learning function. (**A**) Schematic of the typical arrangement of coactivity in the hidden-hidden connections and item-hidden connections in the 2/6 condition. The competitor units do not activate strongly. As a result, the competitor-shared connections are severed because they fall in the dip of the U-shaped function; this, in turn, leads to differentiation. (**B**) Typical arrangement of coactivity in the higher overlap conditions, 4/6 and 5/6. Here, the competitor units are highly active. Consequently, all connection types fall on the right side of the U-shaped function, leading to integration. Specifically, all units connect to each other more strongly, leading units previously associated with either pairmate A or B to join together. (**C**) As competitor pop-up increases, moving from the situation depicted in panel A to panel B, intermediate levels of competitor activity can result in competitor-competitor coactivity levels falling into the dip of the U-shaped function. If enough competitor-competitor connections weaken, while competitor-shared connections strengthen, this imbalance can lead to an asymmetric form of integration where pairmate 2 moves toward pairmate 1 (see text for details). This happens in some runs of the 3/6 overlap condition.

expect the opposite pattern of results (i.e. pairmate 2 will anchor in place and pairmate 1 will shift away from pairmate 2). We return to these points, and their implications for empirically testing the model's predictions about asymmetry, in the *Discussion* section.

## Two kinds of integration

We also observed that integration could take two different forms, symmetric and asymmetric (*Figure 4B and E*). In this version of the model, the symmetric integration is more common. This can be seen in the MDS plots for conditions 3/6, 4/6, and 5/6: Both pairmate $1_{after}$ and pairmate $2_{after}$ mutually move toward each other. This is because both pairmates end up connecting to all the units that were previously connected to either pairmate individually. Essentially, the hidden units for pairmate 1 and pairmate 2 are 'tied' in terms of strength of excitation, so they are all allowed to be active at once (see Activity and inhibitory dynamics in the *Methods*). However, in the 3/6 condition, some runs show asymmetric integration where pairmate 2 distorts toward pairmate 1. (*Figure 4E*).

Whether symmetric or asymmetric integration occurs depends on the relative strengths of connections between pairs of unique competitor units (*competitor-competitor connections*) compared to connections between unique competitor units and shared units (*competitor-shared connections*) after the first trial (*Figure 5*; note that the figure focuses on connections between hidden units, but the principle also applies to connections that span across layers). Generally, coactivity between unique

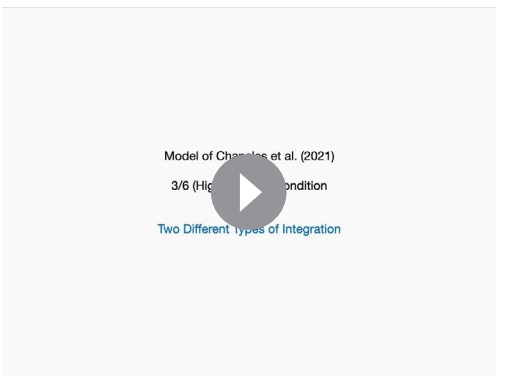

**Video 4.** Two different types of integration. This video illustrates how integration can either be symmetric or asymmetric depending on the amount of competitor pop-up.

https://elifesciences.org/articles/88608/figures#video4

competitor units (*competitor-competitor coactivity*) is less than coactivity between unique competitor units and shared units (*competitor-shared coactivity*), which is less than coactivity between unique target units and shared units (*target-shared coactivity*). In the 2/6 condition (*Figure 5A*), competitor-competitor coactivities fall on the left side of the U-shaped function, and remain unchanged. In the 4/6 and 5/6 conditions, because competitor activity is so high, competitor-competitor coactivities fall on the right side of the U-shaped function (*Figure 5B*). It follows, then, that there is some overlap amount where competitor-competitor coactivities fall in the dip of the U-shaped function (*Figure 5C*). This is what happens in some runs in the 3/6 condition: On trial 1, some competitor-competitor connections (i.e. internal connections within pairmate 2) are severed, while — at the same time — competitor-shared connections are strengthened. On the next trial, when the model is asked to recall pairmate 2, activity flows out of the pairmate 2 representation into the shared units, which are connected to the representations of both pairmate 1 and pairmate 2. Because the representation of pairmate 2 has been weakened by the severing of its internal connections, while the representation of pairmate 1 is still strong, the representation of pairmate 1 outcompetes the representation of pairmate 2 (i.e. pairmate 1 receives substantially more excitation via recurrent connections than pairmate 2) and the neural pattern in the hidden layer 'flips over' to match the original pairmate 1 representation. This hidden-layer pattern is then associated with the pairmate 2 input, resulting in asymmetric integration — from this point forward, both pairmate 2 and pairmate 1 evoke the original pairmate 1 representation (see *Video 4*).

Thus, two kinds of integration can occur — one where both pairmates pick up all units that initially belonged to either pairmate (symmetric), and one where pairmate 2 moves toward pairmate 1 (asymmetric). Generally, as competitor activity is raised, competitor-competitor coactivity is raised, and it is more likely the integration will become symmetric.

## Differentiation

We found that differentiation requires a high learning rate. In our model, the change in connection weights on each trial is multiplied by a learning rate (*LRate*), which is usually set to 1. Lowering the *LRate* value, consequently, leads to smaller learning increments. When we cycle through *LRate* values for all projections other than the output-to-output connection (where *LRate* is zero), we find that differentiation fails to occur in the 2/6 condition if *LRate* is too low (*Figure 6A*): If the connections between competitor (pairmate 2) and shared units are not fully severed on Trial 1, the (formerly) shared units may still receive enough excitation to strongly activate when the model is asked to recall pairmate 2 on Trial 2. This can lead to two possible outcomes: Sometimes the activity pattern in the hidden layer ends up matching the original pairmate 2 representation (i.e. the shared units are co-active with the unique pairmate 2 units), resulting in a re-forming of the original representation. In other cases, asymmetric integration occurs: If connections between pairmate 2 units and shared units have weakened

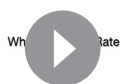

**Video 5.** Effects of lowering learning rate. This video illustrates how differentiation can fail to occur when the learning rate is too low.

https://elifesciences.org/articles/88608/figures#video5

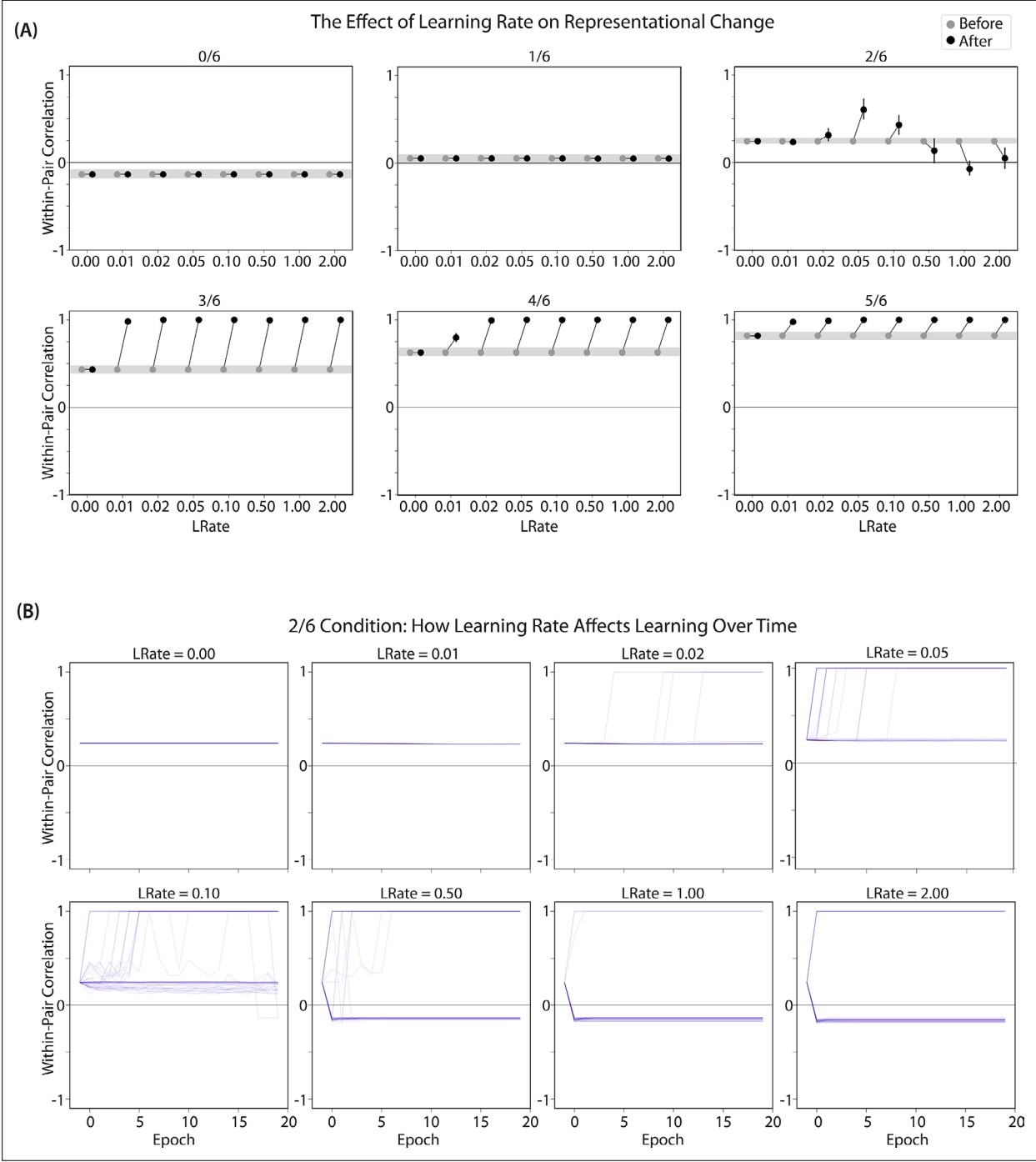

**Figure 6.** Learning rate and representational change. (**A**) The learning rate (*LRate*) parameter was adjusted and the within-pair correlation in the hidden layer was calculated for each overlap condition. In each plot, the gray horizontal bar indicates baseline similarity (prior to NMPH learning); values above the gray bar indicate integration and values below the gray bar indicate differentiation. Error bars indicate the 95% confidence interval around the mean (computed based on 50 model runs). The default *LRate* for simulations in this paper is 1. In the low overlap conditions (0/6 and 1/6), adjusting the *LRate* has no impact on representational change. In the 2/6 condition, differentiation does not occur if the *LRate* is lowered. In the high overlap conditions (3/6, 4/6, and 5/6), integration occurs regardless of the *LRate* (assuming it is set above zero). (**B**) For each *LRate* value tested, the within-pair correlation over time in the 2/6 condition is shown, where each purple line is a separate run (darker purple lines indicate many lines superimposed on top of each other). When *LRate* is set to 0.50 or higher, some model runs show abrupt, strong differentiation, resulting in negative within-pair correlation values; these negative values indicate that the hidden representation of one pairmate specifically excludes units that belong to the other pairmate.

somewhat, while the connections between shared units and pairmate 1 units are still strong, then spreading activity from the (re-activated) shared units on Trial 2 can lead to the original pairmate 1 hidden-layer representation outcompeting the original pairmate 2 hidden-layer representation, in which case the pairmate 2 inputs will become associated with the original pairmate 1 hidden-layer representation. See *Video 5* for illustrations of both possible outcomes. A useful analogy may be escape velocity from astrophysics. Spaceships need to be going a certain speed to escape the pull of gravity. Similarly, the competitor representation needs to get a certain distance away from its pairmate in one trial, or else it will get pulled back in.

A corollary of the fact that differentiation requires a high learning rate is that, when it does happen in the model, it happens abruptly. After competitor-shared connections are weakened, this can have two possible effects. If the learning rate is high enough to sever the competitor-shared connections, differentiation will be evident the next time the competitor is presented. If the amount of weakening is insufficient, the formerly shared units will be reactivated and no differentiation will occur. The abruptness of differentiation can be seen in *Figure 6B*, which shows learning across trials for individual model runs (with different random seeds) as a function of *LRate*: When the learning rate is high enough to cause differentiation, it always happens between the first and second epochs of training. The prediction that differentiation should be abrupt is also supported by empirical studies. For instance, *Wanjia et al., 2021* showed that behavioral expressions of successful learning are coupled with a temporally abrupt, stimulus-specific decorrelation of CA3/dentate gyrus activity patterns for highly similar memories.

In contrast to these results showing that differentiation requires a large *LRate*, the integration effects observed in the higher-overlap conditions do not depend on *LRate*. Once two items are close enough to each other in representational space to strongly coactivate, a positive feedback loop ensues: Any learning that occurs (no matter how small) will pull the competitor closer to the target, making them even more likely to strongly coactivate (and thus further integrate) in the future.

## Pairs of items that differentiate show anticorrelated representations

*Figure 6B* also highlights that, for learning rates where robust differentiation effects occur in aggregate (i.e. there is a reduction in mean pattern similarity, averaging across model runs), these aggregate effects involve a bimodal distribution across model runs: For some model runs, learning processes give rise to anticorrelated representations, and for other model runs the model shows integration; this variance across model runs is attributable to random differences in the initial weight configuration of the model and/or in the order of item presentations across training epochs. The aggregate differentiation effect is therefore a function of the proportion of model runs showing differentiation (here, anticorrelation) and the proportion of model runs showing integration. The fact that differentiation shows up as anticorrelation in the model's hidden layer relates to the learning effects discussed earlier: Unique competitor units are sheared away from (formerly) shared units, so the competitor ends up not having any overlap with the target representation (i.e. the level of overlap is less than you would expect due to chance, which mathematically translates into anticorrelation). We return to this point and discuss how to test for anticorrelation in the *Discussion* section.

### Take-home lessons

Our model of *Chanales et al., 2021* shows that the NMPH can explain the results observed in the study, namely that moderate color similarity can lead to repulsion and that, if color similarity is increased beyond that point, the repulsion is eliminated. Furthermore, our model shows how the behavioral changes in this paradigm are linked to differentiation and integration of the underlying neural representations. The simulations also enrich our NMPH account of these phenomena in several ways, beyond the verbal account provided in *Ritvo et al., 2019*. In particular, the simulations expose some important boundary conditions for when representational change can occur according to the NMPH (e.g. that differentiation depends on a large learning rate, but integration does not), and the simulations provide a more nuanced account of exactly how representations change (e.g. that differentiation driven by the NMPH is always asymmetric, whereas integration is sometimes asymmetric and sometimes symmetric; and that, when differentiation occurs on a particular model run, it tends to give rise to anticorrelated representations in the model's hidden layer).

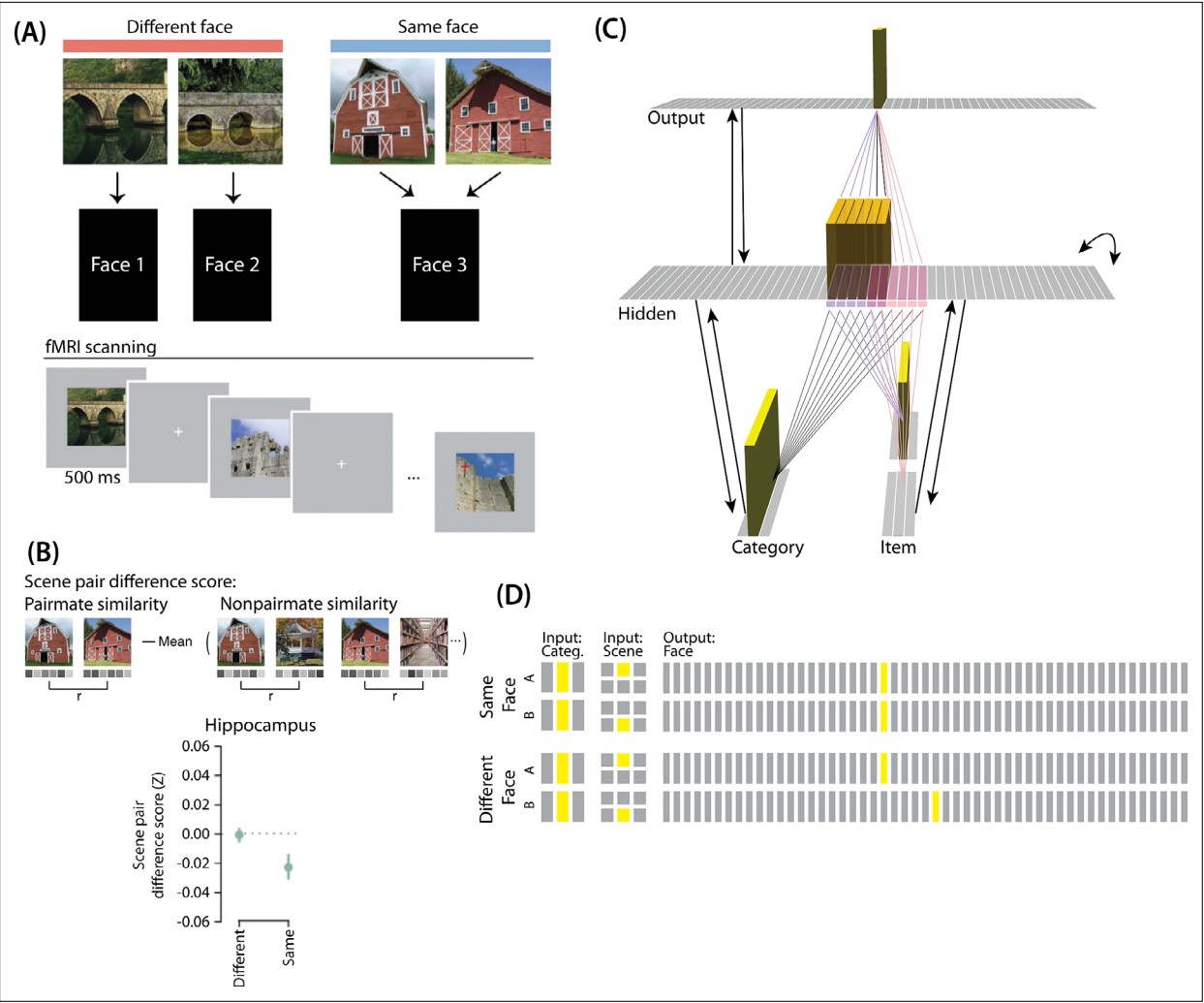

**Figure 7.** Modeling *Favila et al., 2016*. (**A**) Participants learned to associate individual scenes with faces (faces not shown here due to bioRxiv rules). Each scene had a pairmate (another, similar image from the same scene category, e.g., another barn), and categories were not re-used across pairs (e.g. if the stimulus set included a pair of barns, then none of the other scenes would be barns). Pairmates could be associated with the same face, different faces, or no face at all (not shown). Participants were scanned while looking at each individual scene in order to get a measure of neural representations for each scene. This panel was adapted from Figure 1 of *Favila et al., 2016*. (**B**) Neural similarity was measured by correlating scene-evoked patterns of fMRI activity. A scene pair difference score was calculated by subtracting non-pairmate similarity from pairmate similarity; this measure shows the relative representational distance of pairmates. Results for the different-face and same-face condition in the hippocampus are shown here: Linking scenes to the same face led to a negative scene pair difference score, indicating that scenes became less similar to each other than they were to non-pairmates (differentiation). This panel was adapted from Figure 2 of *Favila et al., 2016*. (**C**) To model this study, we used the same basic structure that was described in *Basic Network Properties*. In this model, the category layer represents the type of scene (e.g. barn, bridge, etc.), and the item layer represents an individual scene. The output layer represents the face associate. Activity shown is for pairmate A in the same-face condition. Category-to-hidden, item-to-hidden and hidden-to-hidden connections are pre-wired similarly to the 2/6 condition of our model of *Chanales et al., 2021* (see *Figure 2*). The hidden A and B units have random, low-strength connections to all output units, but are additionally pre-wired to connect strongly to either one or two units in the output layer. In the different-face condition, hidden A and B units are pre-wired to connect to two different face units, but in the same-face condition, they are pre-wired to connect to the same face unit. (**D**) This model has two conditions: same face and different face. The only difference between conditions is whether the hidden A and B units connect to the same or different face unit in the output layer.

There are several aspects of *Chanales et al., 2021* left unaddressed by our model. For instance, they interleaved the color recall task (simulated here) with an associative memory test (not simulated here) where a colored object appeared as a cue and participants had to select the associated face. Our goal was to show how the simplest form of their paradigm could lead to the distortion effects that were observed; future simulations can assess whether these other experiment details affect the predictions of the model.

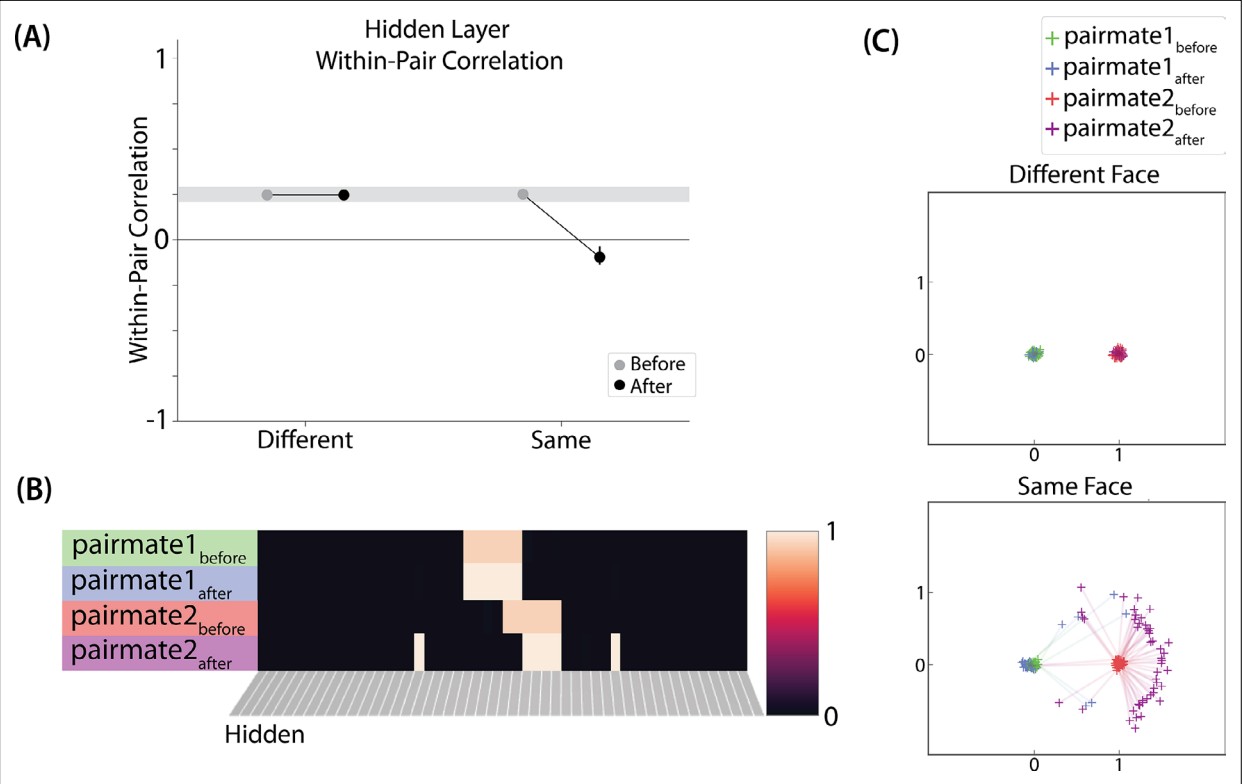

**Figure 8.** Model of *Favila et al., 2016* results. (**A**) Within-pair correlation between A and B hidden layer representations before and after learning. Error bars indicate the 95% confidence interval around the mean (computed based on 50 model runs). In the same-face condition, the within-pair correlation is reduced after learning, indicating differentiation. (**B**) Activity patterns of both pairmates in the hidden layer before and after learning for a sample "same-face" run are shown. Asymmetry in distortion can be seen in how pairmate 1's representation is unchanged and pairmate 2 picks up additional units that did not previously belong to either item (note that there is no topography in the hidden layer in this simulation, so we would not expect the newly-acquired hidden units to fall on one side or the other of the layer). (**C**) MDS plots for each condition illustrate representational change in the hidden layer. The differentiation in the same-face condition is asymmetric: Pairmate 2$_{after}$ generally moves further away from pairmate 1 in representational space, while pairmate 1 generally does not change.

The online version of this article includes the following figure supplement(s) for figure 8:

**Figure supplement 1.** Results from an alternative parameterization, where the different-face condition shows some differentiation and the same-face condition shows more differentiation.

## Model of *Favila et al., 2016*: similar and different predictive associations

### Key experimental findings

*Favila et al., 2016* provided neural evidence for differentiation following competition, using a shared associate to induce competition between pairmates. In this study (*Figure 7A*), participants were instructed to learn scene-face associations. Later, during the repeated face-test task, participants were shown a scene and asked to pick the correct face from a bank of faces. Scenes were made up of highly similar pairs (e.g. two bridges, two barns). Sometimes, two paired scenes predicted the same face, sometimes different faces, and sometimes no face at all (in the latter case, the paired scenes appeared in the study task and not the face-test task). Participants were never explicitly told that some scene pairs shared a common face associate.

When the two scenes predicted different faces, the hippocampal representations for each scene were relatively orthogonalized — hippocampal representations of the two pairmate scenes were just as similar to each other as to non-pairmate scenes. However, when the two scenes predicted the same face, differentiation resulted, such that the hippocampal representations of the two pairmate scenes were less similar to each other than to non-pairmate scenes (*Figure 7B*).

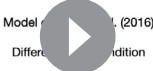

**Video 6.** Model of *Favila et al., 2016* different face condition. This video illustrates how no representational change occurs in the different face condition of our simulation of *Favila et al., 2016*.
https://elifesciences.org/articles/88608/figures#video6

## Potential NMPH explanation

As noted earlier, these results contradict supervised learning models, which predict that pairing two stimuli with the same associate would lead to integration, not differentiation. The NMPH, however, can potentially explain these results: Linking the pairmates to a shared face associate provides an additional pathway for activity to spread from the target to the competitor (i.e. activity can spread from scene A, to the shared face, to scene B). If competitor activity falls on the left side of the U-shaped function in the different-face condition, then the extra spreading activity in the same-face condition could push the activity of the competitor into the 'dip' of the function, leading to differentiation.

## Model set up

### Model architecture

Our model of *Chanales et al., 2021* can be adapted for *Favila et al., 2016*. Instead of mapping from an object and face to a color, as in *Chanales et al., 2021*, the *Favila et al., 2016* study involves learning mappings between scenes and faces. Also, the way in which competition is manipulated is different across the studies: In *Chanales et al., 2021*, competition is manipulated by varying the similarity of the stimuli (specifically, the color difference of the objects), whereas in *Favila et al., 2016*, competition is manipulated by varying whether pairmate scenes are linked to the same vs. different face.

To adapt our model for *Favila et al., 2016*, the interpretation of each layer must be altered to fit the new paradigm (*Figure 7C*). The category layer now represents the category for the scene pairmates (e.g. barn, bridge, etc.). The item layer represents the individual scene (e.g. where A represents barn 1 and B represents barn 2). Just as before, the input for each stimulus is composed of two units (one in each of the two input layers); the category-layer unit is consistent for A and B, but the item-layer unit differs.

The output layer in this model represents the face associate, such that each individual output-layer unit could be thought of as a single face. For this model, the ordering of units in the output layer is not meaningful — two units next to each other in the output layer are no more similar than to any other units.

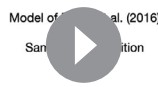

**Video 7.** Model of *Favila et al., 2016* same face condition. This video illustrates how differentiation occurs in the same face condition of our simulation of *Favila et al., 2016*.
https://elifesciences.org/articles/88608/figures#video7

### Knowledge built into the network

As before, we were interested in the moment that competition first happens, so we pre-wired connections as if some initial learning had occurred. We pre-wired the connections between the hidden layer and both input layers (and from hidden layer to itself) to be similar to the 2/6 condition of our model of *Chanales et al., 2021*, so there is some baseline amount of overlap between A and B (i.e. reflecting the similar-looking scenes).

To mimic the learning of scene-face associates, all hidden A units are connected to a single unit in the face layer, and all hidden B units are connected to a single unit in the face layer. In the different-face condition, A and B hidden units are connected to different face units, to reflect that the two scenes were predictive of two separate

faces. In the same-face condition, A and B hidden units connect to the same face unit, to reflect that A and B predict the same face.

## Manipulation of competitor activity

Competitor activity is modulated by the similarity of predictive consequences in this version of the model — that is, whether the hidden units for pairmates A and B are pre-wired to connect strongly to the same unit or different units in the output layer. Stronger connections to the same face unit should provide an extra conduit for excitation to flow to the competitor units.

## Task

The task performed by the model was to guess the face associated with each scene. We clamped the external input for the scenes and allowed activity to spread through the hidden layer to the output layer so it could make a guess for the correct face. No correct answer was shown, and no error-driven learning was used. Although the exact parameter values used in this simulation were slightly different from the values used in the previous simulation (see *Methods* for details), inhibition and oscillations were implemented in the same way as before.

## Results

For this study, the key dependent measure was representational change within the hidden layer. Specifically, we sought to capture the hippocampal pattern similarity results reported by *Favila et al., 2016*.

### Differentiation and integration

The different-face condition in this model led to no representational change (see *Video 6*) whereas the same-face condition led to differentiation (see *Video 7*), as measured using within-pair correlation (*Figure 8A*). Differentiation is indicated by the fact that new units are added that did not previously belong to either pairmate. In this version of the model, the topography of the hidden layer is not meaningful other than the units assigned to A and B, so the new units that are added to the representation could be on either side. The reason why the model shows differentiation in the same-face condition (but not in the different-face condition) aligns with the *Potential NMPH Explanation* provided earlier: The shared face associate in the same-face condition provides an conduit for extra activity to spread to the competitor scene pairmate, leading to moderate activity that triggers differentiation. Note also that the exact levels of differentiation that are observed in the different-face and same-face conditions are parameter dependent; for an alternative set of results showing some differentiation in the different-face condition (but still less than is observed in the same-face condition), see *Figure 8— figure supplement 1*.

### Nature of representational change

The representational change in the hidden layer shows the same kind of asymmetry that occurred in our model of *Chanales et al., 2021*; *Figure 8B and C*: In the same-face condition, pairmate 1 typically anchors in place, whereas pairmate 2 acquires new units that did not previously belong to either either pairmate (*Figure 8B*), resulting in it shifting away from pairmate 1 (*Figure 8C*). Although this pattern is present on most runs, the MDS plot also shows that some runs fail to differentiate and instead show integration (see the *Discussion* for an explanation of how conditions that usually lead to differentiation may sometimes lead to integration instead). Note that the neural measure of differentiation used by *Favila et al., 2016* does not speak to the question of whether representational change was symmetric or asymmetric in their experiment — to measure the (a)symmetry of representation change, it is necessary to take 'snapshots' of the representations both before and after learning (e.g. *Schapiro et al., 2012*), but the method used by *Favila et al., 2016* only looked at post-learning snapshots (comparing the neural similarity of pairmates and non-pairmates). We return in the *Discussion* to this question of how to test predictions about asymmetric differentiation.

*Figure 8—figure supplement 1* also indicates that, as in our simulation of *Chanales et al., 2021*, individual model runs where differentiation occurs show anticorrelation between the pairmate

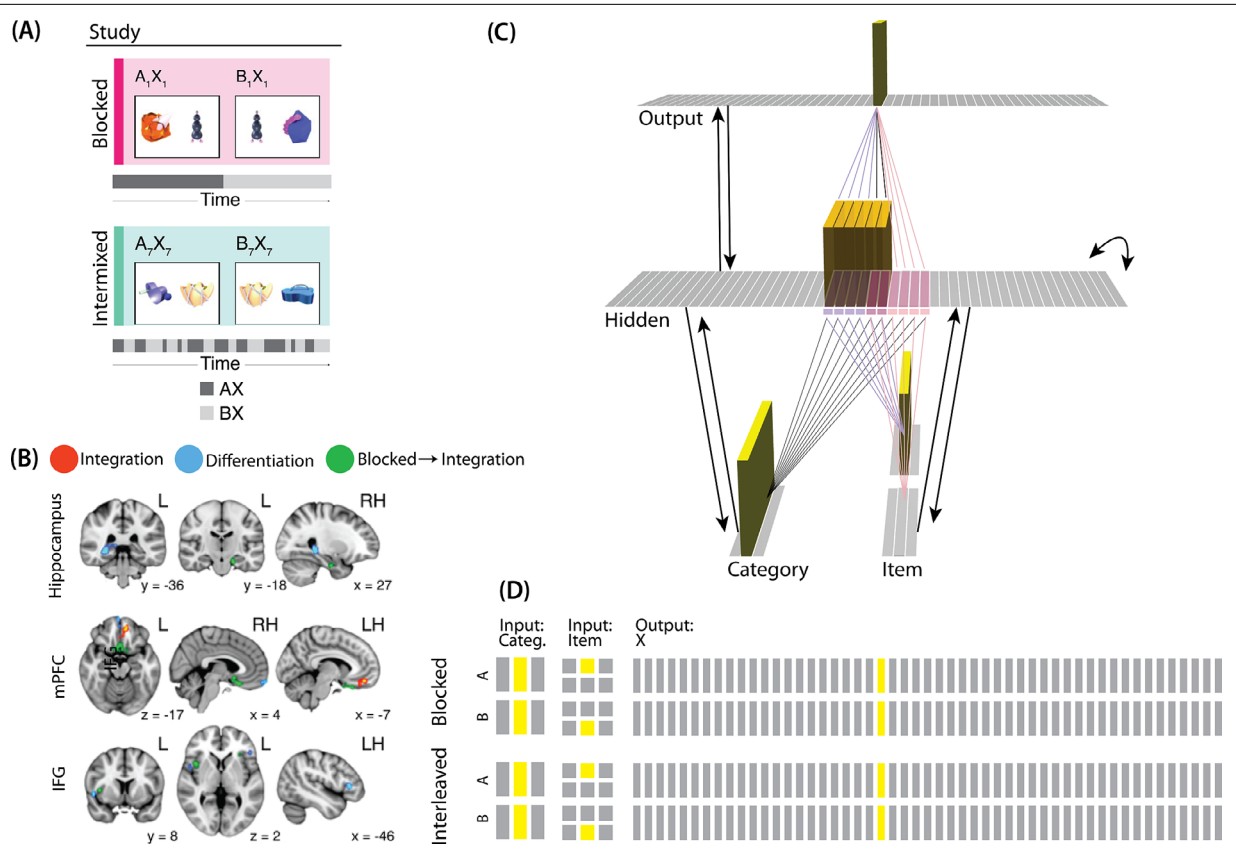

**Figure 9.** Modeling *Schlichting et al., 2015*. (**A**) Participants in *Schlichting et al., 2015* learned to link novel objects with a common associate (i.e., AX and BX). Sometimes these associates were learned in a blocked design (i.e. all AX before any BX), and sometimes they were learned in an interleaved design. The items were shown side-by-side, and participants were not explicitly told the structure of the shared items. Before and after learning, participants were scanned while observing each item alone, in order to get a measure of the neural representation of each object. This panel was adapted from Figure 1 of *Schlichting et al., 2015*. (**B**) Some brain regions (e.g. right posterior hippocampus) showed differentiation for both blocked and interleaved conditions, some regions (e.g. left mPFC) showed integration for both conditions, and other regions (e.g. right anterior hippocampus) showed integration in the blocked condition but differentiation in the interleaved condition. This panel was adapted from Figures 3, 4 and 5 of *Schlichting et al., 2015*. (**C**) To model this study, we used the network structure described in *Basic Network Properties*, with the following modifications: The structure of the network was similar to the same-face condition in our model of *Favila et al., 2016* (see *Figure 7*), except we altered the connection strength from units in the hidden layer to the output unit corresponding to the shared item (item X) to simulate what would happen depending on the learning curriculum. In both conditions, the pre-wired connection linking the item B hidden units to the item X output unit is set to .7. In the interleaved condition, the connection linking the item A hidden units to the item X output unit is set to .8, to reflect some amount of initial AX learning. In the blocked condition, the connection linking the item A hidden units to the item X output unit is set a higher value (0.999), to reflect extra AX learning. (**D**) Illustration of the input and output patterns, which were the same for the blocked and interleaved conditions (the only difference in how we modeled the conditions was in the initial connection strengths, as described above).

representations, and gradations in the aggregate level of differentiation that is observed across conditions reflect differences in the proportion of trials showing this anticorrelation effect.

## Differentiation requires a high learning rate

As in our model of *Chanales et al., 2021*, we again found that a high *LRate* is needed for differentiation. Specifically, lowering *LRate* below its standard value (e.g, to a value of 0.10) eliminated the differentiation effect in the same-face condition. Changing the learning rate did not impact the different-face condition. In this condition, the pop-up is low enough that all competitor-shared connections on Trial 1 fall on the left side of the U-shaped function, so no weight change occurs, regardless of the learning rate setting.

## Take-home lessons

This simulation demonstrates that the NMPH can explain the results of *Favila et al., 2016*, where learning about stimuli that share a paired associate can lead to differentiation. The model shows how

linking items with the same or different associates can modulate competitor activity and, through this, modulate representational change. As in our simulation of *Chanales et al., 2021*, we found that the NMPH-mediated differentiation was asymmetric, manifested as anticorrelation between pairmate representations on individual model runs, and required a high learning rate, leading to abrupt representational change.

## Model of *Schlichting et al., 2015*: blocked and interleaved learning

### Key experimental findings

For our third simulation, we focused on a study by *Schlichting et al., 2015*. This study examined how the learning curriculum affects representational change in different brain regions. Participants learned to link novel objects with a common associate (i.e. AX and BX). Sometimes these associates were presented in a blocked fashion (all AX before any BX) and sometimes they were presented in an interleaved fashion (*Figure 9A*).

The analysis focused on the hippocampus, the medial prefrontal cortex (mPFC), and the inferior frontal gyrus (IFG). Some brain regions (e.g. right posterior hippocampus) showed differentiation for both blocked and interleaved conditions, some regions (e.g. left mPFC) showed integration for both conditions, and others (e.g. right anterior hippocampus) showed integration in the blocked condition but differentiation in the interleaved condition (*Figure 9B*).

### Potential NMPH explanation

The NMPH can potentially explain how the results differ by brain region, since the overall level of inhibition in a region can limit the competitor's activity. For instance, regions that tend to show differentiation (like posterior hippocampus) have sparser activity (*Barnes et al., 1990*). Higher inhibition in these areas could cause the activity of the competitor to fall into the moderate range, leading to differentiation. For regions with lower inhibition, competitor activity may fall into the high range, leading to integration.

The result that some regions (e.g. right anterior hippocampus) show differentiation in the interleaved condition and integration in the blocked condition could also be explained by the NMPH. By the first BX trial, we would expect that the connections between A and X would be much stronger in the blocked condition (after many AX trials) compared to the interleaved condition (after one or a few AX trials). This stronger A-X connection could allow more activity to flow from B through X to the A competitor. Consequently, competitor activity in the blocked condition would fall farther to the right of the U-shaped function compared to the interleaved condition, allowing for integration in the blocked condition but differentiation in the interleaved condition.

### Model set up

#### Model architecture

In *Schlichting et al., 2015*, A, B, and X were all individual pictures of objects shown in the same fashion; the only difference is that X was paired with two distinct items (A and B) but A and B were only paired with one other item (X). We decided to represent A and B pairmates in the input layer and X as a single unit in the output layer. Concretely, we represented A and B in the input layer by having a unique (single) unit in the item layer for each of A and B; the same (single) unit is active in the category layer for both stimuli (it can be thought of as representing the category of 'novel objects' to which both A and B belong). Putting A and B in the same layer, where their representations directly compete, captures the fact that A and B were mutually exclusive (i.e. they were never viewed together). Putting X in a different layer made it easier for the model to represent X at the same time as A or B. Note that, because connections in the model are bidirectional and symmetric in strength, the X output was able to influence activity elsewhere in the network, which proves to be critical for the learning effects described below. This approach also makes the isomorphism to *Favila et al., 2016* clearer — just as a shared face associate in that study served as a conduit for activity to spread to the scene pairmate, the shared X object associate in *Schlichting et al., 2015* serves as a conduit for activity to spread between B and A pairmates.

#### Knowledge built into the network

The connections between layers for this model are similar to the connections we used in the same-face condition in our model of *Favila et al., 2016*. Since both A and B have a shared associate (X), all

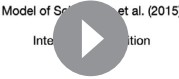

**Video 8.** Model of *Schlichting et al., 2015* interleaved condition. This video illustrates how differentiation occurs in the interleaved condition of our simulation of *Schlichting et al., 2015*.
https://elifesciences.org/articles/88608/figures#video8

hidden A and B units are connected to the same unit in the output layer. However, we adjusted the connection strength to the shared X unit in order to simulate blocked or interleaved learning (see *Figure 9C* caption for details).

Matching the previous two simulations, we pretrained the weights so the hidden representations of the stimuli initially had 2/6 units in common. Even though the A and B stimuli used in the actual experiment did not have obvious feature overlap (they were randomly selected novel objects), it is important to note that the hidden layer is not simply a representation of the sensory features of the A and B stimuli; the hidden layer also receives input from the output layer, which represents the shared associate of A and B (X). We think that the presence of this shared associate justifies our use of initially-overlapping hidden representations.

## Modulation of competitor activity

We hypothesized that competition would be modulated by the blocked vs. interleaved manipulation. If AX and BX are learned in a blocked design, then the connections between A (in the hidden layer) and X (in the output layer) will be stronger before the first BX trial than in the interleaved condition. That extra strength can provide a pathway for more excitation to flow from X to A during BX trials. In our simulations, we operationalized this difference by varying the strength of the connection between A units (in the hidden layer) and X units (in the output layer; see *Figure 9* caption for details).

## Task

External inputs corresponding to the A or B stimuli (given by the two input layers) are clamped, and activity is allowed to flow through the hidden layer to the output layer, producing a guess of its associate. This deviates somewhat from the task in the original study, which showed AX or BX stimuli side-by-side. We could have clamped the X external values so it would be shown rather than guessed, but we decided to let the activity flow through the network to be consistent with the other versions of our model. This change does not have any impact on the outcome of the simulation — X ends up being strongly active here even though it is not being externally clamped, so the dynamics of activity elsewhere in the network (and the resulting learning) end up the same as if external clamping had been applied.

In the blocked condition, we assume AX has been fully learned, so we start with a BX training trial and no AX trials follow. To be consistent, we also start the interleaved condition with a BX training trial; however, in the interleaved condition, both AX and BX may follow after that initial BX trial.

Although the exact values of parameters used in this simulation were slightly different than the values used in the previous simulations (see *Methods* for details), inhibition and oscillations were implemented in the same manner as before.

## Modulation of inhibitory dynamics

For this version of our model, we added an extra manipulation. To model the fact that some brain regions showed overall differentiation and others

**Video 9.** Model of *Schlichting et al., 2015* blocked condition. This video illustrates how integration occurs in the blocked condition of our simulation of *Schlichting et al., 2015*.
https://elifesciences.org/articles/88608/figures#video9

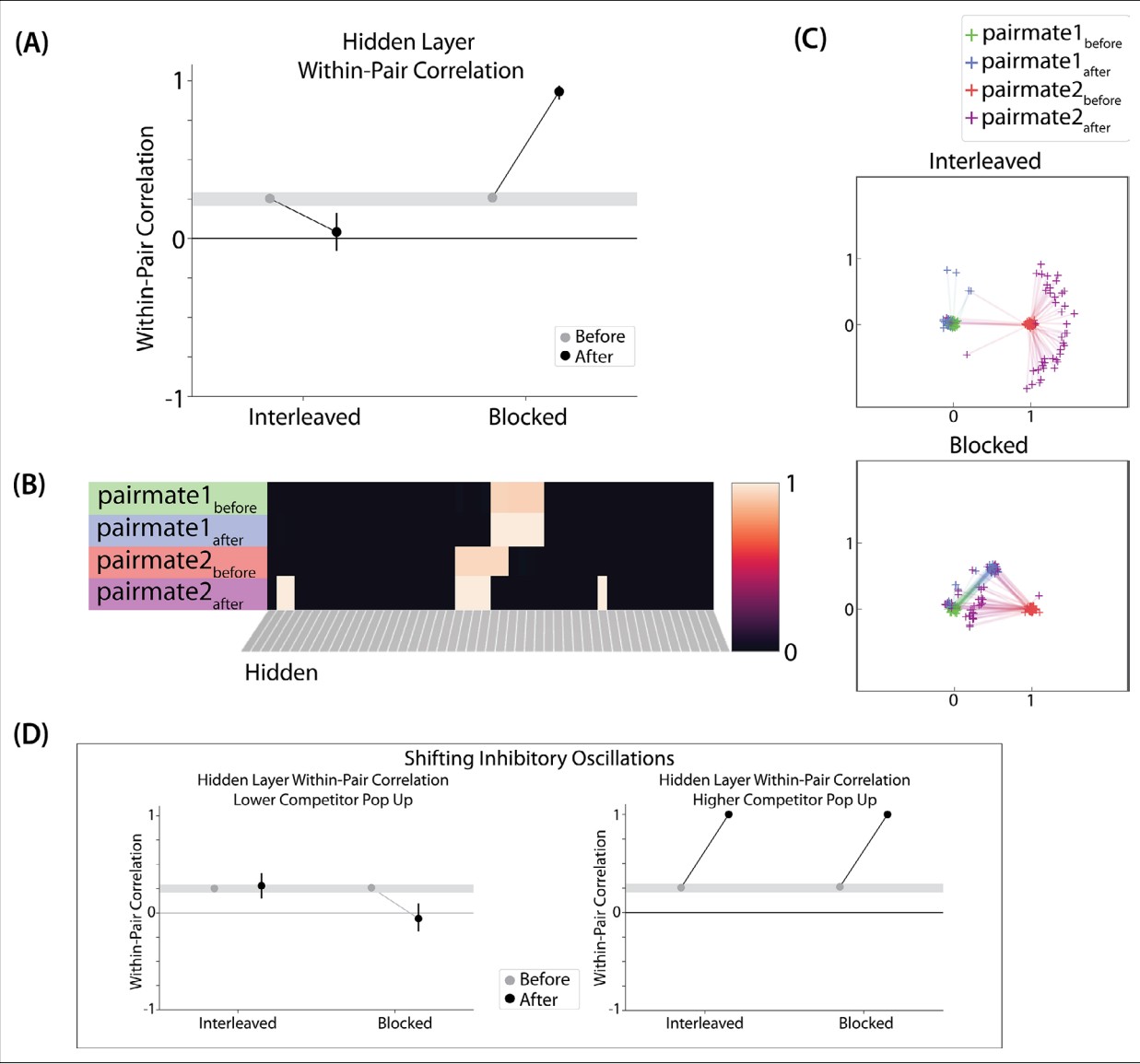

**Figure 10.** Model of *Schlichting et al., 2015* results: in this model, pairmate B is always the stimulus shown first during the competitive learning part of the simulation (after initial AX learning), so we refer to pairmate B as 'pairmate 1' in the figure and to pairmate A as 'pairmate 2'. (**A**) Within-pair correlation between hidden-layer A and B representations is shown before and after learning; here, the oscillation amplitude *Osc* was set to 0.0623. In the interleaved condition, the within-pair correlation is reduced after learning, indicating differentiation. In the blocked condition, the within-pair correlation increases, indicating integration. (**B**) Activity patterns of both pairmates in the hidden layer before and after learning are shown for a sample run in the interleaved condition. Asymmetry in distortion can be seen in how pairmate 2, but not pairmate 1, picks up additional units that did not previously belong to either representation. (**C**) MDS plots for each condition illustrate the pattern of representational change in the hidden layer. In the blocked condition, the pairmates integrate and move toward each other. This integration is mostly symmetric, but on many trials it is asymmetric: Pairmate 2 moves toward pairmate 1 rather than pairmates 1 and 2 meeting in the middle. In the interleaved condition, asymmetric differentiation occurs: Pairmate 2 moves away from pairmate 1. (**D**) To investigate how these results might vary across brain regions with different inhibitory dynamics, we manipulated the inhibitory oscillation amplitude to change the amount of competitor pop-up. No parameters other than the inhibitory oscillation amplitude were changed. When oscillation amplitude is reduced to 0.0525, less competitor pop-up happens, and the blocked, but not interleaved, condition leads to differentiation. When oscillation amplitude is raised to 0.09, more competitor pop-up happens, and both conditions lead to integration. For panels A and D, error bars indicate the 95% confidence intervals around the mean (computed based on 50 model runs).

The online version of this article includes the following figure supplement(s) for figure 10:

**Figure supplement 1.** Results from an alternative parameterization, where reducing the amplitude of inhibitory oscillations leads to differentiation in both the interleaved and blocked conditions.

overall integration, we varied the amplitude of the inhibitory oscillations in the hidden layer. If the amplitude of the sinusoidal function on inhibition is larger (or smaller), then the competitor is allowed to pop up more (or less), affecting the outcome of learning. We first present results from a version of the model where the amplitude is set to an intermediate level (0.0623); we also present results from model variants where we allowed less competitor pop-up (by lowering the oscillation amplitude to 0.0535) and more competitor pop-up (by raising the oscillation amplitude to 0.09), in order to simulate representational change in brain regions that have more or fewer restrictions on competitor activity.

## Results
### Differentiation and integration
Examining the correlation between the hidden-layer representations of A and B, we found that the interleaved condition leads to differentiation (see *Video 8*) whereas the blocked condition (where the AX connection is stronger) leads to integration (see *Video 9*; *Figure 10A*). The reason why the model shows this pattern of results aligns with the *Potential NMPH Explanation* provided earlier: Because the X output unit is more strongly connected to A's hidden-layer representation in the blocked condition, more activity spreads from B via X to A in the blocked condition. This increased competitor activity results in the two representations integrating (as opposed to differentiating).

Note that the key feature driving integration in the blocked condition of this simulation is not the high strength of the connection from X to A *on its own* — rather, it is the *asymmetry* in the pretrained connection strengths from X to A (0.999) and from X to B (0.7). This asymmetry, which is meant to reflect the extensive training on A-X that occurred before the initial presentation of B-X, results in the A-X hidden representation decisively winning the competition during B-X presentation, which then leads to the B input also being linked to this representation (i.e. integration). It is instructive to compare this to the same-face condition from our simulation of *Favila et al., 2016*: In that simulation, the two pairmates are also linked strongly (0.99 initial connection strength) to a shared associate, but in that case the connections are equally strong, so there is more balanced competition — in this case, the competitor representation only comes to mind moderately (instead of displacing the target representation), so the result is differentiation instead of integration.

### Nature of representational hange
The results of this simulation again indicate that differentiation is asymmetric. *Figure 10B* shows a single run in the interleaved condition: Pairmate 1 anchors in place and pairmate 2 picks up units that did not previously belong to either representation. The MDS plot (*Figure 10C*) shows how, in the interleaved condition, pairmate 2 consistently shifts away from pairmate 1. Also, as in our other simulations, when differentiation occurs on a particular model run it tends to give rise to anticorrelated representations (results not shown).

The MDS results from the blocked condition show that integration is mostly symmetric, but there are many runs that show asymmetric integration (just like in the 3/6 condition of our model of *Chanales et al., 2021*). The reason for asymmetric integration is the same here: Sometimes not all of the competitor-competitor coactivity values reach the right side of the U-shaped function, and connections that fall into the 'dip' of the U-shaped function are weakened. When these competitor-competitor connections weaken and the competitor-shared connections strengthen, this imbalance can cause pairmate 2 to flip from its original representation to sharing pairmate 1's original representation.

### Differentiation requires a high learning rate, but integration does not
Just as in our previous simulations, a high learning rate is needed for differentiation. Specifically, lowering *LRate* below its standard value (e.g. to a value of 0.1) eliminated differentiation in the interleaved condition. Lowering the learning rate to 0.1 did not, however, eliminate the integration observed in the blocked condition.

### Adjusting oscillation amplitude modulates representational change
In this model, we additionally adjusted the amplitude of inhibitory oscillations, to simulate different inhibitory dynamics across brain regions (*Figure 10D*). When the competitor was less able to activate

(as a result of smaller inhibitory oscillations), only the blocked condition led to differentiation. When the competitor was allowed to activate more (because of larger inhibitory oscillations), both conditions led to integration. *Figure 10—figure supplement 1* shows results from an alternative parameterization where, in the low-oscillation-amplitude condition, differentiation is observed in both the blocked and interleaved conditions (mirroring results from *Schlichting et al., 2015*, who found differentiation in both conditions in several regions of interest, including parts of the hippocampus and medial prefrontal cortex). Taken together, these simulations provide an 'in principle' account of how differences in levels of inhibition across brain regions can modulate representational change.

It is worth emphasizing that — in this simulation, as in our simulation of *Chanales et al., 2021* — manipulations that raise competitor activity (here, increasing oscillation strength) are associated with a transition from no representational change to differentiation to integration; this is a straightforward consequence of the U shape of the NMPH curve. Notably, this appears to be inconsistent with some recent results from *Molitor et al., 2021*, who used a paradigm similar to *Schlichting et al., 2015* and measured competitor activity with a multivariate fMRI pattern classifier. *Molitor et al., 2021* found that — in a combined DG / CA2,3 hippocampal ROI — lower levels of competitor activity were associated with integration and higher levels of competitor activity were associated with differentiation. Although there are several potential ways that any single finding of this sort could be explained away (see e.g., *Tarder-Stoll et al., 2021*), such a pattern of results could prove troublesome for the model if it turns out to be reliable across studies.

### Take-home lessons

This version of our model shows that the NMPH can account for the results of *Schlichting et al., 2015*, where the learning curriculum (blocked or interleaved) had been shown to affect representational change. Additionally, the model reveals how the inhibitory dynamics of different brain regions can affect these outcomes. As in the other versions of our model, differentiation requires a high learning rate, and — on model runs when it occurs — it is asymmetric and gives rise to anticorrelated representations.

## Discussion

Recent studies have presented a challenge to supervised learning models, showing that linking items to a shared associate can result in differentiation rather than integration. The goal of the current research was to instantiate our unsupervised-learning account of representational change in a neural network model and assess how well it can account for extant data on differentiation and integration. We simulated three studies, each of which modulated the amount of competitor activity in different ways: *Chanales et al., 2021* modulated competitor activity with stimulus similarity; *Favila et al., 2016* modulated competitor activity through the use of shared (vs. unshared) predictive consequences; and *Schlichting et al., 2015* modulated competitor activity by organizing learning in a blocked or interleaved order. The *Schlichting et al., 2015* model also explored the effects of (simulated) regional differences in the strength of local inhibition, and the *Chanales et al., 2021* model additionally explored the effect of varying the learning rate.

Our model provides an existence proof that a network imbued with NMPH learning can explain these findings. Using unsupervised NMPH learning alone, we showed how: (1) Increasing the similarity of color memories leads progressively to a repulsion effect and then an attraction effect; (2) pairing two stimuli with the same associate can lead to differentiation; and (3) learning in an interleaved vs. blocked fashion can lead to differentiation and integration, respectively, and that changing inhibitory dynamics can affect these outcomes. In addition to qualitatively replicating the results from the studies we simulated, our model gives rise to several novel predictions — most notably, that differentiation driven by the NMPH requires a rapid learning rate, and when it occurs for a particular pair of items, it is asymmetric and gives rise to anticorrelated representations.

### Differentiation requires a high learning rate and is sensitive to activity dynamics

Our model predicts that a high learning rate is required for differentiation: As shown in our simulation of *Chanales et al., 2021*, if connections between the unique features of the competitor and the

(formerly) shared features are not sufficiently weakened after one trial, the shared features will be strongly reactivated when the competitor is next presented; in the model, this reactivation of shared features leads to either restrengthening of the previously weakened connections ('undoing' the differentiation) or integration of the target and competitor memories.

We also found that differentiation is highly sensitive to activity dynamics. Anything that affects how strongly the competitor comes to mind can impact the kinds of representational change that are observed. We used a variety of methods to influence how strongly competitors come to mind in the three models, but this is shown particularly clearly in the simulation of *Schlichting et al., 2015* where we modulated the oscillation amplitude. The results from this simulation show that the direction of representational change (i.e. differentiation vs. integration) within a given condition (interleaved or blocked) can switch depending on how much the competitor comes to mind.

These results have implications for where to look for differentiation in the brain. Our finding that differentiation requires a high learning rate suggests that differentiation will be more evident in the hippocampus than in neocortex, insofar as hippocampus is thought to have a higher learning rate than neocortex (*McClelland et al., 1995*). In keeping with this prediction, numerous studies have found differentiation effects in hippocampus but not in neocortical regions involved in sensory processing (e.g. *Chanales et al., 2017*; *Favila et al., 2016*; *Zeithamova et al., 2018*). At the same time, some studies have found differentiation effects in neocortex (e.g. *Schlichting et al., 2015*; *Wammes et al., 2022*). One possible explanation of these neocortical differentiation effects is that they are being 'propped up' by top-down feedback from differentiated representations in the hippocampus. This explanation implies that disruptions of hippocampal processing (e.g. lesions, stimulation) will eliminate these neocortical differentiation effects; we plan to test this prediction in future work.

Additionally, the simulations where we adjusted the oscillation amount (using our model of *Schlichting et al., 2015*) imply that differentiation will be most evident in brain regions where it is relatively hard to activate competitors. Given the U shape of the NMPH learning rule, limiting competitor activity makes it less likely that plasticity will 'cross over' from weakening (and differentiation) to strengthening (and integration). Thus, within the hippocampus, subregions with sparser activity (e.g. dentate gyrus, and to a lesser extent, CA3; *Barnes et al., 1990*; *GoodSmith et al., 2017*; *West et al., 1991*) will be more prone to differentiation. There is strong empirical support for this prediction. For example, *Wammes et al., 2022* manipulated the similarity of stimuli in a statistical learning experiment and found that moderate levels of visual similarity were associated with significant differentiation in the dentate gyrus but not other subregions. Also, numerous studies have found greater differentiation in dentate gyrus / CA3 than in CA1 (e.g. *Dimsdale-Zucker et al., 2018*; *Wanjia et al., 2021*; *Molitor et al., 2021*; *Kim et al., 2017*; but see *Zheng et al., 2021*).

A corollary of the model's prediction that a high learning rate is required for differentiation is that, when differentiation does occur, it should happen abruptly. That is, when the preconditions for differentiation (fast learning, moderate competitor activity) are met, differentiation should be fully evident the next time the competitor is presented. In practice, testing this prediction about abrupt differentiation is challenging because it is not always clear *when* (in the time course of learning) to look for the abrupt change. For the sake of simplicity, we set up our model so the key moment of competition that drives differentiation occurs on the first trial. However, in actual experiments, the timing of this key moment of competition (and the ensuing differentiation) may vary across stimuli. For example, if a participant was inattentive the first time a stimulus was presented, this could delay the onset of competition. If each pair of items has an abrupt change at a different point in the experiment, it could look as though learning is gradual if the time courses of all pairs are averaged during analysis. For instance, *Chanales et al., 2021* found that the percentage of color responses away from the pairmate (indicating repulsion) increased gradually over learning blocks; this could reflect a truly gradual change or it may be an aggregate of multiple abrupt changes occurring at different times.

One way to address this problem would be to model each item's repulsion timecourse using either a gradual function or a step function and see which fits better. Another approach is to use converging behavioral data to identify the moment of differentiation, and then 'time lock' the neural analysis to that moment. *Wanjia et al., 2021* did exactly this: They identified the moment when participants were first able to confidently recall the correct associates that had been linked to a set of pairmates (visually similar scenes, as in *Favila et al., 2016*), and then looked before and after that moment at the neural similarity of the pairmates. As predicted by our model, differentiation was not evident before this

point, and it was clearly evident after this point. Yet another approach would be to use neural activity to identify the trial when a competitor first activates, and then look for differentiation after this point. There are several practical challenges with this approach, however. One challenge is that, although moderate activity is predicted to lead to differentiation, stronger activity is predicted to lead to integration, and there is no a priori way to determine whether a given level of measured activity (e.g. using fMRI decoding) corresponds to the moderate-activity or high-activity portion of the U-shaped curve (*Ritvo et al., 2019*). Another issue is that, for similar pairmates, it can be difficult to tease them apart neurally (e.g. if you are looking at one barn, it can be difficult to sensitively detect additional neural activity of a competing, similar barn). One way to address this challenge is to link pairmates to associates from other categories (e.g. linking one pairmate to a scene and other pairmate to a face), and then look for activity of the associated category (e.g. during viewing of the face-linked pairmate, look for scene activity as a proxy for retrieval of the scene-linked pairmate); for an example of this approach see *Molitor et al., 2021*.

Although the results from *Wanjia et al., 2021* provide strong support for the model's prediction that differentiation will be abrupt, they raise another question: What explains variance across items in *when* this abrupt change takes place? The answer to this question remains to be seen, but one possibility is encoding variability: If we assume that participants stochastically sample (i.e. attend to) the features of the scene pairmates, it is possible that participants might initially fail to sample the features that distinguish the scene pairmates, which can be quite subtle — and if the distinguishing features of the pairmates are not represented in high-level visual regions (i.e. the pairmates are represented in these regions as having the same features), this could delay the onset of differentiation until the point at which the distinguishing features happen (by chance) to be sampled.

## Asymmetry of representational change

Our model predicts that representational change will often be asymmetric. Specifically, the model predicts that differentiation will always be asymmetric, such that the item that first pops up as a competitor is the one that distorts. By contrast, integration in the model is sometimes symmetric and sometimes asymmetric (such that the hidden-layer representation of one item flips to the hidden-layer representation of the other).

As discussed earlier, testing predictions about asymmetric representational change requires a measurement of how much *each individual item* has moved as a result of learning. To make this measurement, it is necessary to collect both pre-learning and post-learning snapshots for each pairmate, although some fMRI studies of differentiation have done this (*Wammes et al., 2022*; *Schlichting et al., 2015*; *Schapiro et al., 2012*; *Kim et al., 2017*; *Molitor et al., 2021*; *Wanjia et al., 2021*), others have not (*Chanales et al., 2017*; *Favila et al., 2016*; *Dimsdale-Zucker et al., 2018*; *Zeithamova et al., 2018*). Importantly, even with a pre-post comparison, there is still the matter of determining which pairmate will anchor and which will distort. Our model predicts that the pairmate that pops up first as the competitor is the one that will distort, but in practice it is not trivial to identify which pairmate will pop up first. For example, we simplified our simulation of *Chanales et al., 2021* by pre-wiring strong memories for both pairmates before presenting one of the pairmates (pairmate 1) to the network. In this situation, pairmate 2 pops up as a competitor when pairmate 1 is studied, so pairmate 2 repels away from pairmate 1 and not vice versa (*Figure 4B*). However, in the actual *Chanales et al., 2021* experiments, encoding strength can vary across trials for numerous reasons (e.g. fluctuating attention); if pairmate 1 is the first to establish a strong representation, it will pop up and distort when pairmate 2 is next studied, but if pairmate 2 is the first to establish a strong representation, it will pop up and distort when pairmate 1 is next studied. Just as encoding variability in *Wanjia et al., 2021* could affect how long it takes to learn to discriminate between pairmates, encoding variability could also make it difficult to predict a priori which item in a given pair will distort in the actual *Chanales et al., 2021* study.

One way to predict the asymmetry is to use a paradigm where we can be more confident which item will be the first to pop up as a competitor. For instance, an AX-BX paradigm using blocked trials like the one used by *Schlichting et al., 2015* may be useful, because all AX trials occur before any BX trials. As such, we can expect that A will pop up as a competitor in response to B (and thus be the memory that distorts) and not vice versa. Studies that rely on the retrieval practice paradigm (or the reverse retrieval-induced forgetting paradigm) to look at representational change (*Hulbert and*

*Norman, 2015*) may also be useful, because one item in each pair (Rp+) undergoes retrieval practice whereas the other (Rp-) item does not (it is either restudied, or not presented at all during the retrieval practice phase). This arrangement makes it more likely that the Rp- item will pop up as a competitor in response to the Rp +item than vice versa. Another approach would be to use neural activity to predict which item would distort. For instance, competitor activity could be measured on a trial-by-trial basis as pairmates are studied to determine which pairmate is the first to pop up as a competitor; this approach would face the same challenges noted above (i.e. the difficulty of teasing apart target-related activity from competitor-related activity, which could possibly be addressed using the approach described by *Molitor et al., 2021* the difficulty of determining whether activity is moderate or strong, which is less easily addressed).

Another way to test the model's prediction about asymmetry, without having to predict which item will move and which item will anchor in place, would be to look for a bimodal distribution of results: In a paradigm like the one used by *Chanales et al., 2021*, one could look for a bimodal distribution of behavioral repulsion values across pairs (i.e. in some pairs, pairmate 1 would repel from its original color value and pairmate 2's color value would stay relatively constant, and in other pairs the opposite would hold). The same logic could be applied to neural analysis: In some pairs, pairmate 1 would show a large change from the pre-learning to the post-learning snapshot, but pairmate 2 would not, and in other pairs the opposite would hold. The viability of this approach (i.e. trying to establish whether the results are best described by a bimodal or a unimodal distribution) will depend on the level of noise in the individual-trial data — if the measurements are too noisy, this will make it difficult to distinguish between the bimodal and unimodal alternatives.

The asymmetry of differentiation also has implications for how to measure behavioral repulsion effects, even when one is not trying to detect asymmetry. In paradigms like *Chanales et al., 2021*, our model predicts that only one item in each pair will repel. This implies that, if you measure differentiation by looking at all items (i.e. both items in a pair), you will be averaging across the pairmate that moved and the pairmate that stayed put, weakening the measured effect. Statistically, it is more powerful to use designs where there is a way of predicting a priori which item will shift and which will anchor, so you can focus on the subset of items that shift without diluting the analysis by including items that anchor.

## Testing the model's prediction about anticorrelation

Even though we operationally define differentiation as a reduction in similarity with learning, the way that it actually shows up on individual model runs is as anticorrelation between pairmates; in the model, the size of the aggregate differentiation effect is determined by the proportion of model runs that show this anticorrelation effect (vs. no change or integration). This implies that, if we could get a clean measurement of the similarity of pairmates in an experiment, we might see a multimodal distribution, with some pairmates showing anticorrelation, and others showing increased correlation (integration) or no change in similarity. This kind of clean readout of the similarity of individual pairs might be difficult to obtain with fMRI; it is more feasible that this could be obtained with electrophysiology. Another challenge with using fMRI to test this prediction is that anticorrelation at the individual-neuron level might not scale up to yield anticorrelation at the level of the BOLD response; also, fMRI pattern similarity values can be strongly affected by preprocessing choices — so a negative pattern similarity value does not necessarily reflect anticorrelation at the individual-neuron level. A final caveat is that, while we predict that differentiation will show up as anticorrelation in the brain region that *gives rise to* the differentiation effect, this might not translate into anticorrelation in areas that are downstream of this region (e.g. if the hippocampus is the source of the differentiation effect, we would expect anticorrelation there, but not necessarily in neocortical regions that receive input from the hippocampus; we revisit this point later in the *Discussion*, when we address limitations and open questions).

## Reconciling the prevalence of differentiation in the model and in the data

A key lesson from our model is that, from a computational perspective, it is challenging to obtain differentiation effects: The region of parameter space that gives rise to differentiation is much smaller than the one that gives rise to integration (for further discussion of this issue, see the section in

*Methods* on Practical advice for getting the model to show differentiation). However, the fact that integration is more prevalent in our simulations across parameter configurations does not mean that integration will be more prevalent than differentiation in real-life circumstances. What really matters in predicting the prevalence of differentiation in real life is how the parameters of the brain map on to parameters of the model: If the parameters of the brain align with regions of model parameter space that give rise to differentiation (even if these regions are small), this would explain why differentiation has been so robustly observed in extant studies. Indeed, this is exactly the case that we sought to make above about the hippocampus — that is its use of especially sparse coding and a high learning rate will give rise to the kinds of neural dynamics that cause differentiation (as opposed to integration). As another example, while it is true that half of the overlap conditions in our simulation of *Chanales et al., 2021* give rise to integration, this does not imply that integration will occur half of the time in the *Chanales et al., 2021* study; it may be that the levels of overlap that are actually observed in the brain in *Chanales et al., 2021* are more in line with the levels of overlap that give rise to differentiation in our model.

## Limitations and open questions

Our model can account for differentiation and integration across several scenarios. We think this provides important computational support for the NMPH explanation of representational change, and useful predictions for future work. Nonetheless, there are several ways our model can be extended in the future.

We intentionally kept our model very simple in order to create a 'sandbox' to explore the dynamics at play during learning. We used no initial study task for our model, and we instead opted for pre-wired connections in order to precisely manipulate the initial hidden-layer overlap between the pairmates and the relative strengths of their representations.

Rather than skipping initial stimulus learning and focusing on the first moment that competition happens, future work can investigate the trials leading up to this moment in the course of learning. For instance, our model of *Schlichting et al., 2015* instantiates interleaved vs. blocked learning by changing the initial weights connecting the hidden layer to the unit representing the shared associate in the output layer. Future models could instead show stimuli in a blocked or interleaved fashion with learning turned on, which should lead to the kind of weights we pre-wired into the model.

The model could also be expanded to be more biologically realistic. For example, instantiating the NMPH in a more detailed model of the hippocampus (e.g. *Schapiro et al., 2017*; *Norman and O'Reilly, 2003*) would allow us to simulate the contributions of different hippocampal subfields to representational change (e.g. contrasting the monosynaptic and trisynaptic pathways; *Schapiro et al., 2017*). Extending this further, building a model that incorporates both hippocampal and neocortical regions would make it possible to explore how NMPH learning in the hippocampus can support neocortical learning. Although our simulations suggest that neocortex on its own cannot enact differentiation via NMPH learning (due to neocortex's slow learning rate, which causes it to 'relapse' and reabsorb formerly shared units), we hypothesize that — in the longer term — hippocampus may be able to act as a teacher to support long-lasting differentiation within neocortex. Specifically, we hypothesize that — once differentiated hippocampal representations have been formed — these representations can provide a source of top-down activity to the unique features of the corresponding neocortical representations (e.g. unique features of the similar barns in *Favila et al., 2016*). This hippocampally-mediated attention to unique features in neocortex may help neocortex (gradually) learn to represent and leverage subtle differences between pairmates. Importantly, while hippocampus can boost the representation of unique features in neocortex, we expect that neocortex will continue to represent shared perceptual features (e.g. in *Favila et al., 2016*, the fact that both pairmates are photos of barns). For this reason, in paradigms like the one used by *Favila et al., 2016*, the predicted effect of hippocampal differentiation on neocortical representations will be a reduction in pattern similarity (due to upregulation in the representation of unique pairmate features) but neocortex should not cross over into anticorrelation in these paradigms (due to its continued representation of shared perceptual features). Indeed, this is exactly the pattern that *Wanjia et al., 2021* observed in their study, which used similar stimuli to those used in *Favila et al., 2016*. We will explore these ideas in future simulations.

**Table 1.** Parameters for layer inhibitory and activity dynamics.

*kWTA Point* = a value between 0 and 1, which indicates how far toward the $k + 1^{th}$ unit to place the current inhibitory level (the higher *kWTA Point*, the lower the inhibition value). *Target Diff* = the threshold for determining whether units after the $k^{th}$ unit should be allowed to be active (see text). *Osc* = the amplitude of the oscillation function that multiplies the overall inhibition level of the layer. Note that for the model of *Schlichting et al., 2015*, we tested three different hidden-layer oscillation amounts: 0.0623, 0.0525 and 0.09. *XX1 Gain* = the multiplier on the S-shaped activity function, where lower values means that activity will be more graded. *Clamp Gain* multiplies the external input, modifying how strongly it contributes to the activity of the layer.

| Model | Layer | K | K Max | kWTA Point | Target Diff | Osc | XX1 Gain | Clamp Gain |
|---|---|---|---|---|---|---|---|---|
| Chanales | Output | 6 | 15 | 0.95 | 0.05 | 0.115 | 30 | — |
| Chanales | Hidden | 6 | 10 | 0.75 | 0.03 | 0.11 | 100 | — |
| Chanales | Category | 1 | ∞ | 0.75 | 0 | 0 | 100 | 2 |
| Chanales | Item | 1 | ∞ | 0.95 | 0.2 | 0.22 | 100 | 0.3 |
| Favila | Output | 1 | ∞ | 0.75 | 0.03 | 0.07 | 100 | — |
| Favila | Hidden | 6 | 10 | 0.8 | 0.02 | 0.067 | 100 | — |
| Favila | Category | 1 | ∞ | 0.75 | 0 | 0 | 100 | 2 |
| Favila | Item | 1 | ∞ | 0.95 | 0.2 | 0.2 | 100 | 0.3 |
| Schlichting | Output | 1 | ∞ | 0.75 | 0.03 | 0.03 | 100 | — |
| Schlichting | Hidden | 6 | 10 | 0.8 | 0.02 | 0.0623 | 100 | — |
| Schlichting | Category | 1 | ∞ | 0.75 | 0 | 0 | 100 | 2 |
| Schlichting | Item | 1 | ∞ | 0.95 | 0.2 | 0.156 | 100 | 0.3 |

Our keep-things-simple approach led us to focus on NMPH learning in the simulations presented here; in future work, we plan to explore how NMPH interacts with error-driven (supervised) learning. As discussed in the *Introduction*, supervised learning is not sufficient to explain effects modeled here. However, there is widespread agreement that error-driven learning happens in the brain (although its mechanisms are still under active investigation; *Richards et al., 2019*; *Payeur et al., 2021*; *Lillicrap et al., 2020*; *Whittington and Bogacz, 2019*), and it may interact with and/or complement the NMPH principles explored here in interesting ways. In the computer vision literature, supplementing supervised learning algorithms with unsupervised learning algorithms (some of which are very similar to the NMPH, in that they pull together the representations of strongly similar items and push apart the embeddings of moderately similar items; *Zhuang et al., 2019*) can boost model performance in some circumstances (*Zhuang et al., 2021*). However, mixing unsupervised with supervised learning in models of neocortex has a substantial drawback: Once an unsupervised learning algorithm has decided that two stimuli are similar enough to pull together their representations, it can be very difficult to pull them apart again, even if this needs to be done to solve a task; this point was initially demonstrated in the domain of auditory learning by *McClelland et al., 1999* and *Mccandliss et al., 2002*. In this situation, hippocampal differentiation (which is *not* part of extant computer vision models) may play a key role in helping neocortex avoid this kind of 'premature collapse' of representations. In future work, we also plan to explore the influence of other biologically inspired learning principles, including the principle of metaplasticity (whereby the transition points between strengthening and weakening are adjusted as a function of activity), which has previously been shown to play an important role in NMPH-style learning (*Bear, 2003*).

Another important future direction is to apply the model to a wider range of learning phenomena involving representational change — for example, acquired equivalence, which (like some of the studies modeled here) involves linking distinct stimuli to a shared associate (see, e.g. *Honey and Hall, 1989*; *Shohamy and Wagner, 2008*; *Myers et al., 2003*; *Meeter et al., 2009*; *de Araujo Sanchez and Zeithamova, 2023*). It is possible that some of these phenomena might be better explained by supervised learning, or a mixture of unsupervised and supervised learning, than by unsupervised learning alone.

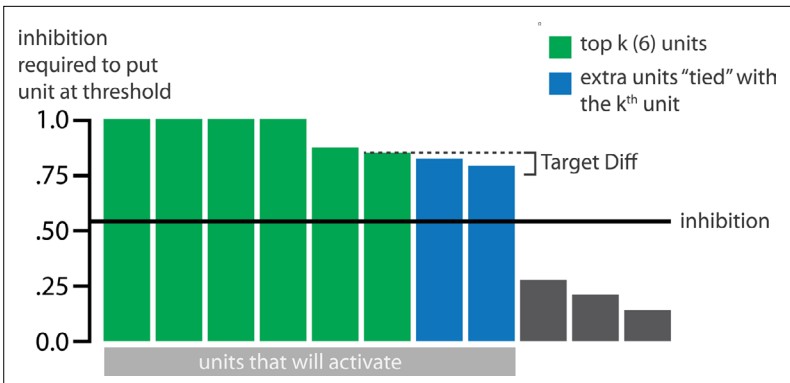

**Figure 11.** Schematic of adjusted KWTA algorithm. Units are ranked according to the amount of inhibition that would be needed to put the unit at threshold of activity. This is proportional to excitation: The more excitation the unit receives, the more inhibition is needed to cancel out the excitation and put the unit at threshold. In the classic KWTA algorithm, inhibition is set such that only the $k$ highest ranked units activate. We added a *Target Diff* parameter to potentially allow more units to activate, if units are 'tied' with the $k^{th}$ unit. If a unit below the $k^{th}$ unit in the rank ordering of units is within *Target Diff* of the $k^{th}$ unit, then it is considered to be 'tied' with the $k^{th}$ unit, and it is allowed to activate. In this example, the 6th, 7th, and 8th unit are tied in the ranking, because the difference is less than *Target Diff*. Consequently, inhibition is set such that 8 units activate.

Yet another direction to explore is how cognitive control and attention modulate representational change. The simulations described in this paper provide an existence proof of how — if the conditions are right (e.g. moderate competitor activity, suitably high learning rate) — differentiation can occur automatically, without participants having to deliberately focus their attention on discriminative features of the pairmates. However, it is surely the case that attention can modulate these learning effects (*Amer and Davachi, 2023*; see the Practical advice section in the *Methods* for a brief discussion of this point).

## Summary

The model presented in this paper provides a concrete computational instantiation of the NMPH account of representational change set forth in *Ritvo et al., 2019*. By modulating competitor activity in different ways (by varying stimulus similarity, presence of shared associates, learning curriculum, and inhibitory dynamics) and tracking how this affects learning, our model serves several purposes: It provides an existence proof of how the NMPH can explain otherwise-puzzling findings regarding representational change (e.g. why linking to a shared associate can promote differentiation); it provides principled new explanations of certain patterns in the literature (e.g. why differentiation is more frequently observed in the hippocampus than in neocortex); and it makes novel, testable predictions (e.g. regarding the asymmetry of differentiation). Although more work remains to be done to explore the consequences of this U-shaped learning function for representational change, we hope our model can be useful in framing future modeling and empirical research on learning.

## Methods
### Model architecture

We built our computational model using the Leabra algorithm (*O'Reilly, 2012*) within the Emergent neural network simulation software (*Aisa et al., 2008*), which is written in the programming language Go. We constructed the model as described in *Basic Network Properties*, and adapted the parameters to model the three individual studies.

### Approach to parameterization and data fitting

The overall goal of this modeling work is to account for key empirical regularities regarding differentiation and integration and to establish boundary conditions on these regularities. As such, the modeling work described below focuses more on qualitative fits to general properties of the data space than on quantitative fits to results from specific studies. Automatic parameter optimization is not feasible for

**Table 2.** Projection parameters: *Wt Range* = range of the uniform distribution used to initialize the random weights between each projection (range does not include the maximally strong pre-wired connections described in the text, which were set to 0.99 unless stated otherwise).
*Wt Scale* = the scaling of the projection, operationalized as an absolute multiplier on the weights in the projection.

| Model | Projection | Wt Range | Wt Scale (Forwards / Backwards) |
|---|---|---|---|
| Chanales | Hidden ↔ Hidden | 0.45–0.55 | 1.8/1.8 |
| Chanales | Hidden↔ Output | 0.01–0.03 | 3.0/2.0 |
| Chanales | Category↔Hidden | 0.01–0.03 | 0.2/0.2 |
| Chanales | Item↔Hidden | 0.45–0.55 | 0.2/0.2 |
| Chanales | Output↔Output | 0.01–0.03 | 1.0/1.0 |
| Favila | Hidden↔Hidden | 0.45–0.55 | 1.8/1.8 |
| Favila | Hidden↔Output | 0.01–0.03 | 1.2/1.7 |
| Favila | Category↔Hidden | 0.01–0.03 | 0.1/0.1 |
| Favila | Item↔Hidden | 0.45–0.55 | 0.3/0.2 |
| Schlichting | Hidden↔Hidden | 0.45–0.55 | 1.9/1.9 |
| Schlichting | Hidden↔Output | 0.01–0.03 | 1.2/1.7 |
| Schlichting | Category↔Hidden | 0.01–0.03 | 0.2/0.1 |
| Schlichting | Item↔Hidden | 0.45–0.55 | 0.3/0.2 |

this kind of model, given the large number of model parameters and the highly interactive, nonlinear nature of competitive dynamics in the model; consequently, model fitting was done by hand.

These complex interactions between parameters also make it infeasible to list 'critical parameter ranges' for generating particular model outcomes. Our experience in working with the model has been that activation dynamics are what matter most for learning, and that disparate parameter sets can give rise to the same activation dynamics and — through this — the same learning effects; likewise, similar parameter sets can give rise to different activation dynamics and different learning outcomes. Consequently, in this paper we have focused on characterizing the dynamics that give rise to different learning effects (and how they can be affected by local parameter perturbations, e.g. relating to learning rate and oscillation size), rather than the — impossible, we believe — task of enumerating the full set of parameter configurations that give rise to a particular result.

While the core model architecture and dynamics were the same for all three simulations, the specific parameters that we used differed in small ways across the three simulations. The need to make these parameter adjustments is a consequence of the small-scale nature of the model. In a more realistic, large-scale model with more stored knowledge, changing the structure of what is being learned in the to-be-simulated paradigm would not meaningfully affect the overall knowledge state of the model (adding two memories if you already have thousands is not a big deal) and thus would not require parameter changes. However, if you are starting with a 'blank brain', the exact way that you structure the to-be-learned memories matters; for example, having a topographic color projection to represent repulsion effects in our simulation of *Chanales et al., 2021* can have large effects on network dynamics that require downstream adjustments in other parameters.

To generate the results shown in the figures here, we ran the model 50 times, each time starting with a different random seed. Results plots show the mean level of performance across model runs and the 95% confidence interval around the mean.

## Activity and inhibitory dynamics

Emergent simplifies the discrete on-off firing of individual neurons into a rate code approximation where a unit's activity can range from 0 to 1; this rate code reflects the unit's current level of excitatory and inhibitory inputs. A given unit M's excitatory input is a function of the activity of all other units N connected to it, weighted by the strengths of the connections between the N units and M. The

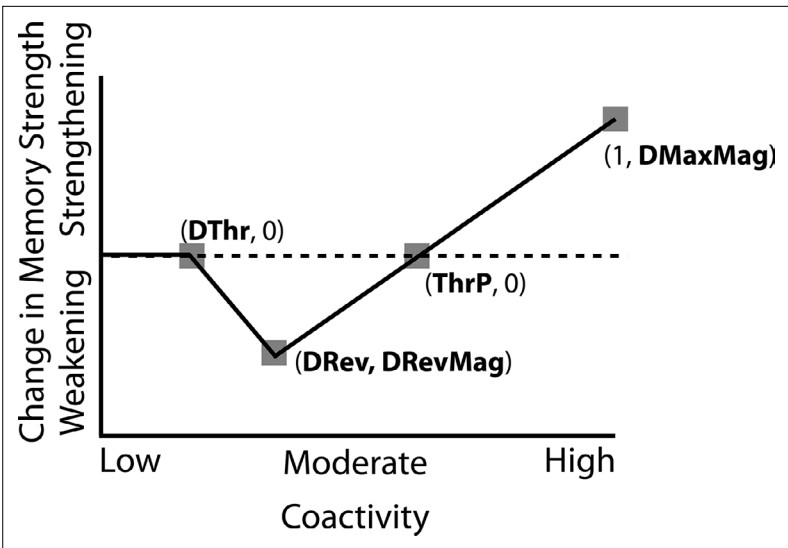

**Figure 12.** U-shaped learning function. *DThr*, *DRev*, and *ThrP* are X-axis coordinates. *DRevMag* and *DMaxMag* are Y-axis coordinates, and indicate the amount of peak weakening or strengthening.

function relating excitatory input to activity (after accounting for inhibition; see below and *O'Reilly, 2012*) is S-shaped and bounded between 0–1, where the steepness of the function is modulated by a gain factor. Lowering the gain creates a more graded response, which enables moderate pop up of competitor units. The gain for each layer is written in *Table 1* (*XX1 Gain*), along with the other activity and inhibitory parameters for these models. When an external input is clamped to a layer, the external input is multiplied by a *Clamp Gain* factor, which modulates how strongly it should contribute to activity compared to other excitatory inputs that the unit is receiving (from elsewhere in the network).

A unit's activity is also modulated by the inhibitory dynamics in the layer. We implemented inhibition using a variant of the *k-winners take all* (kWTA) algorithm (*O'Reilly and Munakata, 2000*). This algorithm imposes a 'set point'-like behavior on inhibition, ensuring that at most $k$ units are allowed to activate in a layer. In the standard version of the kWTA algorithm, the units in a layer are ranked in terms of the amount of inhibition that would be needed to put that unit at the threshold of activating, given its current excitatory input. Then, inhibition is set such that only the $k$ highest-ranked units are able to reach the activity threshold.

In our simulations, we adjusted the kWTA algorithm to give the model the ability to activate more than $k$ units in certain circumstances (*Figure 11*). This flexibility is helpful for integration: Allowing more than $k$ units to be active makes it possible for the model to incorporate units from both pair-mates into the new, integrated representation. To provide this flexibility, we adjusted the kWTA algorithm to allow units that are tied in the ranking (within a threshold amount reflected by the *Target Diff* parameter) to activate as well. We also included a cap on the total number of units allowed to activate even with the tie (reflected by the *K_Max* parameter). So if $k = 6$ and *K_Max* = 10, the top 6 units will be allowed to activate, as well as optionally any units beyond the top 6 that are tied (within some threshold specified by *Target Diff*), but no more than 10 units can be active.

Note that, in addition to allowing more than $k$ units to be active, this modification also helps to solve an issue with the classic implementation of kWTA that emerges when the inhibition needed to put the $k^{th}$ unit at threshold is very close to the value needed for the $k + 1^{th}$ unit. In the classic implementation of kWTA, this situation results in the $k^{th}$ unit's activity being very close to zero (since it is only slightly above threshold); effectively, fewer than $k$ units end up being active. This issue would occur, for example, with the configuration of values shown in *Figure 11*; with classic kWTA, the 5th and 6th units would be just above threshold and thus barely active. This issue can make it difficult for the competitor to pick up new units in conditions that lead to differentiation (because the new units often receive very similar levels of excitation). With the modified kWTA algorithm, the model is free to find a 'break point' beyond the $k^{th}$ unit that allows the new units to be more robustly active.

**Table 3.** Learning parameters: All parameters are defined in *Figure 12*.

All bidirectional connections used the same parameters for the U-shaped learning function (e.g. the parameters for item-to-hidden matched the parameters for hidden-to-item).

| Model | Projection | DThr | DRev | DRevMag | ThrP | DMaxMag |
|---|---|---|---|---|---|---|
| Chanales | Hidden↔Hidden | 0.15 | 0.24 | −4.5 | 0.4 | 0.1 |
| Chanales | Hidden↔Output | 0.1 | 0.44 | −10 | 0.6 | 1.5 |
| Chanales | Category↔Hidden | 0.2 | 0.3 | −0.1 | 0.46 | 0.06 |
| Chanales | Item↔Hidden | 0.2 | 0.3 | −2.5 | 0.46 | 0.3 |
| Chanales | Output↔Output | 0.53 | 0.6 | −0.3 | 0.68 | 0.3 |
| Favila | Hidden↔Hidden | 0.11 | 0.23 | −1.5 | 0.4 | 0.1 |
| Favila | Hidden↔Output | 0.11 | 0.23 | −0.01 | 0.4 | 0.5 |
| Favila | Category↔Hidden | 0.2 | 0.3 | −0.1 | 0.46 | 0.06 |
| Favila | Item↔Hidden | 0.215 | 0.4 | −2.5 | 0.6 | 0.3 |
| Schlichting | Hidden↔Hidden | 0.11 | 0.23 | −1.5 | 0.4 | 1 |
| Schlichting | Hidden↔Output | 0.11 | 0.23 | −0.01 | 0.4 | 0.5 |
| Schlichting | Category↔Hidden | 0.2 | 0.3 | −0.1 | 0.46 | 0.06 |
| Schlichting | Item↔Hidden | 0.11 | 0.23 | −1.5 | 0.4 | 1 |

Lastly, inhibition is modulated by oscillations; these inhibitory oscillations play a key role in regulating the amount of competitor activity. The layer's inhibition is initially constant (determined by the kWTA calculation described above), and then at cycle 125 is varied according to a sine wave (with amplitude set by the parameter *Osc*) until the end of the trial at cycle 200. Concretely, inhibition is sinusoidally raised above baseline from cycles 125–163, and then lowered below baseline from cycles 164–200. The raising of inhibition does not have much impact on the network activity (because the external clamping is strong enough to offset the raising of inhibition), but lowering the inhibition allows the competitors to activate.

## Projections between layers

The weight of any given connection between two units could range from 0 to 1. The two input layers are connected to the hidden layer, which is in turn connected to the output layer. The hidden layer also has recurrent connections. Generally the output layer does not have recurrent connections, except for our model of *Chanales et al., 2021*, where we included fixed-strength recurrent output-to-output connections to simulate topographically organized color representations. All projections were bidirectional: Activity was allowed to flow forwards and backwards through the network. Parameters for each bidirectional projection were identical except for the *Wt Scale*, which indicated the overall strength of the projection from one layer to another.

All layers that were connected were connected fully — that is, all units in one layer were connected to all units in the other layer. Most of these connections were random and low in magnitude, except for a set of pre-wired connections that were stronger, as described in *Basic Network Properties*. Any pre-wired connections that differed between versions of the model are included in the sections of this paper on the set up of each model. All pre-wired, non-random connections were set to have a strength of 0.99 unless otherwise indicated. The parameters for the random connections are shown in *Table 2*.

The random low-level connections constitute one of two sources of variance in the model, the other being the trial order for most simulations (described below in *Stimulus Presentation*).

## A note on prewiring representations

In our model, our practice of 'prewiring' memory representations for the A and B pairmates serves two functions. In some cases, it is meant to stand in for actual training (as in the blocked / interleaved manipulation; the connections supporting the AX association are prewired to be stronger in the blocked condition than in the interleaved condition). However, the other, more fundamental role

of prewiring is to ensure that the A and B input patterns evoke sparse distributed representations in the hidden layer (i.e. where some units are strongly active but most other units are inactive). In the real brain, this happens automatically because the weight landscape has been extensively sculpted by both experience and evolution. For example, in the real hippocampus, when the second pairmate is presented for the first time, it will evoke a sparse distributed representation in the CA3 subfield (potentially overlapping with the first pairmate's CA3 representation) even before any learning of the second pairmate has occurred, due to the strong, sparse mossy fiber projections that connect the dentate gyrus to CA3 (*McNaughton and Morris, 1987*). As discussed above, we hypothesize that this initial, partial overlap between the second pairmate's representation and the first pairmate's representation can lead to pop-up of the unique features of the first pairmate's representation, triggering learning that leads to differentiation or integration. In our small-scale model, we are effectively starting with a 'blank brain'; in the absence of prewiring, the A and B inputs would activate overly diffuse representations that do not support these kinds of competitive dynamics. As such, prewiring in our model is necessary for its proper functioning. The presence of prewired A and B representations should therefore not be interpreted as reflecting a particular training history (except in the blocked / interleaved case above); rather, these prewired representations constitute the minimum step we would take to ensure well-defined competitive dynamics in our small-scale model.

The fact that connection strengths serve this dual function — sometimes reflecting effects of training (as in our simulation of *Schlichting et al., 2015*) and in other cases reflecting necessary prewiring — complicates the interpretation of these strength values in the model. Our view is that this is a necessary limitation of our simplified modeling approach — one that can eventually be surmounted through the use of more biologically-detailed architectures (see the *Limitations and Open Questions* section in the *Discussion*).

## Learning

We used only unsupervised NMPH learning in this model since we wanted to test whether the NMPH was sufficient to produce the representational changes observed in these experiments. Each of the projections in the model could be modified through learning, except for the topographic output-to-output projection in the model of *Chanales et al., 2021*.

After each training trial, connection weights were adjusted based on the coactivity of the units on that trial. For each pair of connected units, the coactivity is computed as the product of both units' medium-term running average activity (defined below). The U-shaped function is defined by five parameters (*Figure 12*), and is applied to this coactivity value to calculate the magnitude and direction of the weight change.

Intuitively, medium-term running average activity is an integrated measure of the unit's activity across the trial and is hierarchically computed from the unit's super-short- and short-term average activity. The equations for computing medium-term running average activity were taken from the Leabra algorithm described in *O'Reilly, 2018*; for additional conceptual motivation for approach, see *O'Reilly, 2012*. First, the activity of each unit is integrated to yield a super-short-term average of unit activity:

$$\mu_t^{\text{super-short}} = \mu_{t-1}^{\text{super-short}} + \alpha^{\text{super-short}} \left( x_t - \mu_{t-1}^{\text{super-short}} \right)$$

where $x_t$ is the activity of the unit at time $t$, $\mu_{t-1}^{\text{super-short}}$ is the super-short-term average activity at time $t-1$, and $\alpha^{\text{super-short}}$ is the super-short-term time scale constant. Next, super-short-term average activity is integrated to yield short-term average activity:

$$\mu_t^{\text{short}} = \mu_{t-1}^{\text{short}} + \alpha^{\text{short}} \left( \mu_t^{\text{super-short}} - \mu_{t-1}^{\text{short}} \right)$$

where $\mu_{t-1}^{\text{short}}$ is the short-term activity at time $t-1$ and $\alpha^{\text{short}}$ is the short-term time scale constant. Lastly, short-term average activity is integrated to yield the raw medium-term average activity:

$$\mu_t^{\text{raw medium}} = \mu_{t-1}^{\text{raw medium}} + \alpha^{\text{raw medium}} \left( \mu_t^{\text{short}} - \mu_{t-1}^{\text{raw medium}} \right)$$

where $\mu_{t-1}^{\text{raw medium}}$ is the raw medium-term activity at time $t-1$ and $\alpha^{\text{raw medium}}$ is the medium-term time scale constant. The final medium-term average activity is computed as a linear combination of raw medium-term average activity and the short-term average activity:

$$\mu_t^{\text{medium}} = \alpha \mu_t^{\text{raw medium}} + \beta \mu_t^{\text{short}}$$

The coactivity $\kappa$ between the receiver unit and the sender unit is the product between the final medium-long term average activity between those two units:

$$\kappa = \mu_{\text{receiver}}^{\text{medium}} \mu_{\text{sender}}^{\text{medium}}$$

This coactivity is fed into the U-shaped function, and then the resulting value is multiplied by a scalar learning rate parameter, *LRate*, to obtain the final weight change value. *LRate* was set to 1 for all of our simulations except for the simulations where we explicitly manipulated learning rate.

The values of the time scale parameters used to compute running averages were unchanged from the defaults used in *O'Reilly, 2018*: $\alpha^{\text{super-short}} = 0.5, \alpha^{\text{short}} = 0.5, \alpha^{\text{raw medium}} = 0.1$. The linear combination parameters were set to $\alpha = 0.9, \beta = 0.1$ (whereas in the original Leabra models they were set to $\alpha = 0.1, \beta = 0.9$).

The five parameters that define the U-shaped function were set separately for each projection in each model, such that — when moderate competitor pop-up occurred — the competitor-shared connections would be severed. All reciprocal connections (e.g. output-to-hidden and hidden-to-output) had identical U-shaped functions. Once the U-shaped function was set for a connection, those were the parameters used for all runs of the model, in all conditions. The parameters for each learning function can be found in *Table 3*.

Evidence for the U-shaped plasticity function used here (where low activation leads to no change, moderate activation leads to weakening, and higher levels of activation lead to strengthening) was previously reviewed in *Ritvo et al., 2019*. In brief, there are three lines of work that support the U shape: First, multiple neurophysiological studies have found that moderate postsynaptic depolarization leads to synaptic weakening and higher levels of depolarization lead to synaptic strengthening (e.g. *Artola et al., 1990*; *Hansel, 1996*). Second, human neuroscience studies have used pattern classifiers, applied to fMRI and EEG data, to measure memory activation, and have related this measure to subsequent memory accessibility; several studies using this approach have found that low levels of activation lead to no change in memory strength, moderate levels of activation lead to impaired subsequent memory, and higher levels of activation lead to increased subsequent memory (e.g. *Newman and Norman, 2010*; *Detre et al., 2013*; *Kim et al., 2014*; for related findings, see *Lewis-Peacock and Norman, 2014*; *Wang et al., 2019*). Third, a recent human fMRI study by *Wammes et al., 2022* manipulated memory activation by varying the visual similarity of pairmates and observed a U-shaped function relating visual similarity to representational change in the hippocampus, whereby low levels of pairmate similarity were associated with no change, moderate levels of similarity were associated with differentiation, and the differentiation effect went away at higher levels of similarity.

## Stimulus presentation

Each model run included only two stimuli — pairmates A and B. This modeling choice reflects our assumption that the competitive dynamics of interest occur between pairmates and not across pairs (given that pairs are typically constructed to be dissimilar to other pairs and thus are not confusable with them). The middle unit in the category layer was active for both pairmates (reflecting input features shared between them) and distinct units in the item layer were active for pairmates A and B. These external inputs were the same for all models. Every epoch consisted of a single presentation of A and B, in a random order, except in our model of *Schlichting et al., 2015* where B was always shown first (and, in the blocked condition, no A trials followed).

We ran a test epoch before the first training epoch to attain a 'baseline' measure of the representations of A and B, and we ran another test epoch after each subsequent training epoch. The only difference between test and training epochs was that (1) test epochs included no oscillations, so we could get a pure guess for the state of the representation, and (2) connection weights were not adjusted at the end of test epochs (i.e. learning was turned off). We ran the model for 20 train/test epochs, and the final pattern of unit activity for each test epoch was recorded and used for the analyses.

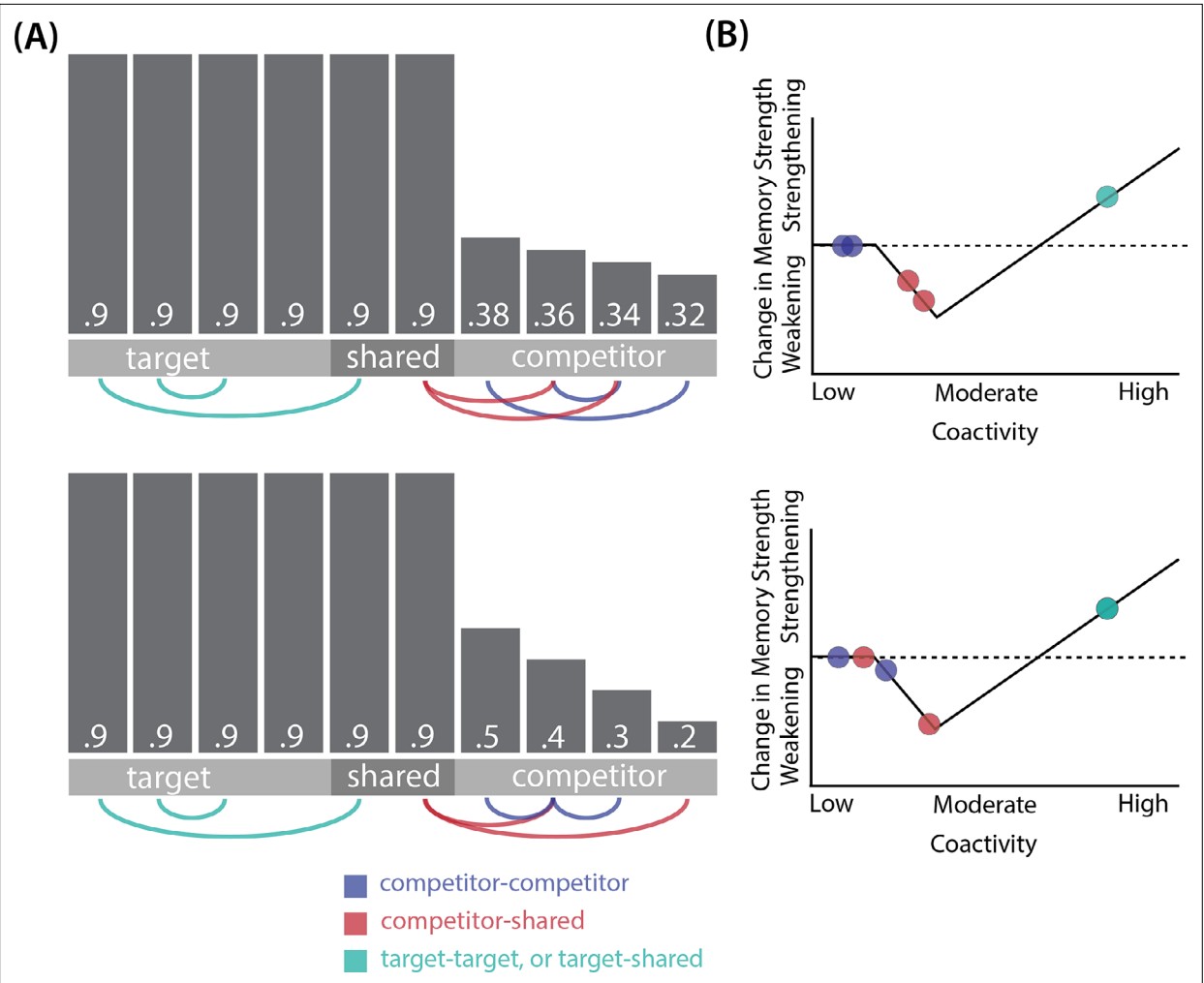

**Figure 13.** How subtle changes to activity can affect representational change: (**A**) Two sample activity patterns in the hidden layer during the first trial are shown. Units are labelled as belonging to either the target, competitor, or both (shared), and vertical bars indicate activity level (along with activity values). A moderate amount of pop-up of the competitor occurs, which should lead to differentiation if the competitor-shared connections are appropriately weakened. Although the two patterns are very similar, the bottom pattern has slightly more variable activity, which could arise if the level of random noise in the strengths of the hidden-hidden connections is higher. (**B**) A U-shaped function for each activity pattern is shown, with the coactivity values for a subset of the hidden-unit pairings (those highlighted by the arcs in part A) plotted along the X axis. In the top example, all competitor-competitor coactivities are lower than all competitor-shared coactivities, which are in turn lower than all target-shared and target-target coactivities. This means that it is possible to preserve all of the competitor-competitor connections while severing all of the competitor-shared connections. However, in the bottom example, there is some interlacing of the competitor-competitor coactivities and competitor-shared coactivities; this scenario makes it impossible to fully preserve all competitor-competitor connections while severing all competitor-shared connections. With the U-shaped function shown here, the pattern of activity in the bottom example will result in sparing of some of the competitor-shared connections, making it less likely that differentiation will occur.

## Practical advice for getting the model to show differentiation

As discussed above, differentiation is highly sensitive to activity dynamics. To obtain differentiation, pairmate 2 (the competitor) must activate moderately on trial 1, and the following criteria must be met on trial 2:

1. On trial 2 (when pairmate 2 is presented), hidden-layer units that were initially shared between the pairmates cannot be reactivated.
2. On trial 2, the unique hidden-layer units associated with pairmate 2 must be reactivated.

This seems straightforward, but meeting these conditions can be difficult. Sometimes the parameter adjustments needed to satisfy #1 make #2 harder to satisfy, and vice versa. Most of the failures to differentiate that we observed happened if these two conditions were not fully met.

## On trial 2, pairmate 1 hidden units cannot be reactivated

Even if some weakening of competitor-shared connections has occurred on trial 1, it is possible for pairmate 1 units to activate on trial 2. If this happens, any initial differentiation effects are undone. This happens if the excitation for pairmate 1 units remains too high, and there are several potential reasons why this might occur.

We have discussed one potential cause already, when learning rate is low and the competitor-shared connections are only mildly weakened on trial 1. Additionally, if only some (but not all) competitor-shared connections weaken, the intact connections may reactivate pairmate 1 on trial 2, undoing any initial differentiation. For example, if variation in the random connection weights is too high, not all competitor units will pop up the same amount. This can lead to a situation where some of the competitor-competitor coactivity values are greater than some of the competitor-shared coactivity values, making it impossible to preserve within-competitor links while simultaneously severing links between competitor units and shared units (*Figure 13*).

Another relevant parameter is the oscillation amplitude *Osc*, which can shift the placement of competitor-shared coactivity on the U-shaped function. If the oscillation is not set to the appropriate amplitude, the resulting level of competitor unit activity may not cause appropriate severing of competitor-shared connections. This can be seen in the results of our model of *Schlichting et al., 2015*: When the oscillation amount is adjusted, this can have strong effects on the representational changes that occur (*Figure 10A and D*).

Yet another factor to consider is how much excitation the hidden representation of pairmate 1 receives on trial 2. There are some projections that will always send excitation to pairmate 1 on trial 2, so some amount of excitation of the hidden pairmate 1 units is unavoidable. One projection for which this is true (for all models) is the category-to-hidden projection, since the category unit is part of both the A and B input patterns. Some of the models also have other units that maintain a connection to both A and B; for instance, in the shared-face condition of our model of *Favila et al., 2016* and also our model of *Schlichting et al., 2015*, the shared output unit is connected to both items' hidden representations. In the model, lowering the strengths of these projections (i.e. category-to-hidden, output-to-hidden) by adjusting the corresponding *Wt Scale* parameters will reduce the amount of excitation received by pairmate 1. In an actual experiment, participants could accomplish this by attending less to shared features. As noted earlier, selective attention is not *necessary* for differentiation to occur in our model, but this account suggests that attention could modulate the strength of differentiation effects by modulating the relative influence of shared vs. unique features (for discussion of potential mechanisms of these effects, see *Amer and Davachi, 2023*).

Learning that occurs in the category-hidden and output-hidden projections can also affect the relative amount of excitation received by the hidden representations of pairmate 1 and pairmate 2. On trial 1, if shared input/output features are strongly activated and unique hidden features of pairmate 2 are moderately activated, then NMPH learning can result in weakening of the connections between the shared input/output features and the unique hidden features of pairmate 2. If this occurs, then, on trial 2, shared input/output features will selectively send excitation to pairmate 1, but not pairmate 2, which further increases the odds that pairmate 1's hidden representation will be activated (thwarting the differentiation effect). In our simulations, we were able to avoid this problem by using a small *DRevMag* value for projections from shared input/output features, which effectively reduces the amount of weakening that occurs for these projections.

## On trial 2, unique pairmate 2 hidden units must be reactivated

If connections within the pairmate 2 representation are weakened too much on trial 1 (when it pops up as a competitor), then, when pairmate 2 is presented as the target on trial 2, its hidden representation will not be accessible (i.e. the model will fail to reactivate the unique parts of this hidden representation). When this happens, the hidden representation of pairmate 1 typically ends up being activated instead. This leads to the asymmetric form of integration, where the input units for pairmate 2 end up being linked to the hidden representation of pairmate 1.

Parameter adjustments that lower the amount of pop-up are helpful in addressing this issue; lowering pop-up shifts the competitor-competitor coactivities to the left on the U-shaped function, making it less likely that these connections will be weakened. For instance, decreasing the oscillation amount may be needed to make sure competitor-competitor connections are spared in trial 1. However, if the oscillation amount is decreased too much, it may also cause competitor-shared connections to be spared, as discussed in point #1 above. Successful differentiation requires 'threading the needle' such that competitor-competitor connections are (relatively) spared but connections between competitor and shared units are severed.

## Acknowledgements

This work was supported by NIH R01 MH069456 awarded to KAN and NBT-B.

## Additional information

### Funding

| Funder | Grant reference number | Author |
| --- | --- | --- |
| National Institutes of Health | MH069456 | Nicholas B Turk-Browne Kenneth A Norman |

The funders had no role in study design, data collection and interpretation, or the decision to submit the work for publication.

### Author contributions

Victoria JH Ritvo, Conceptualization, Software, Formal analysis, Investigation, Visualization, Methodology, Writing – original draft, Writing – review and editing; Alex Nguyen, Software, Formal analysis, Validation, Investigation, Visualization, Methodology, Writing – original draft, Writing – review and editing; Nicholas B Turk-Browne, Conceptualization, Supervision, Funding acquisition, Writing – review and editing; Kenneth A Norman, Conceptualization, Resources, Supervision, Funding acquisition, Methodology, Writing – original draft, Writing – review and editing

### Author ORCIDs

Nicholas B Turk-Browne ⓘ https://orcid.org/0000-0001-7519-3001
Kenneth A Norman ⓘ https://orcid.org/0000-0002-5887-9682

Reviewer #1 (Public review): https://doi.org/10.7554/eLife.88608.3.sa1
Reviewer #2 (Public review): https://doi.org/10.7554/eLife.88608.3.sa2
Reviewer #3 (Public review): https://doi.org/10.7554/eLife.88608.3.sa3
Author response https://doi.org/10.7554/eLife.88608.3.sa4

## Additional files

### Supplementary files
• MDAR checklist

### Data availability

The current manuscript is a computational study, so no data have been generated for this manuscript. Simulation code can be downloaded from https://github.com/PrincetonCompMemLab/neurodiff_simulations (copy archived at *Nguyen, 2024*).

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
