## [Editor Report · eLife assessment]

This paper presents **important** computational modeling work that provides a mechanistic account for how memory representations become integrated or differentiated (i.e., having distinct neural representations despite being similar in content). The authors provide **convincing** evidence that simple unsupervised learning in a neural network model, which critically weakens connections of units that are moderately activated by multiple memories, can account for three empirical findings of differentiation in the literature. The paper also provides insightful discussion on the factors contributing to differentiation as opposed to integration, and makes new predictions for future empirical work.

---

## [Referee Report · Reviewer #1 (Public review)]

Ritvo and colleagues present an impressive suite of simulations that can account for three findings of differentiation in the literature. This is important because differentiation-in which items that have some features in common, or share a common associate are less similar to one another than are unrelated items-is difficult to explain with classic supervised learning models, as these predict the opposite (i.e., an increase in similarity). A few of their key findings are that differentiation requires a high learning rate and low inhibitory oscillations, and is virtually always asymmetric in nature.

This paper was very clear and thoughtful-an absolute joy to read. The model is simple and elegant, and powerful enough to re-create many aspects of existing differentiation findings. The interrogation of the model and presentation of the findings were both extremely thorough. The potential for this model to be used to drive future work is huge.

The authors have been very responsive to my previous reviews and I have no further concerns and identify no major weaknesses.

---

## [Referee Report · Reviewer #2 (Public review)]

Summary:

This paper addresses an important computational problem in learning and memory. Why do related memory representations sometimes become more similar to each other (integration) and sometimes more distinct (differentiation)? Classic supervised learning models predict that shared associations should cause memories to integrate, but these models have recently been challenged by empirical data showing that shared associations can sometimes cause differentiation. The authors have previously proposed that unsupervised learning may account for these unintuitive data. Here, they follow up on this idea by actually implementing an unsupervised neural network model that updates the connections between memories based on the amount of coactivity between them. The authors use their modeling framework to simulate three recent empirical studies, showing that their model captures aspects of these findings that are hard to account for with supervised learning.

Overall, this is a strong and clearly described work that is likely to have a positive impact on computational and empirical work in learning and memory. While the authors have written about some of the ideas discussed in this paper previously, a fully implemented and openly available model is a clear advance that will benefit the field. It is not easy to translate a high-level description of a learning rule into a model that actually runs and behaves as expected. The fact that the authors have made all their code available makes it likely that other researchers will extend the model in numerous interesting ways, many of which the authors have discussed and highlighted in their paper.

Strengths:

The authors succeed in demonstrating that unsupervised learning with a simple u-shaped rule can produce results that are qualitatively in line with the empirical reports. In each of the three models, the authors manipulate stimulus similarity (following Chanales et al.), shared vs distinct associations (following Favila et al.), or learning strength (a stand-in for blocked versus interleaved learning schedule; following Schlichting et al.). In all cases, with hand-tuning of additional parameters, the authors are able to produce model representations that fit the empirical results, but that can't easily be accounted for by supervised learning. Demonstrating these effects isn't trivial and a formal modeling framework for doing so is a valuable contribution. Overall, the work is very thorough. The authors investigate many different aspects of the learning dynamics (learning rate, oscillation strength, hidden layer overlap etc) in these models and produce several key insights. Of particular value are their demonstrations that when differentiation occurs, it occurs very quickly and asymmetrically and results in anti-correlated representations, as well as the distinction between symmetric and asymmetric integration in their model. The authors thoroughly acknowledge the relative difficulty of producing differentiation in their models relative to integration, and are now more clear about why they don't necessarily view this as mismatch with the empirical data. The authors are also more clear about the complicated activation dynamics in their model and why critical ranges for some parameters can't be given -- the number of interacting parameters mean that there are many combinations that could produce the critical activation dynamics and thus the same result. Despite this complexity, the paper is very clearly written; the authors do a good job of both formally describing their model as well as giving readers a high level sense of how many of their critical model components work.

Weaknesses:

Though the u-shaped learning rule is essential to this framework, the paper doesn't do any formal investigation of this learning rule or comparison with other learning rules. The authors do have a strong theoretical interest in this rule as well as experimental precedent for testing this rule, which they now thoroughly discuss in the paper. Still, a stronger argument in support of the non monotonic plasticity hypothesis could have been made by comparing this learning rule to alternatives. Additionally, the authors' choice of strongly prewiring associations makes it difficult to think about how their model maps onto experimental contexts where associations are only weakly learned. However, the authors thoroughly acknowledge why this was necessary and discuss this limitation in the paper.

---

## [Referee Report · Reviewer #3 (Public review)]

This paper proposes a computational account for the phenomenon of pattern differentiation (i.e., items having distinct neural representations when they are similar). The computational model relies on a learning mechanism of the nonmonotonic plasticity hypothesis, fast learning rate and inhibitory oscillations. In the revised paper, the authors justified the initialization of the model, added empirical evidence supporting the use of two turning points in the NMPH function and provided details of the learning mechanisms of the model. The relatively simple architecture of the model makes its dynamics accessible to the human mind. Furthermore, using similar model parameters, this model produces simulated data consistent with empirical data of pattern differentiation. The authors also provide insightful discussion on the factors contributing to differentiation as opposed to integration.

---

## [Author Response]

The following is the authors’ response to the original reviews.

**Public Reviews:**

**Reviewer #1 (Public Review):**
Ritvo and colleagues present an impressive suite of simulations that can account for three findings of differentiation in the literature. This is important because differentiation-in which items that have some features in common, or share a common associate are less similar to one another than are unrelated items-is difficult to explain with classic supervised learning models, as these predict the opposite (i.e., an increase in similarity). A few of their key findings are that differentiation requires a high learning rate and low inhibitory oscillations, and is virtually always asymmetric in nature.This paper was very clear and thoughtful-an absolute joy to read. The model is simple and elegant, and powerful enough to re-create many aspects of existing differentiation findings. The interrogation of the model and presentation of the findings were both extremely thorough. The potential for this model to be used to drive future work is huge. I have only a few comments for the authors, all of which are relatively minor.(1) I was struck by the fact that the "zone" of repulsion is quite narrow, compared with the zone of attraction. This was most notable in the modeling of Chanales et al. (i.e., just one of the six similarity levels yielded differentiation). Do the authors think this is a generalizable property of the model or phenomenon, or something idiosyncratic to do with the current investigation? It seems curious that differentiation findings (e.g., in hippocampus) are so robustly observed in the literature despite the mechanism seemingly requiring a very particular set of circumstances. I wonder if the authors could speculate on this point a bit-for example, might the differentiation zone be wider when competitor "pop up" is low (i.e., low inhibitory oscillations), which could help explain why it's often observed in hippocampus? This seems related a bit to the question about what makes something "moderately" active, or how could one ensure "moderate" activation if they were, say, designing an experiment looking at differentiation.

We thank the reviewer for this comment. In the previous version of the manuscript, in the section entitled “Differentiation Requires a High Learning Rate and Is Sensitive to Activation Dynamics”, we discussed some reasons why differentiation may be more likely to be found in the hippocampus – namely, the high learning rate of the hippocampus and the sparsity of hippocampal activation patterns (pp. 27-28):

“These results have implications for where to look for differentiation in the brain. Our finding that differentiation requires a high learning rate suggests that differentiation will be more evident in the hippocampus than in neocortex, insofar as hippocampus is thought to have a higher learning rate than neocortex (McClelland et al., 1995). In keeping with this prediction, numerous studies have found differentiation effects in hippocampus but not in neocortical regions involved in sensory processing (e.g., Chanales et al., 2017; Favila et al., 2016; Zeithamova et al., 2018). At the same time, some studies have found differentiation effects in neocortex (e.g., Schlichting et al., 2015; Wammes et al., 2022). One possible explanation of these neocortical differentiation effects is that they are being ``propped up’’ by top-down feedback from differentiated representations in the hippocampus. This explanation implies that disruptions of hippocampal processing (e.g., lesions, stimulation) will eliminate these neocortical differentiation effects; we plan to test this prediction in future work.

Additionally, the simulations where we adjusted the oscillation amount (using our model of Schlichting et al., 2015) imply that differentiation will be most evident in brain regions where it is relatively hard to activate competitors. Given the U shape of the NMPH learning rule, limiting competitor activity makes it less likely that plasticity will ``cross over'' from weakening (and differentiation) to strengthening (and integration). Thus, within the hippocampus, subregions with sparser activity (e.g., dentate gyrus, and to a lesser extent, CA3; Barnes et al., 1990, GoodSmith et al., 2017; West et al., 1991) will be more prone to differentiation. There is strong empirical support for this prediction. For example, Wammes et al. (2022) manipulated the similarity of stimuli in a statistical learning experiment and found that moderate levels of visual similarity were associated with significant differentiation in the dentate gyrus but not other subregions. Also, numerous studies have found greater differentiation in dentate gyrus / CA3 than in CA1 (e.g., Dimsdale-Zucker et al., 2018; Wanjia et al., 2021; Molitor et al., 2021; Kim et al., 2017; but see Zheng et al., 2021).”

In the revised draft we have supplemented this discussion with a new section entitled “Reconciling the Prevalence of Differentiation in the Model and in the Data” (pp. 30-31):

“A key lesson from our model is that, from a computational perspective, it is challenging to obtain differentiation effects: The region of parameter space that gives rise to differentiation is much smaller than the one that gives rise to integration (for further discussion of this issue, see the section in Methods on Practical Advice for Getting the Model to Show Differentiation). However, the fact that integration is more prevalent in our simulations across parameter configurations does not mean that integration will be more prevalent than differentiation in real-life circumstances. What really matters in predicting the prevalence of differentiation in real life is how the parameters of the brain map on to parameters of the model: If the parameters of the brain align with regions of model parameter space that give rise to differentiation (even if these regions are small), this would explain why differentiation has been so robustly observed in extant studies. Indeed, this is exactly the case that we sought to make above about the hippocampus – i.e., that its use of especially sparse coding and a high learning rate will give rise to the kinds of neural dynamics that cause differentiation (as opposed to integration). As another example, while it is true that half of the overlap conditions in our simulation of Chanales et al. (2021) give rise to integration, this does not imply that integration will occur half of the time in the Chanales et al. (2021) study; it may be that the levels of overlap that are actually observed in the brain in Chanales et al. (2021) are more in line with the levels of overlap that give rise to differentiation in our model.”

(2) With real fMRI data we know that the actual correlation value doesn't matter all that much, and anti-correlations can be induced by things like preprocessing decisions. I am wondering if the important criterion in the model is that the correlations (e.g., as shown in Figure 6) go down from pre to post, versus that they are negative in sign during the post learning period. I would think that here, similar to in neural data, a decrease in correlation would be sufficient to conclude differentiation, but would love the authors' thoughts on that.

We thank the reviewer for bringing this up. In the paper, we define differentiation as the moving apart of representations – so we agree with the reviewer that it would be appropriate to conclude that differentiation is taking place when correlations go down from pre to post.

In addition to the definitional question (“what counts as differentiation”), one can also ask the *mechanistic* question of what is happening in the model at the (simulated) neuronal level in conditions where differentiation (i.e., an average decrease in similarity from pre to post) occurs. Here, the model’s answer is clear: When the similarity of two pairmates decreases, it is because the pairmates have acquired anticorrelated representations at the (simulated) neuronal level. When similarity decreases on average from pre to post, but the average “post” similarity value is not negative, this is because there is a mix of outcomes across runs of the model (due to variance in the initial, random model weights and also variance in the order in which items are presented across training epochs) – some runs lead to differentiation (manifested as anticorrelated pairmate representations) whereas others lead to no change or integration. The average pre-to-post change depends on the relative frequencies with which these different outcomes occur.

We have made several edits to the paper to clarify this point.

We added a new section under “Results” in our simulation of Chanales et al. (2021) entitled, “Pairs of Items that Differentiate Show Anticorrelated Representations” (p. 15):

“Figure 6B also highlights that, for learning rates where robust differentiation effects occur in aggregate (i.e., there is a reduction in mean pattern similarity, averaging across model runs), these aggregate effects involve a bimodal distribution across model runs: For some model runs, learning processes give rise to anticorrelated representations, and for other model runs the model shows integration; this variance across model runs is attributable to random differences in the initial weight configuration of the model. The aggregate differentiation effect is therefore a function of the proportion of model runs showing differentiation (here, anticorrelation) and the proportion of model runs showing integration. The fact that differentiation shows up as anticorrelation in the model's hidden layer relates to the learning effects discussed earlier:

Unique competitor units are sheared away from (formerly) shared units, so the competitor ends up not having any overlap with the target representation (i.e., the level of overlap is less than you would expect due to chance, which mathematically translates into anticorrelation). We return to this point and discuss how to test for anticorrelation in the *Discussion* section.”

We added new text to the “Take-Home Lessons” section in the Chanales et al. (2021) simulation (p. 17):

“In particular, the simulations expose some important boundary conditions for when representational change can occur according to the NMPH (e.g., that differentiation depends on a large learning rate, but integration does not), and the simulations provide a more nuanced account of exactly how representations change (e.g., that differentiation driven by the NMPH is always asymmetric, whereas integration is sometimes asymmetric and sometimes symmetric; and that, when differentiation occurs on a particular model run, it tends to give rise to anticorrelated representations in the model's hidden layer).”

We added new text to the “Nature of Representational Change” section in the Favila et al. (2016) simulation (p. 21):

“Figure 8 - Supplement 1 also indicates that, as in our simulation of Chanales et al. (2021), individual model runs where differentiation occurs show anticorrelation between the pairmate representations, and gradations in the aggregate level of differentiation that is observed across conditions reflect differences in the proportion of trials showing this anticorrelation effect.”

We added new text to the “Take-Home Lessons” section in the Favila et al. (2016) simulation (p.21):

“As in our simulation of Chanales et al., 2021, we found that the NMPH-mediated differentiation was asymmetric, manifested as anticorrelation between pairmate representations on individual model runs, and required a high learning rate, leading to abrupt representational change.”

We added new text to the “Nature of Representational Change” section in the Schlichting et al. (2015) simulation (p. 26):

“Also, as in our other simulations, when differentiation occurs on a particular model run it tends to give rise to anticorrelated representations (results not shown).”

We added new text to the “Take-Home Lessons” section in the Schlichting et al. (2015) simulation (pp. 26-27):

“As in the other versions of our model, differentiation requires a high learning rate, and – on model runs when it occurs – it is asymmetric and gives rise to anticorrelated representations.”

We added new text at the start of the Discussion (p. 27):

“In addition to qualitatively replicating the results from the studies we simulated, our model gives rise to several novel predictions – most notably, that differentiation driven by the NMPH requires a rapid learning rate and, when it occurs for a particular pair of items, it is asymmetric and gives rise to anticorrelated representations.”

We also added a new section in the Discussion entitled “Testing the Model's Prediction about Anticorrelation”, which (among other things) highlights the reviewer’s point that fMRI pattern similarity values can be affected by preprocessing choices (p. 30):

“Even though we operationally define differentiation as a reduction in similarity with learning, the way that it actually shows up on individual model runs is as anticorrelation between pairmates; in the model, the size of the aggregate differentiation effect is determined by the proportion of model runs that show this anticorrelation effect (vs. no change or integration). This implies that, if we could get a clean measurement of the similarity of pairmates in an experiment, we might see a multimodal distribution, with some pairmates showing anticorrelation, and others showing increased correlation (integration) or no change in similarity. This kind of clean readout of the similarity of individual pairs might be difficult to obtain with fMRI; it is more feasible that this could be obtained with electrophysiology. Another challenge with using fMRI to test this prediction is that anticorrelation at the individual-neuron level might not scale up to yield anticorrelation at the level of the BOLD response; also, fMRI pattern similarity values can be strongly affected by preprocessing choices – so a negative pattern similarity value does not necessarily reflect anticorrelation at the individual-neuron level. A final caveat is that, while we predict that differentiation will show up as anticorrelation in the brain region that *gives rise to* the differentiation effect, this might not translate into anticorrelation in areas that are downstream of this region (e.g., if the hippocampus is the source of the differentiation effect, we would expect anticorrelation there, but not necessarily in neocortical regions that receive input from the hippocampus; we revisit this point later in the discussion, when we address limitations and open questions).”

We added new text in the Discussion, under “Limitations and Open Questions” (p. 31):

“Importantly, while hippocampus can boost the representation of unique features in neocortex, we expect that neocortex will continue to represent shared perceptual features (e.g., in Favila et al., 2016, the fact that both pairmates are photos of barns). For this reason, in paradigms like the one used by Favila et al. (2016), the predicted effect of hippocampal differentiation on neocortical representations will be a reduction in pattern similarity (due to upregulation in the representation of unique pairmate features) but neocortex should not cross over into anticorrelation in these paradigms (due to its continued representation of shared perceptual features). Indeed, this is exactly the pattern that Wanjia et al. (2021) observed in their study, which used similar stimuli to those used in Favila et al. (2016).”

Lastly, we updated the Abstract (p. 1)

“What determines when neural representations of memories move together (integrate) or apart (differentiate)? Classic supervised learning models posit that, when two stimuli predict similar outcomes, their representations should integrate. However, these models have recently been challenged by studies showing that pairing two stimuli with a shared associate can sometimes cause differentiation, depending on the parameters of the study and the brain region being examined. Here, we provide a purely unsupervised neural network model that can explain these and other related findings. The model can exhibit integration or differentiation depending on the amount of activity allowed to spread to competitors – inactive memories are not modified, connections to moderately active competitors are weakened (leading to differentiation), and connections to highly active competitors are strengthened (leading to integration). The model also makes several novel predictions – most importantly, that when differentiation occurs as a result of this unsupervised learning mechanism, it will be rapid and asymmetric, and it will give rise to anticorrelated representations in the region of the brain that is the source of the differentiation. Overall, these modeling results provide a computational explanation for a diverse set of seemingly contradictory empirical findings in the memory literature, as well as new insights into the dynamics at play during learning.”

(3) For the modeling of the Favila et al. study, the authors state that a high learning rate is required for differentiation of the same-face pairs. This made me wonder what happens in the low learning rate simulations. Does integration occur?

For the same-face condition of the Favila simulation, lowering learning rate does not result in an overall integration effect:

**Author response image 1. sa4fig1:** 

In other cases, we do see integration emerge at lower learning rates – e.g., in the Schlichting interleaved condition we see a small integration effect emerge for a learning rate value of 0.3:

**Author response image 2. sa4fig2:** 

Our view is that, while integration can emerge at low learning rates, it is not a reliable property of the model – in some cases, there is a “window” of learning rates where there is enough learning to drive integration but not enough to drive differentiation, and in other cases there is not. Given this lack of reliability across simulations, we would prefer not to discuss this in the paper.

This paradigm has a lot of overlap with acquired equivalence, and so I am thinking about whether these are the sorts of small differences (e.g., same-category scenes and perhaps a high learning rate) that bias the system to differentiate instead of integrate.

We agree that it would be very interesting to use the model to explore acquired equivalence and related phenomena, but we think it is out of scope of the current paper. We have added some text to the Discussion under “Limitations and Open Questions” (p. 32):

“Another important future direction is to apply the model to a wider range of learning phenomena involving representational change – for example, acquired equivalence, which (like some of the studies modeled here) involves linking distinct stimuli to a shared associate (see, e.g., Honey and Hall, 1989; Shohamy and Wagner, 2008; Myers et al., 2003; Meeter et al., 2009; de Araujo Sanchez and Zeithamova, 2023). It is possible that some of these phenomena might be better explained by supervised learning, or a mixture of unsupervised and supervised learning, than by unsupervised learning alone.”

(4) For the simulations of the Schlichting et al. study, the A and B appear to have overlap in the hidden layer based on Figure 9, despite there being no similarity between the A and B items in the study (in contrast to Favila et al., in which they were similar kinds of scenes, and Chanales et al., in which they were similar colors). Why was this decision made? Do the effects depend on some overlap within the hidden layer? (This doesn't seem to be explained in the paper that I saw though, so maybe just it's a visualization error?)

Overlap in the pretrained hidden representations of A and B is not strictly necessary for these effects – it would be possible to reconfigure other parameters to get high levels of competition even if there were no overlap (e.g., by upregulating the strengths of connections from shared input features). Having said that, it is definitely true that overlap between the pretrained hidden representations boosts competition, and we think it is justified to posit this in the Schlichting simulation. We have now added an explanation for this in the paper (p. 23):

“New text in Schlichting, “Knowledge Built into the Network”

Matching the previous two simulations, we pretrained the weights so the hidden representations of the stimuli initially had 2/6 units in common. Even though the A and B stimuli used in the actual experiment did not have obvious feature overlap (they were randomly selected novel objects), it is important to note that the hidden layer is not simply a representation of the sensory features of the A and B stimuli; the hidden layer also receives input from the output layer, which represents the shared associate of A and B (X). We think that the presence of this shared associate justifies our use of initially-overlapping hidden representations.”

(5) It seems as though there were no conditions under which the simulations produced differentiation in both the blocked and intermixed conditions, which Schlichting et al. observed in many regions (as the present authors note). Is there any way to reconcile this difference?

We thank the reviewer for bringing this up. If we set the connection strength between X (in the output layer) and A (in the hidden layer) in the blocked condition to .9 instead of .999 (keeping this connection strength at .8 for the interleaved condition) and we set *Osc* to .0615, we observe differentiation in both conditions.

Rather than replacing the original results in the paper, which would entail re-making the associated videos, etc., we have added a supplementary figure (Figure 10 - Supplement 1), which is included on p. 46.

We also added the following to the Results section of the Schlichting simulation in the main text (p. 26):

“Figure 10 - Supplement 1 shows results from an alternative parameterization where, in the low-oscillation-amplitude condition, differentiation is observed in both the blocked and interleaved conditions (mirroring results from Schlichting et al., 2015, who found differentiation in both conditions in several regions of interest, including parts of the hippocampus and medial prefrontal cortex).”

(6) A general question about differentiation/repulsion and how it affects the hidden layer representation in the model: Is it the case that the representation is actually "shifted" or repelled over so it is no longer overlapping? Or do the shared connections just get pruned, such that the item that has more "movement" in representational space is represented by fewer units on the hidden layer (i.e., is reduced in size)? I think, if I understand correctly, that whether it gets shifted vs. reduce would depend on the strength of connections along the hidden layer, which would in turn depend on whether it represents some meaningful continuous dimension (like color) or not. But, if the connections within the hidden layer are relatively weak and it is the case that representations become reduced in size, would there be any anticipated consequences of this (e.g., cognitively/behaviorally)?

The representations are shifted – this is discussed in the Chanales results section:

“Because the activity ``set point'' for the hidden layer (determined by the kWTA algorithm) involves having 6 units active, and the unique parts of the competitor only take up 4 of these 6 units, this leaves room for activity to spread to additional units. Given the topographic projections in the output layer, the model is biased to ``pick up'' units that are adjacent in color space to the currently active units; because activity cannot flow easily from the competitor back to the target (as a result of the aforementioned severing of connections), it flows instead {\em away} from the target, activating two additional units, which are then incorporated into the competitor representation. This sequence of events (first a severing of the shared units, then a shift away from the target) completes the process of neural differentiation, and is what leads to the behavioral repulsion effect in color recall (because the center-of-mass of the color representation has now shifted away from the target).”

**Reviewer #2 (Public Review):**
This paper addresses an important computational problem in learning and memory. Why do related memory representations sometimes become more similar to each other (integration) and sometimes more distinct (differentiation)? Classic supervised learning models predict that shared associations should cause memories to integrate, but these models have recently been challenged by empirical data showing that shared associations can sometimes cause differentiation. The authors have previously proposed that unsupervised learning may account for these unintuitive data. Here, they follow up on this idea by actually implementing an unsupervised neural network model that updates the connections between memories based on the amount of coactivity between them. The goal of the authors' paper is to assess whether such a model can account for recent empirical data at odds with supervised learning accounts. For each empirical finding they wish to explain, the authors built a neural network model with a very simple architecture (two inputs layers, one hidden layer, and one output layer) and with prewired stimulus representations and associations. On each trial, a stimulus is presented to the model, and inhibitory oscillations allow competing memories to pop up. Pre-specified u-shaped learning rules are used to update the weights in the model, such that low coactivity leaves model connections unchanged, moderate coactivity weakens connections, and high coactivity strengthens connections. In each of the three models, the authors manipulate stimulus similarity (following Chanales et al), shared vs distinct associations (following Favila et al), or learning strength (a stand in for blocked versus interleaved learning schedule; following Schlichting et al) and evaluate how the model representations evolve over trials.As a proof of principle, the authors succeed in demonstrating that unsupervised learning with asimple u-shaped rule can produce qualitative results in line with the empirical reports. For instance, they show that pairing two stimuli with a common associate (as in Favila et al) can lead to *differentiation* of the model representations. Demonstrating these effects isn't trivial and a formal modeling framework for doing so is a valuable contribution. Overall, the authors do a good job of both formally describing their model and giving readers a high level sense of how their critical model components work, though there are some places where the robustness of the model to different parameter choices is unclear. In some cases, the authors are very clear about this (e.g. the fast learning rate required to observe differentiation)**.** However, in other instances, the paper would be strengthened by a clearer reporting of the critical parameter ranges.

We thank the reviewer for raising this point. The interdependence of parameters in our model makes it infeasible to identify critical parameter ranges. We have added a paragraph to the “Approach to Parameterization and Data Fitting” section in the Methods to address this point (p. 33):

“The overall goal of this modeling work is to account for key empirical regularities regarding differentiation and integration and to establish boundary conditions on these regularities. As such, the modeling work described below focuses more on qualitative fits to general properties of the data space than on quantitative fits to results from specific studies. Automatic parameter optimization is not feasible for this kind of model, given the large number of model parameters and the highly interactive, nonlinear nature of competitive dynamics in the model; consequently, model fitting was done by hand.

These complex interactions between parameters also make it infeasible to list “critical parameter ranges” for generating particular model outcomes. Our experience in working with the model has been that activation dynamics are what matter most for learning, and that disparate parameter sets can give rise to the same activation dynamics and -- through this -- the same learning effects; likewise, similar parameter sets can give rise to different activation dynamics and different learning outcomes. Consequently, in this paper we have focused on characterizing the dynamics that give rise to different learning effects (and how they can be affected by local parameter perturbations, e.g., relating to learning rate and oscillation size), rather than the – impossible, we believe – task of enumerating the full set of parameter configurations that give rise to a particular result.”

For instance, it's clear from the manipulation of oscillation strength in the model of Schlichting et al that this parameter can dramatically change the direction of the results. The authors do report the oscillation strength parameter values that they used in the other two models, but it is not clear how sensitive these models are to small changes in this value.

In some cases, the effects of oscillation strength are relatively smooth. For example, in the Favila simulation, increasing the oscillation amplitude *Osc* effectively recapitulates the U-shaped curve (i.e., higher levels of *Osc* lead to more competitor activation, which initially leads to weakening / differentiation but then gives way to strengthening / integration), as is shown for the Favila Different Face condition in this plot:

**Author response image 3. sa4fig3:** 

In the Chanales 2/6 overlap condition, the effects of varying *Osc* are more nonlinear:

**Author response image 4. sa4fig4:** 

We think this is attributable to the increased “all-or-none” recurrent dynamics in this simulation (due to the recurrent projections within the output layer), which make it more difficult to evoke moderate (vs. high) levels of activation. This difficulty in reliably obtaining graded activation dynamics is likely a consequence of the small-scale (“toy”) nature of the model and the simple inhibitory mechanisms employed here, as opposed to being a generalizable property of the brain – presumably, the actual brain employs more nuanced and effective means of controlling activation. Furthermore, we don’t think that the high prevalence of integration in the model’s parameter space necessarily translates into a prediction that integration should be more prevalent overall – see the new “Reconciling the Prevalence of Differentiation in the Model and in the Data” section described in response to one of the reviewer’s other points below. Due to the paper already being quite long, we have opted not to include the above plots / discussion in the paper.

Similarly, it's not clear whether the 2/6 hidden layer overlap (only explicitly manipulated in the model of Chanales et al) is required for the other two models to work.

When we were parameterizing the model, we opted to keep the 2/6 level of overlap for all of the simulations and we adjusted other parameters to fit the data; in part, this was because overlap can only be adjusted in discrete jumps, whereas other influential parameters in the model can be adjusted in a more graded, real-valued way. Our use of 2/6 overlap (as opposed to, say, 1/6 or 3/6 overlap) for the Favila and Schlichting models was done out of convenience, and should not be interpreted as a strong statement that this particular level of overlap is necessary for obtaining differentiation; we could easily get the model to show differentiation given other overlap levels by adjusting other parameters.

Finally, though the u-shaped learning rule is essential to this framework, the paper does little formal investigation of this learning rule. It seems obvious that allowing the u-shape to collapse too much toward a horizontal line would reduce the model's ability to account for empirical results, but there may be other more interesting features of the learning rule parameterization that are essential for the model to function properly.

Given that the paper is already quite long, we have opted not to include further exploration of the parameters of the U-shaped learning rule in the paper. However, for the reviewer’s information, we report the effects of a few illustrative manipulations of these parameters below. As a general principle, the effects of these manipulations make sense in light of the theoretical framework described in the paper.

For example, the parameter “DRevMag” controls the size of the negative “dip” in the U-shaped curve (more negative values = a larger dip). Given that this negative dip is essential for severing weights to competitors and causing differentiation, shifting DRevMag upwards towards zero should shift the balance of the model away from differentiation and towards integration. This is indeed what we observe, as shown in this parameter sweep from the Chanales simulation:

**Author response image 5. sa4fig5:** 

As another example: The “DRev” parameter controls where the U-shaped curve transitions from negative weight change to positive weight change. Lower values of DRev mean that the region of coactivity values leading to negative weight change will be smaller, and the region of coactivity values leading to positive weight change will be larger. As such, we would expect that lower values of DRev would bias the model toward integration. That is indeed the case, as shown in this parameter sweep from the Schlichting Blocked simulation:

**Author response image 6. sa4fig6:** 

There are a few other points that may limit the model's ability to clearly map onto or make predictions about empirical data. The model(s) seems very keen to integrate and do so more completely than the available empirical data suggest. For instance, there is a complete collapse of representations in half of the simulations in the Chanales et al model and the blocked simulation in the Schlichting et al model also seems to produce nearly complete integration Even if the Chanales et al paper had observed some modest behavioral attraction effects, this model would seem to over-predict integration. The author's somewhat implicitly acknowledge this when they discuss the difficulty of producing differentiation ("Practical Advice for Getting the Model to Show Differentiation") and not of producing integration, but don't address it head on.

We thank the reviewer for this comment – R1 had a similar comment. We have added a new section to the Discussion to address this point (p. 30):

“Reconciling the Prevalence of Differentiation in the Model and in the Data.

A key lesson from our model is that, from a computational perspective, it is challenging to obtain differentiation effects: The region of parameter space that gives rise to differentiation is much smaller than the one that gives rise to integration (for further discussion of this issue, see the section in Methods on Practical Advice for Getting the Model to Show Differentiation). However, the fact that integration is more prevalent in our simulations across parameter configurations does not mean that integration will be more prevalent than differentiation in real-life circumstances. What really matters in predicting the prevalence of differentiation in real life is how the parameters of the brain map on to parameters of the model: If the parameters of the brain align with regions of model parameter space that give rise to differentiation (even if these regions are small), this would explain why differentiation has been so robustly observed in extant studies. Indeed, this is exactly the case that we sought to make above about the hippocampus – i.e., that its use of especially sparse coding and a high learning rate will give rise to the kinds of neural dynamics that cause differentiation (as opposed to integration). As another example, while it is true that half of the overlap conditions in our simulation of Chanales et al. (2021) give rise to integration, this does not imply that integration will occur half of the time in the Chanales et al. (2021) study; it may be that the levels of overlap that are actually observed in the brain in Chanales et al. (2021) are more in line with the levels of overlap that give rise to differentiation in our model.”

Second, the authors choice of strongly prewiring associations in the Chanales and Favila models makes it difficult to think about how their model maps onto experimental contexts where competition is presumably occurring while associations are only weakly learned. In the Chanales et al paper, for example, the object-face associations are not well learned in initial rounds of the color memory test. While the authors do justify their modeling choice and their reasons have merit, the manipulation of AX association strength in the Schlichting et al model also makes it clear that the association strength has a substantial effect on the model output. Given the effect of this manipulation, more clarity around this assumption for the other two models is needed.

We thank the reviewer for bringing this up. We have edited the section entitled “A Note on Prewiring Representations” in the Methods to further justify our choice to prewire associations in the Chanales and Favila models (p. 37):

“In our model, our practice of ``prewiring'' memory representations for the A and B pairmates serves two functions. In some cases, it is meant to stand in for actual training (as in the blocked / interleaved manipulation; the connections supporting the AX association are prewired to be stronger in the blocked condition than in the interleaved condition). However, the other, more fundamental role of prewiring is to ensure that the A and B input patterns evoke sparse distributed representations in the hidden layer (i.e., where some units are strongly active but most other units are inactive). In the real brain, this happens automatically because the weight landscape has been extensively sculpted by both experience and evolution. For example, in the real hippocampus, when the second pairmate is presented for the first time, it will evoke a sparse distributed representation in the CA3 subfield (potentially overlapping with the first pairmate’s CA3 representation) even before any learning of the second pairmate has occurred, due to the strong, sparse mossy fiber projections that connect the dentate gyrus to CA3 (McNaughton & Morris, 1987). As discussed above, we hypothesize that this initial, partial overlap between the second pairmate’s representation and the first pairmate’s representation can lead to pop-up of the unique features of the first pairmate’s representation, triggering learning that leads to differentiation or integration. In our small-scale model, we are effectively starting with a ``blank brain''; in the absence of prewiring, the A and B inputs would activate overly diffuse representations that do not support these kinds of competitive dynamics. As such, prewiring in our model is necessary for proper functioning. The presence of prewired A and B representations should therefore not be interpreted as reflecting a particular training history (except in the blocked / interleaved case above); rather, these prewired representations constitute the minimum step we would take to ensure well-defined competitive dynamics in our small-scale model.

The fact that connection strengths serve this dual function – sometimes reflecting effects of training (as in our simulation of Schlichting et al., 2015) and in other cases reflecting necessary prewiring – complicates the interpretation of these strength values in the model. Our view is that this is a necessary limitation of our simplified modeling approach – one that can eventually be surmounted through the use of more biologically-detailed architectures (see *Limitations and Open Questions* in the *Discussion*).”

Overall, this is strong and clearly described work that is likely to have a positive impact on computational and empirical work in learning and memory. While the authors have written about some of the ideas discussed in this paper previously, a fully implemented and openly available model is a clear advance that will benefit the field. It is not easy to translate a high-level description of a learning rule into a model that actually runs and behaves as expected. The fact that the authors have made all their code available makes it likely that other researchers will extend the model in numerous interesting ways, many of which the authors have discussed and highlighted in their paper.
**Reviewer #3 (Public Review):**
This paper proposes a computational account for the phenomenon of pattern differentiation (i.e., items having distinct neural representations when they are similar). The computational model relies on a learning mechanism of the nonmonotonic plasticity hypothesis, fast learning rate and inhibitory oscillations. The relatively simple architecture of the model makes its dynamics accessible to the human mind. Furthermore, using similar model parameters, this model produces simulated data consistent with empirical data of pattern differentiation. The authors also provide insightful discussion on the factors contributing to differentiation as opposed to integration. The authors may consider the following to further strengthen this paper:The model compares different levels of overlap at the hidden layer and reveals that partial overlap seems necessary to lead to differentiation. While I understand this approach from the perspective of modeling, I have concerns about whether this is how the human brain achieves differentiation. Specifically, if we view the hidden layer activation as a conjunctive representation of a pair that is the outcome of encoding, differentiation should precede the formation of the hidden layer activation pattern of the second pairmate. Instead, the model assumes such pattern already exists before differentiation. Maybe the authors indeed argue that mechanistically differentiation follows initial encoding that does not consider similarity with other memory traces?Related to the point above, because the simulation setup is different from how differentiation actually occurs, I wonder how valid the prediction of asymmetric reconfiguration of hidden layer connectivity pattern is.

We thank the reviewer for this comment. In the revised manuscript, we have edited the “Note on Prewiring Representations” in the Methods to clarify how our assumptions about prewiring relate to what we really think is happening in the brain (p. 37):

“In our model, our practice of ``prewiring'' memory representations for the A and B pairmates serves two functions. In some cases, it is meant to stand in for actual training (as in the blocked / interleaved manipulation; the connections supporting the AX association are prewired to be stronger in the blocked condition than in the interleaved condition). However, the other, more fundamental role of prewiring is to ensure that the A and B input patterns evoke sparse distributed representations in the hidden layer (i.e., where some units are strongly active but most other units are inactive). In the real brain, this happens automatically because the weight landscape has been extensively sculpted by both experience and evolution. For example, in the real hippocampus, when the second pairmate is presented for the first time, it will evoke a sparse distributed representation in the CA3 subfield (potentially overlapping with the first pairmate’s CA3 representation) even before any learning of the second pairmate has occurred, due to the strong, sparse mossy fiber projections that connect the dentate gyrus to CA3 (McNaughton & Morris, 1987). As discussed above, we hypothesize that this initial, partial overlap between the second pairmate’s representation and the first pairmate’s representation can lead to pop-up of the unique features of the first pairmate’s representation, triggering learning that leads to differentiation or integration. In our small-scale model, we are effectively starting with a ``blank brain''; in the absence of prewiring, the A and B inputs would activate overly diffuse representations that do not support these kinds of competitive dynamics. As such, prewiring in our model is necessary for proper functioning. The presence of prewired A and B representations should therefore not be interpreted as reflecting a particular training history (except in the blocked / interleaved case above); rather, these prewired representations constitute the minimum step we would take to ensure well-defined competitive dynamics in our small-scale model.

The fact that connection strengths serve this dual function – sometimes reflecting effects of training (as in our simulation of Schlichting et al., 2015) and in other cases reflecting necessary prewiring – complicates the interpretation of these strength values in the model. Our view is that this is a necessary limitation of our simplified modeling approach – one that can eventually be surmounted through the use of more biologically-detailed architectures (see *Limitations and Open Questions* in the *Discussion*).”

Although as the authors mentioned, there haven't been formal empirical tests of the relationship between learning speed and differentiation/integration, I am also wondering to what degree the prediction of fast learning being necessary for differentiation is consistent with current data. According to Figure 6, the learning rates lead to differentiation in the 2/6 condition achieved differentiation after just one-shot most of the time. On the other hand, For example, Guo et al (2021) showed that humans may need a few blocks of training and test to start showing differentiation.

We thank the reviewer for mentioning this. We have added a paragraph to the “Differentiation Requires a High Learning Rate and Is Sensitive to Activity Dynamics” section of the Discussion that addresses this point (pp. 28-29):

“Although the results from Wanjia et al. (2021) provide strong support for the model's prediction that differentiation will be abrupt, they raise another question: What explains variance across items in *when* this abrupt change takes place? The answer to this question remains to be seen, but one possibility is encoding variability: If we assume that participants stochastically sample (i.e., attend to) the features of the scene pairmates, it is possible that participants might initially fail to sample the features that distinguish the scene pairmates, which can be quite subtle – and if the distinguishing features of the pairmates are not represented in high-level visual regions (i.e., the pairmates are represented in these regions as having the same features), this could delay the onset of differentiation until the point at which the distinguishing features happen (by chance) to be sampled.”

Related to the point above, the high learning rate prediction also seems to be at odds with the finding that the cortex, which has slow learning (according to the theory of complementary learning systems), also shows differentiation in Wammes et al (2022).

We now address this point in the section of the Discussion entitled “Differentiation Requires a High Learning Rate and Is Sensitive to Activity Dynamics” (p. 27):

“Our finding that differentiation requires a high learning rate suggests that differentiation will be more evident in the hippocampus than in neocortex, insofar as hippocampus is thought to have a higher learning rate than neocortex (McClelland et al., 1995). In keeping with this prediction, numerous studies have found differentiation effects in hippocampus but not in neocortical regions involved in sensory processing (e.g., Chanales et al., 2017; Favila et al., 2016; Zeithamova et al., 2018). At the same time, some studies have found differentiation effects in neocortex (e.g., Schlichting et al., 2015; Wammes et al., 2022). One possible explanation of these neocortical differentiation effects is that they are being ``propped up’’ by top-down feedback from differentiated representations in the hippocampus.”

More details about the learning dynamics would be helpful. For example, equation(s) showing how activation, learning rate and the NMPH function work together to change the weight of connections may be added. Without the information, it is unclear how each connection changes its value after each time point.

We thank the reviewer for this comment. We have made two major changes to address this concern. First, we have edited the “Learning” section within “Basic Network Properties” in the main text (pp. 6-7):

“Connection strengths in the model between pairs of connected units *x* and *y* were adjusted at the end of each trial (i.e., after each stimulus presentation) as a U-shaped function of the coactivity of *x* and *y*, defined as the product of their activations on that trial. The parameters of the U-shaped learning function relating coactivity to change in connection strength (i.e., weakening / strengthening) were specified differently for each projection where learning occurs (bidirectionally between the input and hidden layers, the hidden layer to itself, and the hidden to output layer). Once the U-shaped learning function for each projection in each version of the model was specified, we did not change it for any of the various conditions. Details of how we computed coactivity and how we specified the U-shaped function can be found in the *Methods* section.”

Second, we have added the requested equations to the “Learning” part of the Methods (pp. 37-38):

The right side of the function, strong activation leads to strengthening of the connectivity, which I assume will lead to stronger activation on the next time point. The model has an upper limit of connection strength to prevent connection from strengthening too much. The same idea can be applied to the left side of the function: instead of having two turning points, it can be a linear function such that low activation keeps weakening connection until the lower limit is reached. This way the NMPH function can take a simpler form (e.g., two line-segments if you think the weakening and strengthening take different rates) and may still simulate the data.

We thank the reviewer for mentioning this. We have added a new paragraph in the “Learning” section of the Methods to justify the particular shape of the learning curve (pp. 38-39):

“Evidence for the U-shaped plasticity function used here (where low activation leads to no change, moderate activation leads to weakening, and higher levels of activation lead to strengthening) was previously reviewed in Ritvo et al. (2019). In brief, there are three lines of work that support the U shape: First, multiple neurophysiological studies have found that moderate postsynaptic depolarization leads to synaptic weakening and higher levels of depolarization lead to synaptic strengthening (e.g., Artola et al., 1990; Hansel et al., 1996). Second, human neuroscience studies have used pattern classifiers, applied to fMRI and EEG data, to measure memory activation, and have related this measure to subsequent memory accessibility; several studies using this approach have found that low levels of activation lead to no change in memory strength, moderate levels of activation lead to impaired subsequent memory, and higher levels of activation lead to increased subsequent memory (e.g., Newman and Norman, 2010; Detre et al., 2013; Kim et al., 2014; for related findings, see Lewis-Peacock and Norman, 2014; Wang et al., 2019). Third, a recent human fMRI study by Wammes et al. (2022) manipulated memory activation by varying the visual similarity of pairmates and observed a U-shaped function relating visual similarity to representational change in the hippocampus, whereby low levels of pairmate similarity were associated with no change, moderate levels of similarity were associated with differentiation, and the differentiation effect went away at higher levels of similarity.

We have also included a pointer to this new paragraph in the “Nonmonotonic Plasticity Hypothesis” section of Introduction (p. 2):

(for further discussion of the empirical justification for the NMPH, see the Learning subsection in the *Methods*)”

**Recommendations for the authors:**

**Reviewer #1 (Recommendations For The Authors):**
A few additional minor things about data presentation and the like:(1) Figure 1 legend - a more general description of how to interpret the figure might be helpful for more naive readers (e.g., explaining how one can visualize in the schematic that there is overlap in the hidden layer between A and B). Also, from the Figure 1 depiction, it's not clear what is different about the setup from the initial left hand side panels in A, B, C, to make it such that activity spreads strongly to A in panel A, weakly in panel B, and not at all in panel C since the weights are the same. Is there a way to incorporate this into the graphic, or describe it in words?

To address this point, we have added the following text to the Figure 1 caption (p. 3):

“Note that the figure illustrates the consequences of differences in competitor activation for learning, without explaining why these differences would arise. For discussion of circumstances that could lead to varying levels of competitor activation, see the simulations described in the text.”

(2) I believe not all of the papers cited on lines 193-195 actually have similarity manipulations in them. I'd recommend double checking this list and removing those less relevant to the statement.

Thank you for pointing this out; we have removed the Ballard reference and we have clarified what we mean by similarity reversal (p. 7):

“The study was inspired by recent neuroimaging studies showing ``similarity reversals'', wherein stimuli that have more features in common (or share a common associate) show less hippocampal pattern similarity (Favila et al., 2016; Schlichting et al., 2015; Molitor et al., 2021; Chanales et al., 2017; Dimsdale-Zucker et al., 2018; Wanjia et al., 2021; Zeithamova et al., 2018; Jiang et al., 2020; Wammes et al., 2022).”

(3) I wanted a bit more detail about how the parameters were set in the main paper, not just in the methods. Even something as brief as noting that model fitting was done by hand by tweaking parameters to re-create the empirical patterns (if I'm understanding correctly) would have been helpful for me.

To address this point, we have added the following text under “Basic Network Properties” (p. 4):

“Our goal was to qualitatively fit key patterns of results from each of the aforementioned studies. We fit the parameters of the model by hand as they are highly interdependent (see the *Methods* section for more details).”

(4) In Figure 4E, it would be helpful to describe the x and y axes of the MDS plots in the legend.

To address this point, we have added the following new text to the Figure 4 caption that clarifies how the MDS plots were generated (p. 11):

“MDS plots were rotated, shifted, and scaled such that pairmate 1before is located at (0,0), pairmate 2before is located directly to the right of pairmate 1before, and the distance between pairmate 1before and pairmate 2before is proportional to the baseline distance between the pairmates.”

(5) Figure 6 - at first I thought the thicker line was some sort of baseline, but I think it is just many traces on top of one another. If other readers may be similarly confused, perhaps this could be stated.

Thanks for this comment. We have updated Figure 6 (p. 16).

We have also updated the caption.

I am having a lot of difficulty understanding the terms "competitor-to-competitor,""competitor-to-target/shared," and "target/shared-to-target/shared," and therefore I don't fully get Figure 5. I think it might be helpful to expand the description of these terms where they are first introduced in the paper (p. 13?). I think I am missing something crucial here, and I am not quite sure what that is-which I know is not very helpful! But, to narrate my confusion a bit, I thought that these terms would somehow relate to connections between different connections of the network. For example is competitor-to-competitor within the hidden layer? Or is this somehow combining across relevant connections that might span different pairs of layers in the model? And, I really have no idea why it is "target/shared."

Thank you for these comments. We have updated Figure 5 and we have also made several changes to the main text and the figure caption to address these points.

Changes to the main text (p. 13):

“Whether symmetric or asymmetric integration occurs depends on the relative strengths of connections between pairs of unique competitor units (*competitor-competitor connections*) compared to connections between unique competitor units and shared units (*competitor-shared connections*) after the first trial (Figure 5; note that the figure focuses on connections between hidden units, but the principle also applies to connections that span across layers). Generally, coactivity between unique competitor units (*competitor-competitor coactivity*) is less than coactivity between unique competitor units and shared units (*competitor-shared coactivity*), which is less than coactivity between unique target units and shared units (*target-shared coactivity*).”

(7) Relatedly in Figure 13, I understand how some competitor-to-target/shared connections could be spared in the bottom instance given panel B. However, I'm struggling to understand how that relates to the values in the corresponding chart in panel A. What about panel A, bottom (vs. the top) means lower coactivities between some competitor-to-target/shared? Is it because if the noise level is higher, the "true" activation of competitor-to-target/shared connections is weaker? I think again, I'm missing something critical here! and wonder if other readers may be in the same situation. (I know the authors described this also on p. 36, but I'm still confused!)

We have updated Figure 13 to clarify these points.

(8) In Figure 9, I believe there is no caption for panel D. Also, it looks as though the item unit active for A and B is the same. I wonder if this is an error?

Thank you for catching these errors! They have both been fixed.

**Reviewer #2 (Recommendations For The Authors):**
-Perhaps I missed it, but I think defining coactivity (how it is computed) in the main text would be useful for readers, as this is critical for understanding the model. I did find it in the methods.

We thank the reviewer for this suggestion. We have updated the “Learning” section within “Basic Network Properties” in the main text to address this point (pp. 6-7):

“Connection strengths in the model between pairs of connected units *x* and *y* were adjusted at the end of each trial (i.e., after each stimulus presentation) as a U-shaped function of the coactivity of *x* and *y*, defined as the product of their activations on that trial. The parameters of the U-shaped learning function relating coactivity to change in connection strength (i.e., weakening / strengthening) were specified differently for each projection where learning occurs (bidirectionally between the input and hidden layers, the hidden layer to itself, and the hidden to output layer). Once the U-shaped learning function for each projection in each version of the model was specified, we did not change it for any of the various conditions. Details of how we computed coactivity and how we specified the U-shaped function can be found in the *Methods* section.”

-The modeling results in the different face condition are at odds with the data for the Favila et al model (they observe some differentiation in the paper and the model predicts no change). This could be due to a number of unmodeled factors, but it is perhaps worth noting.

Thank you for pointing this out. It is possible to better capture the pattern of results observed by Favila et al. in their paper (with some differentiation in the different-face condition and even more differentiation in the same-face condition) by slightly adjusting the model parameters (specifically, by setting the oscillation amplitude *Osc* for the hidden layer to .1 instead of .067).

Rather than replacing the old (*Osc* = .067) results in the paper, which would entail re-making the associated videos, etc., we have added a supplementary figure (Figure 8 - Supplement 1; see p.45):

We also added new text to the Favila Results, under “Differentiation and Integration” (p. 20):

“Note also that the exact levels of differentiation that are observed in the different-face and same-face conditions are parameter dependent; for an alternative set of results showing some differentiation in the different-face condition (but still less than is observed in the same-face condition), see Figure 8 - Supplement 1.”

-Related to my comment in the public review about pre-wiring associations, in the caption for Figure 9 (Schlichting model), the authors report "In both conditions, the pre-wired connection linking the "item B" hidden units to the "item X" output unit is set to .7. In the interleaved condition, the connection linking the "item A" hidden units to the "item X" output unit is set to .8, to reflect some amount of initial AX learning. In the blocked condition, the connection linking the "item A" hidden units to the "item X" output unit is set a higher value (.999), to reflect extra AX learning." What are the equivalent values for the other models, especially the Favila model since the structure is the same as Schlichting? I understood all the "strong" connections to be .99 unless otherwise stated. If that's the case, I don't understand why the blocked Schlichting model and the Favila model produce opposite effects. More clarity would be useful here.

We have added a new paragraph to the results section for the Schlicting model (under “Differentiation and Integration”) to clarify why the blocked Schlichting model and the Favila model show different results (p. 24):

“Note that the key feature driving integration in the blocked condition of this simulation is not the high strength of the connection from X to A *on its own* – rather, it is the *asymmetry* in the pretrained connection strengths from X to A (.999) and from X to B (.7). This asymmetry, which is meant to reflect the extensive training on A-X that occurred before the initial presentation of B-X, results in the A-X hidden representation decisively winning the competition during B-X presentation, which then leads to the B input also being linked to this representation (i.e., integration). It is instructive to compare this to the same-face condition from our simulation of Favila et al. (2016): In that simulation, the two pairmates are also linked strongly (.99 initial connection strength) to a shared associate, but in that case the connections are equally strong, so there is more balanced competition -- in this case, the competitor representation only comes to mind moderately (instead of displacing the target representation), so the result is differentiation instead of integration.”

-The meaning of the different colored dots in Figure 5 is bit hard to keep track of, even given the legend labels. The figure might benefit from a model sketch highlighting each of the different coactivity types. The left side of Fig 13 was useful but again somehow mapping on the colors would help further. Another note on these figures: what does having two dots of each color mean? Is it just an illustration of the variance? There would be more dots if there was one dot per coactivity value.

We have updated Figure 5 and Figure 13 to clarify these points (including a clarification that the dots only represent a subset of the possible pairings between units).

-While I appreciate the goal of the paper is to account for these three studies, readers who aren't familiar with or specifically interested in these studies may appreciate a small amount of intuition on why formalizing unsupervised learning models may be broadly important for computational investigations of learning/memory/cognition.

We have added the following text under “Basic Network Properties” in the Introduction to address this point (p. 4):

“Achieving a better understanding of unsupervised learning is an important goal for computational neuroscience, given that learning agents have vastly more opportunities to learn in an unsupervised fashion than from direct supervision (for additional discussion of this point, see, e.g., Zhuang et al., 2021).”